# FEDERATED OFFLINE POLICY LEARNING WITH HETEROGENEOUS OBSERVATIONAL DATA

## ABSTRACT

We consider the problem of learning personalized decision policies from observational bandit feedback data across multiple heterogeneous data sources. Moreover, we examine the practical considerations of this problem in the federated setting where a central server aims to train a policy on data distributed across the heterogeneous sources, or clients, without collecting any of their raw data. We present a policy learning algorithm amenable to federation based on the aggregation of local policies trained with doubly robust offline policy evaluation and learning strategies. We provide a novel regret analysis for our approach that establishes a finite-sample upper bound on a notion of global regret against a mixture distribution of clients. In addition, for any individual client, we establish a corresponding local regret upper bound characterized by measures of relative distribution shift to all other clients. Our analysis and supporting experimental results provide insights into tradeoffs in the participation of heterogeneous data sources in policy learning.

## 1 INTRODUCTION

Offline policy learning from observational bandit feedback data is an effective approach for learning personalized decision policies in applications where obtaining online, real-time data is impractical (Swaminathan & Joachims, 2015; Kitagawa & Tetenov, 2018; Athey & Wager, 2021). Typically, the observational data used in offline policy learning is assumed to originate from a single source distribution. However, in practice, we often have the opportunity to leverage multiple datasets collected from various experiments under different populations, environments, or logging policies. For instance, a healthcare policymaker may have access to data from multiple hospitals that conduct different types of clinical trials on distinct patient populations. Learning from multiple heterogeneous observational datasets, with their more diverse and extensive coverage of the decision space, may lead to better personalized decision policies, assuming sufficient generalization across data sources.

However, several practical constraints, such as privacy concerns, legal restrictions, proprietary interests, or competitive barriers, can hinder the consolidation of datasets across sources. Federated learning (Kairouz et al., 2021) presents a potential solution to such obstacles by offering a framework for training machine learning models in a decentralized manner, thereby minimizing systemic risks associated with traditional, centralized machine learning. Federated learning techniques applied to policy learning can enable platforms to learn targeted policies without centrally storing sensitive information. It also has the potential to incentivize institutions to collaborate on developing policies that are more generalizable across diverse environments without having to share sensitive data, such as clinical patient data in hospitals.

In this work, we formally introduce the problem of learning personalized decision policies on observational bandit feedback data from multiple heterogeneous data sources. Moreover, we examine the practical considerations of this problem in the federated setting where a central server aims to train a policy on data distributed across the data sources, or clients, without collecting any of their raw data. For this purpose, we provide a policy learning algorithm amenable to federation based on the federated averaging algorithm with local model updates given by online cost-sensitive classification oracles. We present a novel regret analysis that distinguishes between the global regret of the central server and the local regret of a client. For both regret notions, we provide finite-sample upper bounds characterized by expressions of client heterogeneity. We experimentally verify the effect of client heterogeneity on regret, and we present design choices to overcome local performance degradation

due to distribution shift. Our analysis and supporting experimental results provide insights into the tradeoffs in the participation of heterogeneous data sources in offline policy learning.

## 2 RELATED WORK

**Offline Policy Learning**  Recent years have witnessed substantial progress in offline policy learning from observational bandit feedback data. Swaminathan & Joachims (2015); Kitagawa & Tetenov (2018) established the framework for structured decision policy learning using offline policy evaluation strategies. Athey & Wager (2021) achieved optimal regret rates under unknown propensities through doubly robust estimators. Kallus (2018) found optimal weights for the target policy directly from observational data. Zhou et al. (2023) extended this to the multi-action setting, while Zhan et al. (2021a) considered adaptively collected observational data, ensuring optimal regret guarantees even with diminishing propensities. Jin et al. (2022) relaxed the uniform overlap assumption to partial overlap under the optimal policy. More relevant to our setting under heterogeneous data sources, work by Agarwal et al. (2017); He et al. (2019); Kallus et al. (2021) leveraged data from multiple historical logging policies, although assuming the same underlying populations and environments. Lastly, we mention contextual bandit methods (Li et al., 2010) often utilize offline policy learning oracles (Bietti et al., 2021; Krishnamurthy et al., 2021; Simchi-Levi & Xu, 2022; Carranza et al., 2022), but for developing adaptive action-assignment algorithms in online policy learning.

**Federated Learning**  Kairouz et al. (2021); Wang et al. (2019) offer comprehensive surveys on federated learning and its challenges. Mohri et al. (2019) presented an agnostic supervised federated learning framework, introducing useful concepts like weighted Rademacher complexity and skewness measures, which we use in our work. Wei et al. (2021) established excess risk bounds for supervised federated learning under data heterogeneity. The impact of client heterogeneity in federated learning has been explored by Li et al. (2019; 2020); Karimireddy et al. (2020). Contextual bandits in federated settings have been studied by Agarwal et al. (2020); Dubey & Pentland (2020); Huang et al. (2021); Agarwal et al. (2023). However, offline policy evaluation and learning in federated settings remain relatively underexplored. Xiong et al. (2021) investigated federated methods for estimating average treatment effects across heterogeneous data sources. Zhou et al. (2022) and Shen et al. (2023) delved into federated offline policy optimization in full reinforcement learning settings but with significant limitations, including relying on strong linear functional form assumptions with highly suboptimal rates and requiring a difficult saddle point optimization problem that focuses more on policy convergence rather than establishing regret rates.

## 3 PRELIMINARIES

### 3.1 SETTING

We introduce the problem of offline policy learning from observational bandit feedback data across multiple heterogeneous data sources. Throughout the paper, we refer to a heterogeneous data source as a *client* and the central planner that aggregates client data/models as the *central server*.

Let $\mathcal{X} \subset \mathbb{R}^p$ be the context space, $\mathcal{A} = \{a_1, \ldots, a_d\}$ be the finite action space with $d$ actions, and $\mathcal{Y} \subset \mathbb{R}$ be the reward space. A *decision policy* $\pi : \mathcal{X} \to \mathcal{A}$ is a mapping from the context space $\mathcal{X}$ to actions $\mathcal{A}$. We assume there is a central server and a finite set of clients $\mathcal{C}$, with each client $c \in \mathcal{C}$ possessing a *local data-generating distribution* $\mathcal{D}_c$ defined over $\mathcal{X} \times \mathcal{Y}^d$ which governs how client contexts $X^c$ and client potential reward outcomes $Y^c(a_1), \ldots, Y^c(a_d)$ are generated. Moreover, the central server specifies a fixed distribution $\lambda$ over the set of clients $\mathcal{C}$ describing how clients will be sampled or aggregated[1], which we will simply refer to as the *client sampling distribution*.

The central server seeks to train a decision policy that performs well on the *global data-generating mixture distribution* defined by $\mathcal{D}_\lambda := \sum_{c \in \mathcal{C}} \lambda_c \mathcal{D}_c$. At the same time, if there is a potential target client of interest, the central server may not want this personalized policy to perform poorly on the local distribution of this client, otherwise their locally trained policy may provide greater utility to the client and thus their participation is disincentivized. In the following section, we introduce the exact policy performance measures that capture these two potentially opposing objectives.

---

[1]Clients are sampled in the cross-device FL setting and aggregated in the cross-silo FL settings.

## 3.2 OBJECTIVE

We consider the immediate reward gained by a client by taking actions according to any given policy. Additionally, we extend this metric to a global version that captures the aggregate reward gained from the mixture of clients under the client sampling distribution.

**Definition 1.** The *local policy value* under client $c$ and the *global policy value* under client sampling distribution $\lambda$ of a policy $\pi$ are, respectively,

$$Q_c(\pi) := \underset{Z^c \sim \mathcal{D}_c}{\mathbb{E}}[Y^c(\pi(X^c))] \quad \text{and} \quad Q_\lambda(\pi) := \underset{c \sim \lambda}{\mathbb{E}} \underset{Z^c \sim \mathcal{D}_c}{\mathbb{E}}[Y^c(\pi(X^c))], \tag{1}$$

where the expectations are taken with respect to the corresponding local data-generating distributions $Z^c = (X^c, Y^c(a_1), \dots, Y^c(a_d)) \sim \mathcal{D}_c$ and the client sampling distribution $c \sim \lambda$.

The performance of a policy is typically characterized by a notion of regret against an optimal policy in a specified *policy class* $\Pi \subset \{\pi : \mathcal{X} \to \mathcal{A}\}$, which we assume to be fixed throughout the paper. We define local and global versions of regret based on their respective versions of policy values.

**Definition 2.** The *local regret* under client $c$ and the *global regret* under client sampling distribution $\lambda$ of a policy $\pi$ relative to the given policy class $\Pi$ are, respectively,

$$R_c(\pi) := \max_{\pi' \in \Pi} Q_c(\pi') - Q_c(\pi) \quad \text{and} \quad R_\lambda(\pi) := \max_{\pi' \in \Pi} Q_\lambda(\pi') - Q_\lambda(\pi). \tag{2}$$

The objective of the central server is to determine a policy in the specified policy class $\Pi$ that minimizes global regret. On the other hand, the central server also aims to characterize the corresponding local regret of a target client under the obtained policy, since this quantity captures the client's corresponding individual utility to a global policy.

## 3.3 DATA-GENERATING PROCESSES

We assume each client $c \in \mathcal{C}$ has a *local observational data set* $\{(X_i^c, A_i^c, Y_i^c)\}_{i=1}^{n_c} \subset \mathcal{X} \times \mathcal{A} \times \mathcal{Y}$ consisting of $n_c \in \mathbb{N}$ triples of contexts, actions, and rewards collected using a *local experimental stochastic policy* $e_c : \mathcal{X} \to \Delta(\mathcal{A})$ in the following manner. For the $i$-th data point of client $c$,

1. nature samples a context and potential outcomes vector $(X_i^c, Y_i^c(a_1), \dots, Y_i^c(a_d)) \sim \mathcal{D}_c$;
2. client $c$ is assigned action $A_i^c \sim e_c(\cdot | X_i^c)$;
3. client $c$ observes the realized outcome $Y_i^c = Y_i^c(A_i^c)$ ;
4. client $c$ logs the data tuple $(X_i^c, A_i^c, Y_i^c)$ locally.[2]

We will let $n := \sum_{c \in \mathcal{C}} n_c$ denote the *total sample size* across clients. Note that although the counterfactual reward outcomes $Y_i^c(a)$ for all $a \in \mathcal{A} \backslash \{A_i^c\}$ exist in the local data-generating process, they are not observed in the realized data. All clients only observe the outcomes associated to their assigned treatments. For this reason, such observational data is also referred to as bandit feedback data (Swaminathan & Joachims, 2015).

Given these data-generating processes, it will be useful to introduce the data-generating distributions that also incorporate how actions are sampled. For each client $c \in \mathcal{C}$, the local historical policy $e_c$ induces a *complete local data-generating distribution* $\bar{\mathcal{D}}_c$ defined over $\mathcal{X} \times \mathcal{A} \times \mathcal{Y}^d$ that dictates how the entire local contexts, actions, and potential reward outcomes for all actions were sampled in the local data-generating process, i.e., $(X_i^c, A_i^c, Y_i^c(a_1), \dots, Y_i^c(a_d)) \sim \bar{\mathcal{D}}_c$. Given this construction of the complete client distributions, we also introduce the *complete global data-generating mixture distribution* defined by $\bar{\mathcal{D}}_\lambda := \sum_{c \in \mathcal{C}} \lambda_c \bar{\mathcal{D}}_c$.

## 3.4 DATA ASSUMPTIONS

We make the following standard assumptions on the data-generating process of any given client.

**Assumption 1** (Local Ignorability). *For any client $c \in \mathcal{C}$, the complete local data-generating distribution $(X^c, A^c, Y^c(a_1), \dots, Y^c(a_d)) \sim \bar{\mathcal{D}}_c$ satisfies:*

(a) *Boundedness: The marginal distribution of $\bar{\mathcal{D}}_c$ on the set of potential outcomes $\mathcal{Y}^d$ has bounded support, i.e., there exists some $B_c > 0$ such that $|Y^c(a)| \leq B_c$ for all $a \in \mathcal{A}$.*

---

[2]If $e_c(A_i^c | X_i^c) = \mathbb{P}(A_i^c | X_i^c)$ is known, it also locally logged as it facilitates policy value estimation.

    *(b) Unconfoundedness: Potential outcomes are independent of the observed action conditional on the observed context, i.e., $(Y^c(a_1), \ldots, Y^c(a_d)) \perp\!\!\!\perp A^c \mid X^c$.*

    *(c) Overlap: For any given context, every action has a non-zero probability of being sampled, i.e., there exists some $\eta_c > 0$ such that $e_c(a|x) = \mathbb{P}(A^c = a | X^c = x) \geq \eta_c$ for any $a \in \mathcal{A}$ and $x \in \mathcal{X}$.*

Note that the *boundedness* assumption is not essential and we only impose it for simplicity in our analysis. With additional effort, we can instead rely on light-tail distributional assumptions such as sub-Gaussian potential outcomes as in (Athey & Wager, 2021). *Unconfoundedness* ensures that action assignment is as good as random after accounting for measured covariates, and it is necessary to ensure valid policy value estimation using inverse propensity-weighted strategies. The *uniform overlap* condition ensures that all actions are taken sufficiently many times to guarantee accurate evaluation of any policy. This assumption may not be entirely necessary as recent work (Jin et al., 2022) introduced an approach that does away with the uniform overlap assumption for all actions and only relies on overlap for the optimal policy. However, in our work, we made the above assumptions to simplify our analysis and maintain focus of our contributions on the effects of data heterogeneity on policy learning. In any case, these assumptions are standard and they are satisfied in many experimental settings such as randomized controlled trials or A/B tests.

Next, we also impose the following local data scaling assumption on each client.

**Assumption 2** (Local Data Scaling). *All local sample sizes asymptotically increase with the total sample size, i.e., for each $c \in \mathcal{C}$, $n_c = \Omega(\nu_c(n))$ where $\nu_c$ is an increasing function of the total samples size $n$.*

This assumption states that, asymptotically, the total sample size cannot increase without increasing across all data sources. We emphasize that this assumption is quite benign since $\nu_c$ could be any slowly increasing function (e.g., an iterated logarithm) and the asymptotic lower bound condition even allows step-wise increments. We only impose this assumption to ensure that the regret bounds in our analysis scale with respect to the total sample size with sensible constants. However, it does come at the cost of excluding scenarios in which a client always contributes $O(1)$ amount of data relative to the total data, no matter how much more total data is made available in aggregate, in which case one may expect it is better to exclude any such client.

## 4  APPROACH

The approach for the central server is to use the available observational data to construct an appropriate estimator of the global policy value and use this estimator to find an optimal global policy.

### 4.1  NUISANCE PARAMETERS

We define the following functions which we refer to as *nuisance parameters* as they will be required to be separately known or estimated in the policy value estimates.

**Definition 3.** The local *conditional response* and *inverse conditional propensity* functions of client $c \in \mathcal{C}$ with complete local data-generating distribution $(X^c, A^c, Y^c(a_1), \ldots, Y^c(a_d)) \sim \bar{\mathcal{D}}_c$ are defined, respectively, for any $x \in \mathcal{X}$ and $a \in \mathcal{A}$ as

$$\mu_c(x; a) := \mathbb{E}[Y^c(a)|X^c = x] \quad \text{and} \quad w_c(x; a) := 1/\mathbb{P}(A^c = a \mid X^c = x). \quad (3)$$

For notational convenience, we let $\mu_c(x) = (\mu_c(x; a))_{a \in \mathcal{A}}$ and $w_c(x) = (w_c(x; a))_{a \in \mathcal{A}}$.

In our estimation strategy, each client must estimate the conditional response and inverse conditional propensity functions when they are unknown. Following the literature on double machine learning Chernozhukov et al. (2018), we make the following high-level assumption on the estimators of these local nuisance parameters.

**Assumption 3.** *For any client $c \in \mathcal{C}$, the local estimates $\hat{\mu}_c$ and $\hat{w}_c$ of the nuisance parameters $\mu_c$ and $w_c$, respectively, trained on $m$ local data points satisfy the following squared error bound:*

$$\mathbb{E}\big[\,\|\hat{\mu}_c(X^c) - \mu_c(X^c)\|_2^2\,\big] \cdot \mathbb{E}\big[\,\|\hat{w}_c(X^c) - w_c(X^c)\|_2^2\,\big] \leq \frac{o(1)}{m}, \quad (4)$$

*where the expectation is taken with respect to the marginal distribution of $\mathcal{D}_c$ over contexts.*

We emphasize that this is a standard assumption in the double machine learning literature, and we can easily construct estimators that satisfy these rate conditions, given sufficient regularity on the nuisance parameters (Zhou et al., 2023). They can be estimated with widely available out-of-the-box regression and classification implementations. See Appendix F.1 for more details. Moreover, this condition is general and flexible enough to allow one to trade-off the accuracies of estimating the nuisance parameters. This is an important property in offline policy learning where distribution shift in the batch data can complicate consistent reward estimation.

## 4.2 POLICY VALUE ESTIMATOR

For any client $c \in \mathcal{C}$, we define the *local augmented inverse propensity weighted* (AIPW) score for each $a \in \mathcal{A}$ to be

$$\Gamma^c(a) := \mu_c(X^c; a) + \big(Y^c(A^c) - \mu_c(X^c; a)\big) w_c(X^c; a) \mathbf{1}\{A^c = a\}, \tag{5}$$

where $(X^c, A^c, Y^c(a_1), \ldots, Y^c(a_d)) \sim \bar{\mathcal{D}}_c$. One can readily show that $Q_c(\pi) = \mathbb{E}_{\bar{\mathcal{D}}_c}[\Gamma^c(\pi(X^c))]$, and therefore $Q_\lambda(\pi) = \mathbb{E}_\lambda \mathbb{E}_{\bar{\mathcal{D}}_c}[\Gamma^c(\pi(X^c))]$ (see proof in Lemma 6). Accordingly, our procedure is to estimate the local AIPW scores and aggregate them to form the global policy value estimator. We assume we have constructed nuisance parameter estimates $\hat{\mu}_c$ and $\hat{w}_c$ that satisfy Assumption 3. Then, for each local data point $(X_i^c, A_i^c, Y_i^c)$ in the observational data set of client $c \in \mathcal{C}$, we define the *approximate local AIPW* score for each $a \in \mathcal{A}$ to be

$$\hat{\Gamma}_i^c(a) := \hat{\mu}_c(X_i^c; a) + \big(Y_i^c - \hat{\mu}_c(X_i^c; a)\big) \hat{w}_c(X_i^c; a) \mathbf{1}\{A_i^c = a\}. \tag{6}$$

Using these estimated scores, we can define the *doubly robust global policy value estimate* to be

$$\hat{Q}_\lambda(\pi) = \mathbb{E}_{c \sim \lambda}[\hat{Q}_c(\pi)], \text{ where } \hat{Q}_c(\pi) = \frac{1}{n_c} \sum_{i=1}^{n_c} \hat{\Gamma}_i^c(\pi(X_i^c)). \tag{7}$$

This estimator is a generalized aggregate version of the doubly robust policy value estimator introduced in the standard offline policy learning setting (Zhou et al., 2023). It is doubly robust in the sense that it is accurate as long as one of nuisance parameter estimates is accurate for each client. Lastly, to ensure we can use the same data to estimate the nuisance parameters and construct the policy value estimates, we utilize a *cross-fitting* strategy locally for each client. See Appendix F.2 for more details on the cross-fitting estimation strategy.

## 4.3 OPTIMIZATION OBJECTIVE

The objective of the central server is to find a policy that maximizes the global policy value estimate:

$$\hat{\pi}_\lambda = \arg\max_{\pi \in \Pi} \hat{Q}_\lambda(\pi). \tag{8}$$

Note that in the centralized setting, this optimization can be done using standard policy optimization techniques (Athey & Wager, 2021) on the centrally accumulated heterogeneous datasets at once. However, as we discussed previously, the centralized collection of datasets can present difficulties in privacy sensitive settings. For this reason, we seek to provide an optimization procedure that is amenable to federated settings to overcome these challenges. In the federated setting, the central server does not have access to client raw data to estimate local policy values nor does it have access to the local policy values; only model updates can be shared through the network. In Section 6, we discuss an optimization procedure for parametric policy classes that manages these constraints.

## 5 REGRET BOUNDS

We establish regret bounds for the global policy solution $\hat{\pi}_\lambda$ to the optimization objective above. Refer to the Appendix for further detailed discussion and proofs of the statements in this section.

### 5.1 COMPLEXITY AND SKEWNESS

First, we introduce important quantities that appear in our regret bounds.

**Policy Class Complexity** The following quantity provides a measure of policy class complexity based on a variation of the classical entropy integral introduced by Dudley (1967), and it is useful in establishing a class-dependent regret bound. See Appendix B.1.1 for more details on its definition.

**Definition 4** (Entropy integral). Let $\mathrm{H}(\pi_1, \pi_2; \tilde{x}) := \frac{1}{\tilde{n}} \sum_{i=1}^{\tilde{n}} \mathbf{1}\{\pi_1(\tilde{x}_i) \neq \pi_2(\tilde{x}_i)\}$ be the Hamming distance between any two policies $\pi_1, \pi_2 \in \Pi$ given a covariate set $\tilde{x} \subset \mathcal{X}$ of size $\tilde{n} \in \mathbb{N}$. The *entropy integral* of a policy class $\Pi$ is

$$\kappa(\Pi) := \int_0^1 \sqrt{\log N_{\mathrm{H}}(\epsilon^2, \Pi)} d\epsilon, \tag{9}$$

where $N_{\mathrm{H}}(\epsilon^2, \Pi)$ is the maximal $\epsilon^2$-covering number of $\Pi$ under the Hamming distance over covariate sets of arbitrary size.

The entropy integral is constant for a fixed policy class, and rather weak assumptions on the class are necessary to ensure it is finite such as sub-exponential growth on its Hamming covering number (Zhou et al., 2023), which is satisfied by many policy classes including parametric and finite-depth tree policy classes. In the binary action setting, the entropy integral of a policy class relates to its VC-dimension with $\kappa(\Pi) = \sqrt{\mathrm{VC}(\Pi)}$, and for $D$-dimensional linear classes $\kappa(\Pi) = \mathcal{O}(\sqrt{D})$.

**Client Skewness** The following quantity measures how far the client sampling distribution is from the *empirical distribution of samples* across clients defined by $\bar{n} := (n_c/n)_{c \in \mathcal{C}}$. This quantity naturally arises in the generalization bounds of weighted mixture distributions (Mansour et al., 2021).

**Definition 5** (Skewness). The *skewness* of a given client sampling distribution $\lambda$ is

$$\mathfrak{s}(\lambda \| \bar{n}) := 1 + \chi^2(\lambda \| \bar{n}), \tag{10}$$

where $\chi^2(\lambda \| \bar{n})$ is the chi-squared divergence of $\lambda$ from $\bar{n}$.

## 5.2 GLOBAL REGRET BOUND

The following result captures a root-$n$ finite-sample bound for the global regret that parallels the optimal regret bounds typically seen in the offline policy learning literature.

**Theorem 1** (Global Regret Bound). *Suppose Assumption 1, 2, and 3 hold. Then, with probability at least $1 - \delta$,*

$$R_\lambda(\hat{\pi}_\lambda) \leq \left( c_1 \kappa(\Pi) + \sqrt{c_2 \log(c_2/\delta)} \right) \sqrt{V \cdot \frac{\mathfrak{s}(\lambda \| \bar{n})}{n}} + o_p \left( \sqrt{\frac{\mathfrak{s}(\lambda \| \bar{n})}{n}} \right), \tag{11}$$

*where $c_1$ and $c_2$ are universal constants and $V = \max_{c \in \mathcal{C}} \sup_{\pi \in \Pi} \mathbb{E}_{\bar{\mathcal{D}}_c} \left[ \Gamma^c(\pi(X^c))^2 \right]$.*

First, note that $V$ captures a notion of the worst-case AIPW score variance across clients. Next, we observe that root-$n$ rate is moderated by a skewness term which can also scale with the total sample size. For example, if $\lambda = \bar{n}$ then $\mathfrak{s}(\lambda \| \bar{n})/n = 1/n$, and if $\lambda = (1, 0, \ldots, 0)$ then $\mathfrak{s}(\lambda \| \bar{n})/n = 1/n_1$. Thus, this skewness-moderated rate smoothly interpolates between the rates one expects from the uniform weighted model and the single source model. Indeed, when clients are identical and $\lambda = \bar{n}$, we recover the bounds from standard policy learning (Zhou et al., 2023). From this observation, it may seem that the best design choice for the client sampling distribution $\lambda$ is the empirical sample distribution $\bar{n}$. However, as we observe in the next section, there are terms in the local regret bounds that introduce trade-offs on the choice of $\lambda$ when considering a specific target client.

## 5.3 LOCAL REGRET BOUND

In this next result, we capture the discrepancy in local and global regret due to client heterogeneity. This result is helpful in understanding the extent at which the global and local regret minimization objectives can be in conflict for a particular target client.

**Theorem 2** (Local Regret Bound). *Suppose Assumption 1 holds. Then, for any client $c \in \mathcal{C}$,*

$$R_c(\hat{\pi}_\lambda) \leq U \cdot \mathrm{TV}(\bar{\mathcal{D}}_c, \bar{\mathcal{D}}_\lambda) + R_\lambda(\hat{\pi}_\lambda), \tag{12}$$

*where $U = 3B/\eta$ with $B = \max_{c \in \mathcal{C}} B_c$ and $\eta = \min_{c \in \mathcal{C}} \eta_c$, and $\mathrm{TV}$ is the total variation distance.*

The first term in this regret bound is inherently irreducible relative to the sample sizes and it is due to distribution shift between the complete local client distribution $\bar{\mathcal{D}}_c$ and the complete global mixture distribution $\bar{\mathcal{D}}_\lambda$. Thus, we can observe how the design choice on the client distribution $\lambda$ must balance a trade-off to achieve low skewness and low expected distribution shift across sources.

Skewing towards the target client will reduce the distribution shift term, but it will moderate the rates in the global regret bound. In our experiments, we take a heuristic approach to obtain a skewed $\lambda$, but this may be optimized as in (Mohri et al., 2019). The constant $U$ in the distribution shift term is defined by the constants in the boundedness and overlap assumptions stated in Assumption 1.[3]

The following result demonstrates how the distribution shift term can be further tensorized into contributions due to distribution shift in the covariates, propensities, and potential outcomes.

**Theorem 3** (Local Distribution Shift Bound). *For any given client $c \in \mathcal{C}$, suppose $(X^c, \vec{Y}^c) \sim \mathcal{D}_c$. We let $p_{X^c}$ denote the marginal distribution of $X^c$ and let $p_{\vec{Y}^c|X^c}$ denote the conditional distribution of $\vec{Y}^c$ given $X^c$. Then, the irreducible distribution shift term in the local regret bound can be further bounded as*

$$\mathrm{TV}(\bar{\mathcal{D}}_c, \bar{\mathcal{D}}_\lambda) \le \mathop{\mathbb{E}}_{k \sim \lambda} \left[ \sqrt{\mathrm{KL}(p_{X^c} || p_{X^k})} + \sqrt{\mathrm{KL}(e_c || e_k)} + \sqrt{\mathrm{KL}(p_{\vec{Y}^c|X^c} || p_{\vec{Y}^k|X^k})} \right], \qquad (13)$$

*where* $\mathrm{TV}$ *is the total variation distance and* $\mathrm{KL}$ *is the Kullback-Leibler divergence.*[4]

This result directly reveals the contribution to local regret from each possible source of distribution shift. If we have prior knowledge that certain components of the data-generating distribution match, then we can claim tighter bounds on the local regret of clients. In summary, the results we presented provide insights into understanding the tradeoffs and value of information in heterogeneous client participation in offline policy learning (see Appendix E for further discussion).

## 6 FEDERATED ALGORITHM

We present a federated algorithm for finding the optimal global policy $\hat{\pi}_\lambda$ for the optimization problem stated in Section 4.3. The standard approach for federated learning is the federated averaging (FedAvg) algorithm for parametric models (Konečnỳ et al., 2016; McMahan et al., 2017), which works iteratively by selecting clients to participate in the training round, locally fine-tuning a parametric model on each client using their own data, and then aggregating local model parameters in the central server via a weighted average. More developed federated algorithms (Li et al., 2019; 2020; Karimireddy et al., 2020) also follow this general framework. Therefore, to make standard federated learning strategies suitable for policy optimization, we consider parametric policy classes $\Pi_\Theta = \{\pi_\theta : \mathcal{X} \to \mathcal{A} \mid \theta \in \Theta\}$, and we construct an iterative parametric policy optimization procedure for the local policy updates in a federated averaging procedure.

First, we observe that the local policy optimization procedure $\arg\max_{\theta \in \Theta} \hat{Q}_c(\pi)$ is equivalent to cost-sensitive multi-class classification (CSMC) (Beygelzimer et al., 2009; Dudik et al., 2011), where the actions are the labels and the AIPW scores are the negative costs for the labels. Therefore, we are able to conduct iterative local policy model updates using widely available online CSMC oracle methods which are often used in policy learning for contextual bandit applications (Agarwal et al., 2014; Bietti et al., 2021). CSMC methods are based on consistent reductions to binary classification (Beygelzimer et al., 2008) or multiple regressions Agarwal et al. (2017) such that the optimal model for the reduced problem leads to an optimal cost-sensitive classifier.

Our federated training procedure works as follows (see Algorithms 1, 2 and 3):

1. *Cross-fitted AIPW*: Prior to policy learning, each client uses a cross-fitting strategy on their local observational data to estimate local nuisance parameters for constructing their local AIPW score estimates. The client then forms a local dataset of contexts and label costs, where the costs are negative AIPW scores. See Algorithm 3 in Appendix F for more details.
2. *FedAvg-CSMC server-side*: The central server initializes a global model and executes FedAvg on the clients, iteratively sending global parameters to clients and updating the global model using a weighted average of the received local model updates. See Algorithm 1.
3. *FedAvg-CSMC client-side*: Each time a client receives global parameters, they initialize their local model with the global parameters and use an online CSMC oracle to update the local model on their data for a fixed number of steps. See Algorithm 2.

---

[3]In Appendix D.3, we provide an alternate bound that does not rely on bounded AIPW scores and instead is scaled by the worst-case AIPW variance, which may be smaller and also appears in our global regret bound.

[4]Note that the last two terms in the expectation of this inequality are conditional KL divergences on $p_{X^c}$. See Appendix D.2 for more details.

| **Algorithm 1** FedAvg-CSMC: Server-Side | **Algorithm 2** FedAvg-CSMC: Client-Side |
|---|---|
| **Require:** clients $\mathcal{C}$, client distribution $\lambda$ | **Require:** local steps $T$, local batch size $B$, |
| 1: Initialize global model parameters $\theta_g$ |     local data $\{(X_i^c, \hat{\Gamma}_i^c(a_1), \ldots, \hat{\Gamma}_i^c(a_d))\}_{i=1}^{n_c}$ |
| 2: **for** each round $t = 1, 2, \ldots$ **do** | 1: Receive global parameters $\theta_g$ from server |
| 3:     Sample a subset of clients $\mathcal{S} \subset \mathcal{C}$ | 2: Initialize local parameters $\theta_c \leftarrow \theta_g$ |
| 4:     **for** each client $c \in \mathcal{S}$ **in parallel do** | 3: **for** $t = 1, \ldots, T$ **do** |
| 5:         Send global parameters $\theta_g$ to client $c$ | 4:     $\mathcal{B} \leftarrow$ sample a batch of $B$ local examples |
| 6:         Await local updates $\theta_c$ from client $c$ | 5:     Update local parameters using CSMC oracle: |
| 7:     **end for** |         $\theta_c \leftarrow \text{CSMC}(\theta_c, \mathcal{B})$ |
| 8:     Update global parameters: | 6: **end for** |
|         $\theta_g \leftarrow \sum_{c \in \mathcal{S}} \lambda_c \theta_c / \sum_{c \in \mathcal{S}} \lambda_c$ | 7: Send local parameters $\theta_c$ to server |
| 9: **end for** | |

Figure 1: Cost-sensitive multi-class classification federated averaging algorithm.

Note that achieving the optimal policy is necessary to achieve the stated regret bounds. FedAvg-CSMC is guaranteed to converge to the optimal policy if the optimization problem is concave at least, which is only the case for special class of policies such as linear policy classes. However, the stated optimization problem is generally non-concave. Nevertheless, it is still possible to formulate a regret bound for nearly optimal policies under general policy classes. The regret bounds would be modified to include an additive term that captures the policy value optimality gap. Under an appropriate choice of policy class and corresponding optimization procedure, this optimality gap can be insignificant or of the same order of magnitude as the other terms in the regret bounds, especially under moderately to highly heterogeneous environments.

# 7 EXPERIMENTS

For our experiments, we compare empirical local and global regret bounds across different experimental settings involving homogeneous and heterogeneous clients. First, we describe the experimental setup common to all of our experiments.

For the local CSMC optimization procedure, we employ the cost-sensitive one-against-all (CSOAA) implementation in Vowpal Wabbit (Langford et al., 2023). This method performs separate online regressions of costs on contexts for each action, and at inference time, it selects the action with the lowest predicted cost. For the parametric policy class, we consider the class induced by linear scores $\pi_\theta(x) = \arg\max_{a \in \mathcal{A}} \langle \theta_a, x \rangle$ for $\theta \in \Theta = \mathbb{R}^{d \times p}$, and we use the CSOAA implementation with online linear multiple regresssion of costs on contexts. For the environments, we consider the client set $\mathcal{C} = [C]$ where $C = 5$, context space $\mathcal{X} = [-1, 1]^p$ for $p = 10$, and action set $\mathcal{A} = \{a_1, \ldots, a_d\}$ with $d = 4$. For any client $c \in \mathcal{C}$, we consider the following data-generating process: $X^c \sim \text{Normal}(0, I_p)$, $A^c \sim \text{Uniform}(\mathcal{A})$, and $Y^c(a) \sim \text{Normal}(\mu_c(X^c; a), \sigma^2)$ for all $a \in \mathcal{A}$, where the choice of reward functions $\mu_c(X^c, a)$ are specified in each experiment below. Thus, any heterogeneities we impose between clients will be solely in their outcome distributions. We found this to be the clearest choice to show empirical differences. We will be evaluating how the different regrets scale with total sample size. Therefore, for a given total sample size $n \in \mathbb{N}$, each client $c \in \mathcal{C}$ is allocated a local sample sample size determined by some function $n_c = \nu_c(n)$. To illustrate the benefits of federation under sample size heterogeneity, we will have client $c = 1$ contribute significantly less data than the others with $n_1 = \nu_1(n) = \lfloor \log n \rfloor$ and all other clients will evenly distribute the rest of the total sample size. Therefore, we will focus on the regret profile of client $c = 1$. Clearly, this entire data-generating process satisfies Assumptions 1, 2, and 3. For our results, we consider a training sample size grid in the range up to $N$ samples, where $N = 1\text{K}$ for the homogeneous experiments and $N = 10\text{K}$ for the heteregeneous experiments (due to slower convervence). For each sample size $n$ in our grid, we sample $n_c = \nu_c(n)$ training samples from each client distribution $\bar{\mathcal{D}}_c$ we constructed. Moreover, we sample an additional 10K test samples for each client. We train the global model using our FedAvg-CSMC algorithm on all clients. For baseline comparison, we also train a local model with the same number $n$ of total samples from $\bar{\mathcal{D}}_c$.

**Homogeneous Clients** First, we consider the homogeneous setting where all clients are identical. For every $c \in \mathcal{C}$, we set $\mu_c(x; a) = \langle \theta_a, x \rangle + h(x)$ where the $\theta_a$ are sufficiently separated random vectors in $\Theta = [-1, 1]^p$ and $h(x) = -\mathbf{1}\{x_1 > 0\} \cdot \max_{a' \in \mathcal{A}} \langle \theta_{a'}, x \rangle$ is step-wise constant function to make the reward function non-linear and thus misspecified under a linear model class.

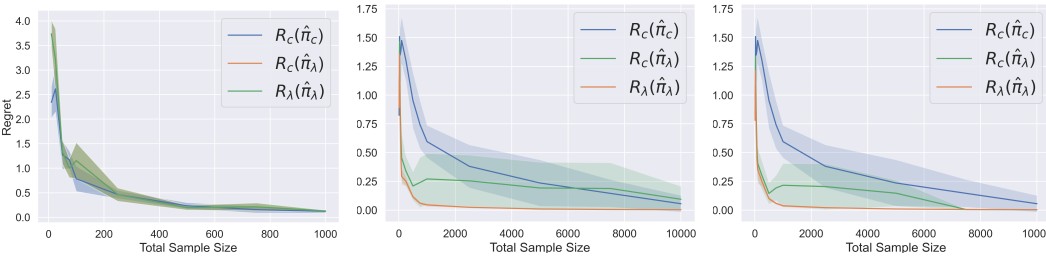

(a) Homogeneous clients, $\lambda = \bar{n}$   (b) Heterogeneous clients, $\lambda = \bar{n}$   (c) Heterogeneous clients, $\lambda = \bar{n} + \bar{\varepsilon}$

Figure 2: Empirical regret curves for simulation experiments. Local regrets are for client 1.

In order to be able to compute the empirical global and local regret on the test data, we first learn the best policy by generating 100K noise-free data samples and directly using the CSMC oracle on the contexts and negative rewards as costs. Figure 2a plots the local regret $R_1(\hat{\pi}_\lambda)$ of the globally trained policy (green) and the global regret $R_\lambda(\hat{\pi}_\lambda)$ of the globally trained policy (orange), all using the empirical mixture $\lambda = \bar{n}$. For comparison, we also plot the local regret $R_1(\hat{\pi}_1)$ of the locally trained policy (blue). The bands show the one standard deviation from the regrets over five different seed runs. As expected, each of these curves is nearly identical since the global and local regrets are identical in this scenario. What is interesting to notice here, however, is that although $\hat{\pi}_\lambda$ only used $n_1 = \lfloor \log n \rfloor$ samples from client 1, $R_1(\hat{\pi}_\lambda)$ has the same profile as $R_1(\hat{\pi}_1)$ using $n$ samples from client 1. This reinforces the fact that federated learning can succesfully leverage data across clients to efficiently learn a policy that matches the performance of a locally trained policy with significantly more data.

**Heterogeneous Clients**   Next, we consider a setting where one client is different than all other clients. For client $c = 1$, we set $\mu_1(x; a) = \sin(\langle \theta_a, x \rangle)$ and for every other client $c \in \mathcal{C}\backslash\{1\}$, we set $\mu_c(x; a) = \langle \theta_a, x \rangle + h(x)$ exactly as in the homogeneous setting. The idea behind this choice is that since the sine function is nearly linear in a neighborhood near the zero value of the argument, there is a wide range of contexts where the best action parameter aligns with the context vectors in the same way it does for the step-wise linear reward. Therefore, there is distribution shift between client 1 and all other clients, but there is some amount of similarity that can be exploited.

We run a similar set of experiments as in the homogeneous setting. We compute the empirical global and local regret on the test data. The best local policies are again learned using 100K noise-free samples from the local distributions and directly using the CSMC oracle on the contexts and negative rewards as costs. The optimal global policy is obtained by combining all 100K noise-free samples and applying the CSMC oracle directly to this data, with each sample receiving a weight $\lambda_c$ during training corresponding to its client source. Figure 2b plots the same type of regret curves as in the homogeneous experiment with $\lambda = \bar{n}$. We observe that $R_1(\hat{\pi}_\lambda)$ significantly suffers from distribution shift. In fact, $\hat{\pi}_1$ performs better than $\hat{\pi}_\lambda$ at a sufficiently large sample size. Figure 2c plots similar regret curves, but instead with the global policy trained with a skewed client sampling distribution $\lambda = \bar{n} + \bar{\varepsilon}$ where $\bar{\varepsilon}_c = -\bar{n}_c/2$ for $c \neq 1$ and $\bar{\varepsilon}_1 = (1 - \bar{n}_1)/2$. Here, we observe that $\hat{\pi}_\lambda$ still suffers some amount in terms of local regret, but not to such an extent that $\hat{\pi}_1$ beats it. Moreover, the local regret shift decreases with larger sample size. The idea is that this skewness upscales the distribution of client 1 to diminish the amount of distribution shift of $\bar{\mathcal{D}}_1$ from $\bar{\mathcal{D}}_\lambda$ especially at larger total sample sizes, at the cost of negatively affecting the other more homogeneous clients. To see how the other clients are affected by this design choice modification to favor client 1, refer to the results in Appendix G. The takeaway there is that the other clients have less distribution shift from the average so their performance degradation is lesser under $\lambda = \bar{n}$, but their performance further degrades under the skewed client sampling distribution $\lambda = \bar{n} + \bar{\varepsilon}$.

## 8   CONCLUSION

We studied the problem of offline policy learning from observational bandit feedback data across multiple heterogeneous data sources. Moreover, we considered the practical aspects of this problem in a federated setting to address privacy concerns. We presented a novel regret analysis and supporting experimental results that demonstrate tradeoffs of client heterogeneity on policy performance. In Appendix H, we provide additional discussion on limitations of our work and potential future work.

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

# A  Auxiliary Results

The following known results will be used in our regret bound proofs. See Chapter 2 of Koltchinskii (2011) for discussions of these results.

**Lemma 1** (Hoeffding's inequality). *Let $Z_1, \ldots, Z_n$ be independent random variables with $Z_i \in [a_i, b_i]$ almost surely. For all $t > 0$, the following inequality holds*

$$\mathbb{P}\left(\left|\sum_{i=1}^{n} Z_i - \mathbb{E}[Z_i]\right| \geq t\right) \leq 2\exp\left(-\frac{2t^2}{\sum_{i=1}^{n}(b_i - a_i)^2}\right).$$

**Lemma 2** (Talagrand's inequality). *Let $Z_1, \ldots, Z_n$ be independent random variables in $\mathcal{Z}$. For any class of real-valued functions $\mathcal{H}$ on $\mathcal{Z}$ that is uniformly bounded by a constant $U > 0$ and for all $t > 0$, the following inequality holds*

$$\mathbb{P}\left(\left|\sup_{h \in \mathcal{H}}\left|\sum_{i=1}^{n} h(Z_i)\right| - \mathbb{E}\left[\sup_{h \in \mathcal{H}}\left|\sum_{i=1}^{n} h(Z_i)\right|\right]\right| \geq t\right) \leq C\exp\left(-\frac{t}{CU}\log\left(1 + \frac{Ut}{D}\right)\right),$$

*where $C$ is a universal constant and $D \geq \mathbb{E}\left[\sup_{h \in \mathcal{H}}\sum_{i=1}^{n} h^2(Z_i)\right]$.*

**Lemma 3** (Ledoux-Talagrand contraction inequality). *Let $Z_1, \ldots, Z_n$ be independent random variables in $\mathcal{Z}$. For any class of real-valued functions $\mathcal{H}$ on $\mathcal{Z}$ and any $L$-Lipschitz function $\varphi$, the following inequality holds*

$$\mathbb{E}\left[\sup_{h \in \mathcal{H}}\left|\sum_{i=1}^{n} \epsilon_i(\varphi \circ h)(Z_i)\right|\right] \leq 2L\mathbb{E}\left[\sup_{h \in \mathcal{H}}\left|\sum_{i=1}^{n} \epsilon_i h(Z_i)\right|\right],$$

*where $\epsilon_1, \ldots, \epsilon_n$ are independent Rademacher random variables.*

Lastly, we state an auxiliary inequality that serves as a typical candidate for the quantity denoted by $D$ above in Talagrand's inequality. This result follows as a corollary of the Ledoux-Talagrand contraction inequality and a symmetrization argument. We provide a proof for completeness.

**Lemma 4.** *Let $Z_1, \ldots, Z_n$ be independent random variables in $\mathcal{Z}$. For any class of real-valued functions $\mathcal{H}$ on $\mathcal{Z}$ and any $L$-Lipschitz function $\varphi$, the following inequality holds*

$$\mathbb{E}\left[\sup_{h \in \mathcal{H}}\sum_{i=1}^{n}(\varphi \circ h)(Z_i)\right] \leq \sup_{h \in \mathcal{H}}\sum_{i=1}^{n}\mathbb{E}\left[(\varphi \circ h)(Z_i)\right] + 4L\mathbb{E}\left[\sup_{h \in \mathcal{H}}\left|\sum_{i=1}^{n} \epsilon_i h(Z_i)\right|\right],$$

*where $\epsilon_1, \ldots, \epsilon_n$ are independent Rademacher random variables.*

*Proof.* We have that

$$\mathbb{E}\left[\sup_{h \in \mathcal{H}}\sum_{i=1}^{n}(\varphi \circ h)(Z_i)\right] - \sup_{h \in \mathcal{H}}\sum_{i=1}^{n}\mathbb{E}\left[(\varphi \circ h)(Z_i)\right] \tag{14}$$

$$= \mathbb{E}\left[\sup_{h \in \mathcal{H}}\sum_{i=1}^{n}(\varphi \circ h)(Z_i) - \sup_{h \in \mathcal{H}}\sum_{i=1}^{n}\mathbb{E}\left[(\varphi \circ h)(Z_i)\right]\right] \tag{15}$$

$$\leq \mathbb{E}\left[\sup_{h \in \mathcal{H}}\left|\sum_{i=1}^{n}(\varphi \circ h)(Z_i) - \sum_{i=1}^{n}\mathbb{E}\left[(\varphi \circ h)(Z_i)\right]\right|\right] \tag{16}$$

$$= \mathbb{E}\left[\sup_{h \in \mathcal{H}}\left|\sum_{i=1}^{n}\left((\varphi \circ h)(Z_i) - \mathbb{E}\left[(\varphi \circ h)(Z_i)\right]\right)\right|\right] \tag{17}$$

$$\leq 2\mathbb{E}\left[\sup_{h \in \mathcal{H}}\left|\sum_{i=1}^{n}\epsilon_i(\varphi \circ h)(Z_i)\right|\right] \tag{18}$$

$$\leq 4L\mathbb{E}\left[\sup_{h \in \mathcal{H}}\left|\sum_{i=1}^{n}\epsilon_i h(Z_i)\right|\right]. \tag{19}$$

Inequality equation 16 follows from the triangle inequality, inequality equation 18 follows from a standard symmetrization argument (see Koltchinskii (2011)), and inequality equation 19 follows

from the Ledoux-Talagrand contraction inequality (see Lemma 3). The result follows by moving the second term in Equation equation 14 to the right-hand side in the last inequality. $\qquad\square$

## B  COMPLEXITY AND HETEROGENEITY MEASURES

In this section, we introduce important quantities of policy class complexity and client heterogeneity that appear in our analysis. All throughout, we let $n = \sum_{c \in \mathcal{C}} n_c$ be the total sample size across clients, $n_{\mathcal{C}} = (n_c)_{c \in \mathcal{C}}$ the vector of sample sizes across clients, and $\bar{n} = (n_c/n)_{c \in \mathcal{C}}$ the empirical distribution over clients.

### B.1  POLICY CLASS COMPLEXITY

#### B.1.1  HAMMING DISTANCE & ENTROPY INTEGRAL

We provide additional details on the definition of the entropy integral introduces in Section 5.1.

**Definition 6** (Hamming distance, covering number, and entropy integral). Consider a policy class $\Pi$ and a multi-source covariate set $x = \{x_i^c \mid c \in \mathcal{C}, i \in [n_c]\} \subset \mathcal{X}$ across clients $\mathcal{C}$ with client sample sizes $n_{\mathcal{C}}$. We define the following:

(a) the Hamming distance between any two policies $\pi_1, \pi_2 \in \Pi$ given multi-source covariate set $x$ is

$$\mathrm{H}(\pi_1, \pi_2; x) := \frac{1}{\sum_{c \in \mathcal{C}} n_c} \sum_{c \in \mathcal{C}} \sum_{i=1}^{n_c} \mathbf{1}\{\pi_1(x_i^c) \neq \pi_2(x_i^c)\};$$

(b) an $\epsilon$-cover of $\Pi$ under the Hamming distance given covariate set $x$ is any policy set $S$ such that for any $\pi \in \Pi$ there exists some $\pi' \in S$ such that $\mathrm{H}(\pi, \pi'; x) \leq \epsilon$;

(c) the $\epsilon$-covering number of $\Pi$ under the Hamming distance given covariate set $x$ is
$$N_{\mathrm{H}}(\epsilon, \Pi; x) := \min\{|S| \mid S \in \mathcal{S}_{\mathrm{H}}(\epsilon, \Pi; x)\},$$
where $\mathcal{S}_{\mathrm{H}}(\epsilon, \Pi; x)$ is the set of all $\epsilon$-covers of $\Pi$ with respect to $\mathrm{H}(\cdot, \cdot; x)$;

(d) the $\epsilon$-covering number of $\Pi$ under the Hamming distance is
$$N_{\mathrm{H}}(\epsilon, \Pi) := \sup\{N_{\mathrm{H}}(\epsilon, \Pi; x) \mid x \in \mathcal{X}_{\mathcal{C}}\},$$
where $\mathcal{X}_{\mathcal{C}}$ is the set of all covariate sets in $\mathcal{X}$ across clients $\mathcal{C}$ with arbitrary sample sizes;

(e) the entropy integral of $\Pi$ is

$$\kappa(\Pi) := \int_0^1 \sqrt{\log N_{\mathrm{H}}(\epsilon^2, \Pi)} d\epsilon.$$

#### B.1.2  $\ell_{\lambda,2}$ DISTANCE

Consider the function class
$$\mathcal{F}_{\Pi} := \{Q(\cdot, \pi) : \Omega \to \mathbb{R} \mid \pi \in \Pi\},$$
where
$$Q(\omega_i^c; \pi) := \gamma_i^c(\pi(x_i^c))$$
for any covariate-score vector $\omega_i^c = (x_i^c, \gamma_i^c) \in \Omega = \mathcal{X} \times \mathbb{R}^d$ and $\pi \in \Pi$, where $\gamma_i^c(a)$ is the $a$-th coordinate of the score vector $\gamma_i^c$.

**Definition 7** ($\ell_{\lambda,2}$ distance and covering number). Consider a policy class $\Pi$, function class $\mathcal{F}_{\Pi}$, and a multi-source covariate-score set $\omega = \{\omega_i^c \mid c \in \mathcal{C}, i \in [n_c]\} \subset \Omega$ across clients $\mathcal{C}$ with client sample sizes $n_{\mathcal{C}}$ and client distribution $\lambda$. We define:

(a) the $\ell_{\lambda,2}$ distance with respect to function class $\mathcal{F}_{\Pi}$ between any two policies $\pi_1, \pi_2 \in \Pi$ given covariate-score set $\omega$ is

$$\ell_{\lambda,2}(\pi_1, \pi_2; \omega) = \sqrt{\frac{\sum_{c \in \mathcal{C}} \sum_{i=1}^{n_c} \frac{\lambda_c^2}{n_c^2}\big(Q(\omega_i^c; \pi_1) - Q(\omega_i^c; \pi_2)\big)^2}{\sup_{\pi_a, \pi_b \in \Pi} \sum_{c \in \mathcal{C}} \sum_{i=1}^{n_c} \frac{\lambda_c^2}{n_c^2}\big(Q(\omega_i^c; \pi_a) - Q(\omega_i^c; \pi_b)\big)^2}};$$

(b) an $\epsilon$-cover of $\Pi$ under the $\ell_{\lambda,2}$ distance given covariate-score set $\omega$ is any policy set $S$ such that for any $\pi \in \Pi$ there exists some $\pi' \in S$ such that $\ell_{\lambda,2}(\pi, \pi'; \omega) \leq \epsilon$;

(c) the $\epsilon$-covering number of $\Pi$ under the $\ell_{\lambda,2}$ distance given covariate-score set $\omega$ is

$$N_{\ell_{\lambda,2}}(\epsilon, \Pi; \omega) := \min\{|S| \mid S \text{ is an } \epsilon\text{-cover of } \Pi \text{ w.r.t. } \ell_{\lambda,2}(\cdot, \cdot; \omega)\}.$$

The following lemma relates the covering numbers of the two policy distances we have defined.

**Lemma 5.** *Let $\omega = \{\omega_i^c \mid c \in \mathcal{C}, i \in [n_c]\} \subset \Omega$ be a multi-source covariate-score set across clients $\mathcal{C}$ with client sample sizes $n_{\mathcal{C}}$ and client distribution $\lambda$. For any $\epsilon > 0$,*

$$N_{\ell_{\lambda,2}}(\epsilon, \Pi; \omega) \leq N_H(\epsilon^2, \Pi).$$

*Proof.* Fix $\epsilon > 0$. Without loss of generality, we assume $N_H(\epsilon^2, \Pi) < \infty$, otherwise the result trivially holds. Let $S_0 = \{\pi_1, \ldots, \pi_{N_0}\}$ be a corresponding Hamming $\epsilon^2$-cover of $\Pi$.

Consider any arbitrary $\pi \in \Pi$. By definition, there exists a $\pi' \in S_0$ such that for any multi-source covariate set $\tilde{x} = \{\tilde{x}_i^c \mid c \in \mathcal{C}, i \in [\tilde{n}_c]\}$ with any given sample sizes $\tilde{n}_c > 0$ the following holds:

$$H(\pi, \pi'; \tilde{x}) = \frac{1}{\tilde{n}} \sum_{c \in \mathcal{C}} \sum_{i=1}^{\tilde{n}_c} \mathbf{1}\{\pi(\tilde{x}_i^c) \neq \pi'(\tilde{x}_i^c)\} \leq \epsilon^2,$$

where $\tilde{n} = \sum_{c \in \mathcal{C}} \tilde{n}_c$. Using this pair of policies $\pi, \pi'$ we generate an augmented data set $\tilde{\omega}$ from $\omega$ as follows. Let $m$ be a positive integer and define $\tilde{\omega}$ to be a collection of multiple copies of all covariate-score tuples $\omega_i^c \in \omega$, where each $\omega_i^c$ appears

$$\tilde{n}_i^c := \left\lceil \frac{m \cdot \frac{\lambda_c^2}{n_c^2} \big(Q(\omega_i^c; \pi) - Q(\omega_i^c; \pi')\big)^2}{\sup_{\pi_a, \pi_b} \sum_{c \in \mathcal{C}} \sum_{i=1}^{n_c} \frac{\lambda_c^2}{n_c^2} \big(Q(\omega_i^c; \pi_a) - Q(\omega_i^c; \pi_b)\big)^2} \right\rceil$$

times in $\tilde{\omega}$. Therefore, the client sample sizes in this augmented data set are $\tilde{n}_c = \sum_{i=1}^{n_c} \tilde{n}_i^c$ and the total sample size is $\tilde{n} = \sum_{c \in \mathcal{C}} \sum_{i=1}^{n_c} \tilde{n}_i^c$. The total sample size is bounded as

$$\tilde{n} = \sum_{c \in \mathcal{C}} \sum_{i=1}^{n_c} \left\lceil \frac{m \cdot \frac{\lambda_c^2}{n_c^2} \big(Q(\omega_i^c; \pi) - Q(\omega_i^c; \pi')\big)^2}{\sup_{\pi_a, \pi_b} \sum_{c \in \mathcal{C}} \sum_{i=1}^{n_c} \frac{\lambda_c^2}{n_c^2} \big(Q(\omega_i^c; \pi_a) - Q(\omega_i^c; \pi_b)\big)^2} \right\rceil$$

$$\leq \sum_{c \in \mathcal{C}} \sum_{i=1}^{n_c} \left( \frac{m \cdot \frac{\lambda_c^2}{n_c^2} \big(Q(\omega_i^c; \pi) - Q(\omega_i^c; \pi')\big)^2}{\sup_{\pi_a, \pi_b} \sum_{c \in \mathcal{C}} \sum_{i=1}^{n_c} \frac{\lambda_c^2}{n_c^2} \big(Q(\omega_i^c; \pi_a) - Q(\omega_i^c; \pi_b)\big)^2} + 1 \right)$$

$$\leq \frac{m \cdot \sum_{c \in \mathcal{C}} \sum_{i=1}^{n_c} \frac{\lambda_c^2}{n_c^2} \big(Q(\omega_i^c; \pi) - Q(\omega_i^c; \pi')\big)^2}{\sup_{\pi_a, \pi_b} \sum_{c \in \mathcal{C}} \sum_{i=1}^{n_c} \frac{\lambda_c^2}{n_c^2} \big(Q(\omega_i^c; \pi_a) - Q(\omega_i^c; \pi_b)\big)^2} + n \leq m + n.$$

Then, we have

$$H(\pi, \pi'; \tilde{\omega}) = \frac{1}{\tilde{n}} \sum_{c \in \mathcal{C}} \sum_{i=1}^{\tilde{n}_c} \mathbf{1}\{\pi(x_i^c) \neq \pi'(x_i^c)\}$$

$$= \frac{1}{\tilde{n}} \sum_{c \in \mathcal{C}} \sum_{i=1}^{n_c} \tilde{n}_i^c \cdot \mathbf{1}\{\pi(x_i^c) \neq \pi'(x_i^c)\}$$

$$\geq \frac{1}{\tilde{n}} \sum_{c \in \mathcal{C}} \sum_{i=1}^{n_c} \frac{m \cdot \frac{\lambda_c^2}{n_c^2} \big(Q(\omega_i^c; \pi) - Q(\omega_i^c; \pi')\big)^2}{\sup_{\pi_a, \pi_b} \sum_{c \in \mathcal{C}} \sum_{i=1}^{n_c} \frac{\lambda_c^2}{n_c^2} \big(Q(\omega_i^c; \pi_a) - Q(\omega_i^c; \pi_b)\big)^2} \mathbf{1}\{\pi(x_i^c) \neq \pi'(x_i^c)\}$$

$$= \frac{m}{\tilde{n}} \sum_{c \in \mathcal{C}} \sum_{i=1}^{n_c} \frac{\frac{\lambda_c^2}{n_c^2} \big(Q(\omega_i^c; \pi) - Q(\omega_i^c; \pi')\big)^2}{\sup_{\pi_a, \pi_b} \sum_{c \in \mathcal{C}} \sum_{i=1}^{n_c} \frac{\lambda_c^2}{n_c^2} \big(Q(\omega_i^c; \pi_a) - Q(\omega_i^c; \pi_b)\big)^2}$$

$$\geq \frac{m}{m + n} \ell_{\lambda,2}^2(\pi, \pi'; \omega).$$

This implies that

$$\ell_{\lambda,2}(\pi, \pi'; \omega) \leq \sqrt{\frac{m+n}{m} \mathrm{H}(\pi, \pi'; \tilde{\omega})} \leq \sqrt{1 + \frac{n}{m}} \cdot \epsilon.$$

Letting $m \to \infty$ yields $\ell_{\lambda,2}(\pi, \pi'; \omega) \leq \epsilon$. This establishes that for any $\pi \in \Pi$, there exists a $\pi' \in S_0$ such that $\ell_{\lambda,2}(\pi, \pi'; \omega) \leq \epsilon$, and thus $N_{\ell_{\lambda,2}}(\epsilon, \Pi; \omega) \leq N_{\mathrm{H}}(\epsilon^2, \Pi)$. $\qquad\square$

### B.1.3 Weighted Rademacher complexity

Our learning bounds will rely on the following notion of *weighted Rademacher complexity* introduced in Mohri et al. (2019).

**Definition 8** (Weighted Rademacher complexity). Suppose there is a set of clients $\mathcal{C}$, with each client $c \in \mathcal{C}$ having a data-generating distribution $\mathcal{P}_c$ defined over a space $\Omega$. Moreover, the clients have fixed sample sizes $n_{\mathcal{C}} = (n_c)_{c \in \mathcal{C}}$ and there is a distribution $\lambda$ over the set of clients $\mathcal{C}$. For each client $c \in \mathcal{C}$, let $W_1^c, \ldots, W_{n_c}^c$ be independent random variables sampled from $\mathcal{P}_c$, and let $W = \{W_i^c \mid c \in \mathcal{C}, i \in [n_c]\}$ represent the collection of samples across all clients.

The *empirical weighted Rademacher complexity* of a function class $\mathcal{F}$ on $\Omega$ given multi-source data $W$ under mixture weights $\lambda$ and sample sizes $n_{\mathcal{C}}$ is

$$\mathfrak{R}_{\lambda, n_{\mathcal{C}}}(\mathcal{F}; W) := \mathbb{E}\left[\sup_{f \in \mathcal{F}} \left| \sum_{c \in \mathcal{C}} \frac{\lambda_c}{n_c} \sum_{i=1}^{n_c} \varepsilon_i^c f(W_i^c) \right| \;\middle|\; W\right],$$

where the expectation is taken with respect to the collection of independent Rademacher random variables $\epsilon = \{\varepsilon_i^c \mid c \in \mathcal{C}, i \in [n_c]\}$. Additionally, the *weighted Rademacher complexity* of $\mathcal{F}$ under mixture weights $\lambda$ and sample sizes $n_{\mathcal{C}}$ is

$$\mathfrak{R}_{\lambda, n_{\mathcal{C}}}(\mathcal{F}) := \mathbb{E}\left[\sup_{f \in \mathcal{F}} \left| \sum_{c \in \mathcal{C}} \frac{\lambda_c}{n_c} \sum_{i=1}^{n_c} \varepsilon_i^c f(W_i^c) \right|\right],$$

where the expectation is taken with respect to the multi-source random variables $W$ and the independent Rademacher random variables $\epsilon$.

## B.2 Client Heterogeneity

### B.2.1 Client Distribution Skewness

An important quantity that arises in our analysis is

$$\sum_{c \in \mathcal{C}} \frac{\lambda_c^2}{\bar{n}_c} = \mathbb{E}_{c \sim \lambda}\left[\frac{\lambda_c}{\bar{n}_c}\right],$$

which captures a measure of the imbalance of the client distribution $\lambda$ relative to the empirical client distribution $\bar{n}$. The following result makes this interpretation more clear:

$$\begin{aligned}
\sum_{c \in \mathcal{C}} \frac{\lambda_c^2}{\bar{n}_c} &= \sum_{c \in \mathcal{C}} \frac{\lambda_c^2}{\bar{n}_c} + \sum_{c \in \mathcal{C}} \bar{n}_c - 2 \sum_{c \in \mathcal{C}} \lambda_c + 1 \\
&= \sum_{c \in \mathcal{C}} \left( \frac{\lambda_c^2}{\bar{n}_c} + \frac{\bar{n}_c^2}{\bar{n}_c} - \frac{2\lambda_c \bar{n}_c}{\bar{n}_c} \right) + 1 \\
&= \sum_{c \in \mathcal{C}} \frac{(\lambda_c - \bar{n}_c)^2}{\bar{n}_c} + 1 \\
&= \chi^2(\lambda \| \bar{n}) + 1.
\end{aligned}$$

where $\chi^2(\lambda \| \bar{n})$ is the chi-squared divergence from $\bar{n}$ to $\lambda$. Following Mohri et al. (2019), we call this quantity the skewness.

**Definition 9** (Skewness). The *skewness* of a given distribution $\lambda$ over clients is

$$\mathfrak{s}(\lambda \| \bar{n}) := 1 + \chi^2(\lambda \| \bar{n}),$$

where $\chi^2(\lambda \| \bar{n})$ is the chi-squared divergence of $\lambda$ from $\bar{n}$.

### B.2.2 CLIENT DISTRIBUTION SHIFT

In Section 3.3, we defined $\bar{\mathcal{D}}_c$ to be the complete local data-generating distribution of client $c \in \mathcal{C}$ and $\bar{\mathcal{D}}_\lambda = \sum_{c \in \mathcal{C}} \lambda_c \bar{\mathcal{D}}_c$ to the complete global data-generating distribution as a mixture of all other complete client data-generating distributions. As we will observe in the local regret bounds, client heterogeneity will be captured by the distribution shift of the local distributions to the global distribution. In particular, the distribution shift of a local distribution $\bar{\mathcal{D}}_c$ from the global distribution $\bar{\mathcal{D}}_\lambda$ will be captured by their total variation distance $\mathrm{TV}(\bar{\mathcal{D}}_c, \bar{\mathcal{D}}_\lambda)$ and also their KL divergence $\mathrm{KL}(\bar{\mathcal{D}}_c || \bar{\mathcal{D}}_\lambda)$. We will also introduce an alternate bound that will capture the distribution shift with their chi-squared divergence $\chi^2(\bar{\mathcal{D}}_c || \bar{\mathcal{D}}_\lambda)$. See Appendix D for these results.

## C BOUNDING GLOBAL REGRET

### C.1 PRELIMINARIES

#### C.1.1 FUNCTION CLASSES

As mentioned in Appendix B.1.2, the function class we will be considering in our analysis is

$$\mathcal{F}_\Pi := \{Q(\cdot; \pi) : \Omega \to \mathbb{R} \mid \pi \in \Pi\}, \tag{20}$$

where

$$Q(\omega_i^c; \pi) := \gamma_i^c(\pi(x_i^c)), \tag{21}$$

for any covariate-score vector $\omega_i^c = (x_i^c, \gamma_i^c) \in \Omega = \mathcal{X} \times \mathbb{R}^d$ and $\pi \in \Pi$, where $\gamma_i^c(a)$ is the $a$-th coordinate of the score vector $\gamma_i^c$. It will also be useful to consider the Minkowski difference of $\mathcal{F}_\Pi$ with itself,

$$\Delta\mathcal{F}_\Pi := \mathcal{F}_\Pi - \mathcal{F}_\Pi = \{\Delta(\cdot; \pi_a, \pi_b) : \Omega \to \mathbb{R} \mid \pi_a, \pi_b \in \Pi\}, \tag{22}$$

where

$$\Delta(\omega_i^c; \pi_a, \pi_b) := Q(\omega_i^c; \pi_a) - Q(\omega_i^c; \pi_b) = \gamma_i^c(\pi_a(x_i^c)) - \gamma_i^c(\pi_b(x_i^c)), \tag{23}$$

for any $\omega_i^c = (x_i^c, \gamma_i^c) \in \Omega$ and $\pi_a, \pi_b \in \Pi$.

#### C.1.2 POLICY VALUE ESTIMATORS

**Augmented Inverse Propensity Weighted Scores**   We use propensity-weighted scores to estimate policy values. For any $c \in \mathcal{C}$, consider the available observable samples $(X_i^c, A_i^c, Y_i^c)$ taken from the partially observable counterfactual sample $Z_i^c = (X_i^c, A_i^c, Y_i^c(a_1), \ldots, Y_i^c(a_d)) \sim \bar{\mathcal{D}}_c$, for $i \in [n_c]$. As discussed in Section 4.2, using this data, we considered construction of the oracle local AIPW scores

$$\Gamma_i^c(a) = \mu_c(X_i^c; a) + \big(Y_i^c(A_i^c) - \mu_c(X_i^c; a)\big) w_c(X_i^c; a) \mathbf{1}\{A_i^c = a\}$$

for each $a \in \mathcal{A}$. Similarly, we discussed the construction of the approximate local AIPW scores

$$\hat{\Gamma}_i^c(a) = \hat{\mu}_c(X_i^c; a) + \big(Y_i^c(A_i^c) - \hat{\mu}_c(X_i^c; a)\big) \hat{w}(X_i^c; a) \mathbf{1}\{A_i^c = a\}$$

for each $a \in \mathcal{A}$, given fixed estimates $\hat{\mu}_c$ and $\hat{w}_c$ of $\mu_c$ and $w_c$, respectively. In practice, we use cross-fitting to make the estimates fixed and independent relative to the data on which they are evaluated. Note that only this second set of scores can be constructed from the observed data. The first set is "constructed" for analytic purposes in our proofs.

**Policy Value Estimates and Policy Value Difference Estimates**   Using the local data and the constructed AIPW scores, we let $W_i^c = (X_i^c, \Gamma_i^c(a_1), \ldots, \Gamma_i^c(a_d))$ and $\hat{W}_i^c = (X_i^c, \hat{\Gamma}_i^c(a_1), \ldots, \hat{\Gamma}_i^c(a_d))$ for each $i \in [n_c]$. We define the oracle and approximate *policy value* estimates of $Q_\lambda(\pi)$, respectively, as

$$\tilde{Q}_\lambda(\pi) = \sum_{c \in \mathcal{C}} \frac{\lambda_c}{n_c} \sum_{i=1}^{n_c} \Gamma_i^c(\pi(X_i^c)) = \sum_{c \in \mathcal{C}} \frac{\lambda_c}{n_c} \sum_{i=1}^{n_c} Q(W_i^c; \pi),$$

$$\hat{Q}_\lambda(\pi) = \sum_{c \in \mathcal{C}} \frac{\lambda_c}{n_c} \sum_{i=1}^{n_c} \hat{\Gamma}_i^c(\pi(X_i^c)) = \sum_{c \in \mathcal{C}} \frac{\lambda_c}{n_c} \sum_{i=1}^{n_c} Q(\hat{W}_i^c; \pi),$$

for any $\pi \in \Pi$, where we use the function class defined in Equation 21 for the alternate representations that we will use throughout our proofs for notational convenience. It will also be very useful to define the following corresponding *policy value difference* quantities:

$$\Delta_\lambda(\pi_a, \pi_b) := Q_\lambda(\pi_a) - Q_\lambda(\pi_b),$$

$$\tilde{\Delta}_\lambda(\pi_a, \pi_b) := \tilde{Q}_\lambda(\pi_a) - \tilde{Q}_\lambda(\pi_b) = \sum_{c \in \mathcal{C}} \frac{\lambda_c}{n_c} \sum_{i=1}^{n_c} \Delta(W_i^c; \pi_a, \pi_b),$$

$$\hat{\Delta}_\lambda(\pi_a, \pi_b) := \hat{Q}_\lambda(\pi_a) - \hat{Q}_\lambda(\pi_b) = \sum_{c \in \mathcal{C}} \frac{\lambda_c}{n_c} \sum_{i=1}^{n_c} \Delta(\hat{W}_i^c; \pi_a, \pi_b),$$

for any $\pi_a, \pi_b \in \Pi$, where we use the function class defined in Equation equation 23 for the alternate representations that we will use throughout our proofs for notational convenience.

**Unbiased Estimates**    The following result can be used to readily show that the oracle estimators for the local and global policy values are unbiased.

**Lemma 6.** *Suppose Assumption 1 holds. For any $\pi \in \Pi$,*

$$\mathbb{E}_{Z^c \sim \bar{\mathcal{D}}_c} [\Gamma^c(\pi(X^c))] = Q_c(\pi)$$

*and*

$$\mathbb{E}_{c \sim \lambda} \mathbb{E}_{Z^c \sim \bar{\mathcal{D}}_c} [\Gamma^c(\pi(X^c))] = Q_\lambda(\pi),$$

*where $Z^c = (X^c, A^c, Y^c(a_1), \ldots, Y^c(a_d)) \sim \bar{\mathcal{D}}_c$.*

*Proof.* First, observe that for any $a \in \mathcal{A}$,

$$\Gamma^c(a) = \mu_c(X^c; a) + (Y^c(A^c) - \mu_c(X^c; a)) \, w_c(X^c; a) \mathbf{1}\{A^c = a\}$$
$$= \mu_c(X^c; a) + (Y^c(a) - \mu_c(X^c; a)) \, w_c(X^c; a) \mathbf{1}\{A^c = a\}$$

due to the indicator in the definition, and so for any $(X^c, A^c, Y^c(a_1), \ldots, Y^c(a_d)) \sim \bar{\mathcal{D}}_c$,

$$\mathbb{E}_{A^c, \vec{Y}^c} [\Gamma^c(a) \mid X^c] = \mu_c(X^c; a) + \mathbb{E}_{A^c, \vec{Y}^c} [(Y^c(a) - \mu_c(X^c; a)) \, w_c(X^c; a) \mathbf{1}\{A^c = a\} \mid X^c]$$
$$= \mu_c(X^c; a) + \mathbb{E}_{\vec{Y}^c} [Y^c(a) - \mu_c(X^c; a) \mid X^c] \cdot \mathbb{E}_{A^c} [w_c(X^c; a) \mathbf{1}\{A^c = a\} \mid X^c]$$
$$= \mu_c(X^c; a) + \mathbb{E}_{\vec{Y}^c} [Y^c(a) - \mu_c(X^c; a) \mid X^c] \cdot w_c(X^c; a) e_c(X^c; a)$$
$$= \mu_c(X^c; a) + \mathbb{E}_{\vec{Y}^c} [Y^c(a) \mid X^c] - \mu_c(X^c; a)$$
$$= \mathbb{E}_{\vec{Y}^c} [Y^c(a) \mid X^c].$$

The second equality follows from the unconfoundedness assumption stated in Assumption 1. This immediately implies that

$$\mathbb{E}_{Z^c \sim \bar{\mathcal{D}}_c} [\Gamma^c(\pi(X^c))] = \mathbb{E}_{Z^c \sim \bar{\mathcal{D}}_c} \left[ \sum_{a \in \mathcal{A}} \mathbf{1}\{\pi(X^c) = a\} \Gamma^c(a) \right]$$

$$= \mathbb{E}_{X^c} \left[ \sum_{a \in \mathcal{A}} \mathbf{1}\{\pi(X^c) = a\} \mathbb{E}_{A^c, \vec{Y}^c} [\Gamma^c(a) \mid X^c] \right]$$

$$= \mathbb{E}_{X^c} \left[ \sum_{a \in \mathcal{A}} \mathbf{1}\{\pi(X^c) = a\} \mathbb{E}_{A^c, \vec{Y}^c} [Y^c(a) \mid X^c] \right]$$

$$= \mathbb{E}_{X^c} \left[ \mathbb{E}_{A^c, \vec{Y}^c} [Y^c(\pi(X^c)) \mid X^c] \right]$$

$$= \mathbb{E}_{Z^c \sim \bar{\mathcal{D}}_c} [Y^c(\pi(X^c))]$$

$$= Q_c(\pi),$$

and

$$\mathbb{E}_{c \sim \lambda} \mathbb{E}_{Z^c \sim \bar{\mathcal{D}}_c} [\Gamma^c(\pi(X^c))] = \mathbb{E}_{c \sim \lambda} [Q_c(\pi)] = Q_\lambda(\pi).$$

$\square$

### C.1.3 DATA-GENERATING DISTRIBUTIONS AND SUFFICIENT STATISTICS

As introduced in the problem setting in Section 3.1, each client has a data-generating distribution $\mathcal{D}_c$ defined over the joint space $\mathcal{X} \times \mathcal{Y}^d$ of contexts and potential outcomes. Moreover, the historical policy $e_c : \mathcal{X} \to \Delta(\mathcal{A})$ induces the complete data-generating distribution $\bar{\mathcal{D}}_c$ defined over the joint space $\mathcal{X} \times \mathcal{A} \times \mathcal{Y}^d$ of contexts, actions, and potential outcomes such that sampling $(X^c, A^c, Y^c(a_1), \ldots, Y^c(a_d)) \sim \bar{\mathcal{D}}_c$ is defined as sampling $(X^c, Y^c(a_1), \ldots, Y^c(a_d)) \sim \mathcal{D}_c$ and $A^c \sim e_c(\cdot | X^c)$.

Note that, by construction, the contexts and AIPW scores are sufficient statistics for the corresponding oracle and approximate estimators of the policy values. Moreover, our results will mostly depend on properties of the sufficient statistics (e.g., AIPW score range and variance). Therefore, it will be useful for notational simplicity in our analysis to define the distribution of the sufficient statistics. For any $Z^c = (X^c, A^c, Y^c(a_1), \ldots, Y^c(a_d)) \sim \bar{\mathcal{D}}_c$, let

$$W^c = (X^c, \Gamma^c(a_1), \ldots, \Gamma^c(a_d))$$

be the sufficient statistic of contexts and oracle AIPW scores, and we denote its induced distribution as $\tilde{\mathcal{D}}_c$ defined over $\Omega = \mathcal{X} \times \mathbb{R}^d$.

*Remark.* For simplicity, without loss of generality, when proving results that only involve the contexts and AIPW scores, we will assume the data is sampled from the distributions of the sufficient statistics, e.g., $W^c \sim \tilde{\mathcal{D}}_c$. When we have a discussion involving constructing the AIPW scores from the observable data, we will be more careful about the source distributions and typically assume the data is sampled from the complete data-generating distributions, e.g., $Z^c \sim \bar{\mathcal{D}}_c$.

### C.1.4 PROOF SKETCH

We describe our general proof strategy. The standard approach for proving finite-sample regret bounds in offline policy learning is to establish uniform concentration bounds around a proper notion of empirical complexity, which is then further bounded by class-dependent vanishing rates, as seen in (Athey & Wager, 2021; Zhou et al., 2023). Typically, this complexity notion involves the Rademacher complexity of an appropriate policy value-based function class. However, this is not applicable to our scenario where the data may not come from the same source distribution. In our proof, we draw inspiration from the work on empirical risk bounds in multiple-source supervised learning settings, particularly (Mohri et al., 2019), to identify the suitable notion of complexity—namely, the weighted Rademacher complexity of the policy value function class $\mathcal{F}_\Pi$. While Mohri et al. (2019) provided a starting framework for a multiple-source analysis in supervised learning, the proof techniques for establishing class-dependent uniform concentration results in offline policy learning are typically more involved than those for empirical risk bounds in supervised learning. Our bounds necessitate more complex Dudley-type chaining arguments with applications of Talagrand's inequalities with some multiple-source modifications mediated by skewness.

To begin, we split the global regret in terms of an oracle regret term of policy value differences and an approximation error term. Let $\pi_\lambda^* = \arg \max_{\pi \in \Pi} Q_\lambda(\pi)$. We can decompose the regret incurred by the global policy $\hat{\pi}_\lambda = \arg \max_{\pi \in \Pi} \hat{Q}_\lambda(\pi)$ as follows:

$$
\begin{aligned}
R_\lambda(\hat{\pi}_\lambda) &= Q_\lambda(\pi_\lambda^*) - Q_\lambda(\hat{\pi}_\lambda) \\
&= \left( Q_\lambda(\pi_\lambda^*) - Q_\lambda(\hat{\pi}_\lambda) \right) - \left( \hat{Q}_\lambda(\pi_\lambda^*) - \hat{Q}_\lambda(\hat{\pi}_\lambda) \right) + \left( \hat{Q}_\lambda(\pi_\lambda^*) - \hat{Q}_\lambda(\hat{\pi}_\lambda) \right) \\
&= \Delta_\lambda(\pi_\lambda^*, \hat{\pi}_\lambda) - \hat{\Delta}_\lambda(\pi_\lambda^*, \hat{\pi}_\lambda) + \left( \hat{Q}_\lambda(\pi_\lambda^*) - \hat{Q}_\lambda(\hat{\pi}_\lambda) \right) \\
&\leq \Delta_\lambda(\pi_\lambda^*, \hat{\pi}_\lambda) - \hat{\Delta}_\lambda(\pi_\lambda^*, \hat{\pi}_\lambda) \\
&\leq \sup_{\pi_a, \pi_b \in \Pi} |\Delta_\lambda(\pi_a, \pi_b) - \hat{\Delta}_\lambda(\pi_a, \pi_b)| \\
&\leq \sup_{\pi_a, \pi_b \in \Pi} |\Delta_\lambda(\pi_a, \pi_b) - \tilde{\Delta}_\lambda(\pi_a, \pi_b)| + \sup_{\pi_a, \pi_b \in \Pi} |\tilde{\Delta}_\lambda(\pi_a, \pi_b) - \hat{\Delta}_\lambda(\pi_a, \pi_b)|.
\end{aligned}
$$

The first inequality holds by definition of the global policy, the second inequality by a straightforward worst-case supremum bound, and the last inequality by the triangle inequality.

The first oracle term in the last inequality will be bounded by the weighted Rademacher complexity of $\Delta \mathcal{F}_\Pi$. Then, using a Dudley chaining argument, the weighted Rademacher complexity will be bounded by a measure of policy class complexity and vanishing rates with respect to the total sample

size. We emphasize that we were able to establish rates with respect to the total sample size, rather than some other more moderate quantity of the sample sizes, such as the average or the minimum. The second term in the last inequality will be bounded by a decomposition of the approximation terms which will be shown to be asymptotically vanishing faster than the rate bounds of the oracle regret with high probability. Establishing these bounds requires the use of Assumption 2 to ensure there is enough data across clients. Altogether, these results will provide a rate bound on the global regret that scales with the total sample size and is mediated by client skewness.

In a later section, we establish bounds for the notion of local regret, unique to our problem setting. This insight arises from recognizing the mismatch between global server-level performance and local client-level performance. We derive a local regret bound dependent on measures of distribution shift between clients, providing valuable insights into the value of information in heterogeneous client participation and how exactly heterogeneity affects policy performance for any given client. This exact quantification is highlighted in our Theorem 2 that decomposes the sources of heterogeneity at the population, environment, and treatment level. We also point to Theorem 3 for an alternative local regret bound that does not require bounded inverse propensity weighted scores.

### C.2 BOUNDING WEIGHTED RADEMACHER COMPLEXITY

First, to simplify our analysis, we can easily bound the weighted Rademacher complexity of $\Delta \mathcal{F}_\Pi$ by that of $\mathcal{F}_\Pi$ as follows.

**Lemma 7.**

$$\mathfrak{R}_{\lambda, n_\mathcal{C}}(\Delta \mathcal{F}_\Pi) \leq 2\mathfrak{R}_{\lambda, n_\mathcal{C}}(\mathcal{F}_\Pi)$$

*Proof.*

$$
\begin{aligned}
\mathfrak{R}_{\lambda, n_\mathcal{C}}(\Delta \mathcal{F}_\Pi) &= \mathbb{E}\left[\sup_{\pi_a, \pi_b \in \Pi} \left| \sum_{c \in \mathcal{C}} \frac{\lambda_c}{n_c} \sum_{i=1}^{n_c} \varepsilon_i^c \Delta(W_i^c; \pi_a, \pi_b) \right| \right] \\
&= \mathbb{E}\left[\sup_{\pi_a, \pi_b \in \Pi} \left| \sum_{c \in \mathcal{C}} \frac{\lambda_c}{n_c} \sum_{i=1}^{n_c} \varepsilon_i^c \Big( Q(W_i^c; \pi_a) - Q(W_i^c; \pi_b) \Big) \right| \right] \\
&\leq \mathbb{E}\left[\sup_{\pi_a, \pi_b \in \Pi} \left| \sum_{c \in \mathcal{C}} \frac{\lambda_c}{n_c} \sum_{i=1}^{n_c} \varepsilon_i^c Q(W_i^c; \pi_a) \right| + \left| \sum_{c \in \mathcal{C}} \frac{\lambda_c}{n_c} \sum_{i=1}^{n_c} \varepsilon_i^c Q(W_i^c; \pi_b) \right| \right] \\
&= 2\mathbb{E}\left[\sup_{\pi \in \Pi} \left| \sum_{c \in \mathcal{C}} \frac{\lambda_c}{n_c} \sum_{i=1}^{n_c} \varepsilon_i^c Q(W_i^c; \pi) \right| \right] \\
&= 2\mathfrak{R}_{\lambda, n_\mathcal{C}}(\mathcal{F}_\Pi).
\end{aligned}
$$

$\square$

Therefore, we can simply focus on bounding the weighted Rademacher complexity of $\mathcal{F}_\Pi$.

**Proposition 1.** *Suppose Assumptions 1 and 2 hold. Then,*

$$\mathfrak{R}_{\lambda, n_\mathcal{C}}(\mathcal{F}_\Pi) \leq (14 + 6\kappa(\Pi)) \sqrt{\frac{V_{\lambda, n_\mathcal{C}}}{n}} + o\left( \sqrt{\frac{\mathfrak{s}(\lambda \| \bar{n})}{n}} \right),$$

*where*

$$V_{\lambda, n_\mathcal{C}} = \sup_{\pi_a, \pi_b \in \Pi} \sum_{c \in \mathcal{C}} \frac{\lambda_c^2}{\bar{n}_c} \mathop{\mathbb{E}}_{W^c \sim \tilde{\mathcal{D}}_c} \left[ \Delta^2(W^c; \pi_a, \pi_b) \right].$$

*Proof.* We follow a chaining argument to bound the weighted Rademacher complexity of $\mathcal{F}_\Pi$.

**Constructing the policy approximation chain.** First, for each client $c \in \mathcal{C}$, let $W_1^c, \ldots, W_{n_c}^c$ be $n_c$ independent random variables sampled from $\tilde{\mathcal{D}}_c$, where each $W_i^c = (X_i^c, \vec{\Gamma}_i^c) \in \Omega = \mathcal{X} \times \mathbb{R}^d$. Additionally, let $W = \{W_i^c \mid c \in \mathcal{C}, i \in [n_c]\}$ represent the corresponding collection of samples across all clients.

Next, set $K = \lceil \log_2 n \rceil$. We will construct a sequence $\{\Psi_k : \Pi \to \Pi\}_{k=0}^K$ of policy approximation operators that satisfies the following properties. For any $k = 0, \ldots, K$,

(P1) $\max_{\pi \in \Pi} \ell_{\lambda,2}(\Psi_{k+1}(\pi), \Psi_k(\pi); Z) \leq \epsilon_k := 2^{-k}$

(P2) $|\{\Psi_k(\pi) \mid \pi \in \Pi\}| \leq N_{\ell_{\lambda,2}}(\epsilon_k, \Pi; Z)$

We use the notational shorthand that $\Psi_{K+1}(\pi) = \pi$ for any $\pi \in \Pi$. We will construct the policy approximation chain via a backward recursion scheme. First, let $\Pi_k$ denote the smallest $\epsilon_k$-covering set of $\Pi$ under the $\ell_{\lambda,2}$ distance given data $Z$. Note, in particular, that $|\Pi_0| = 1$ since the $\ell_{\lambda,2}$ distance is never more than 1 and so any single policy is enough to 1-cover all policies in $\Pi$. Then, the backward recursion is as follows: for any $\pi \in \Pi$,

1. define $\Psi_K(\pi) = \arg\min_{\pi' \in \Pi_K} \ell_{\lambda,2}(\pi, \pi'; W)$;

2. for each $k = K - 1, \ldots, 1$, define $\Psi_k(\pi) = \arg\min_{\pi' \in \Pi_k} \ell_{\lambda,2}(\Psi_{k+1}(\pi), \pi'; W)$;

3. define $\Psi_0(\pi) \equiv 0$.

Note that although $\Psi_0(\pi)$ is not in $\Pi$, it can still serve as a 1-cover of $\Pi$ since the $\ell_{\lambda,2}$ distance is always bounded by 1. Before proceeding, we check that each of the stated desired properties of the constructed operator chain is satisfied:

(P1) Pick any $\pi \in \Pi$. Clearly, $\Psi_{k+1}(\pi) \in \Pi$. Then, by construction of $\Pi_k$, there exists a $\pi' \in \Pi_k$ such that $\ell_{\lambda,2}(\Psi_{k+1}(\pi), \pi'; W) \leq \epsilon_k$. Therefore, by construction of $\Psi_k(\pi)$, we have $\ell_{\lambda,2}(\Psi_{k+1}(\pi), \Psi_k(\pi); W) \leq \ell_{\lambda,2}(\Psi_{k+1}(\pi), \pi'; W) \leq \epsilon_k$.

(P2) By construction of $\Psi_k$, we have that $\Psi_k(\pi) \in \Pi_k$ for every $\pi \in \Pi$. Therefore, $|\{\Psi_k(\pi) \mid \pi \in \Pi\}| \leq |\Pi_k| = N_{\ell_{\lambda,2}}(\epsilon_k, \Pi; W)$.

Thus, the constructed chain satisfies the desired properties. Next, we observe that since $\Psi_0(\pi) \equiv 0$, we have that $Q(W_i^c; \Psi_0(\pi)) = 0$ and

$$Q(W_i^c; \pi) = Q(W_i^c; \pi) - Q(W_i^c; \Psi_0(\pi))$$

$$= Q(W_i^c; \pi) - Q(W_i^c; \Psi_K(\pi)) + \sum_{k=1}^K Q(W_i^c; \Psi_k(\pi) - Q(W_i^c; \Psi_{k-1}(\pi))$$

$$= \Delta(W_i^c; \pi, \Psi_K(\pi)) + \sum_{k=1}^K \Delta(W_i^c; \Psi_k(\pi), \Psi_{k-1}(\pi))$$

Therefore, we can decompose the weighted Rademacher complexity of $\mathcal{F}_\Pi$ as follows:

$$\mathfrak{R}_{\lambda,n_c}(\mathcal{F}_\Pi) = \mathbb{E}\left[\sup_{\pi \in \Pi}\left|\sum_{c \in \mathcal{C}} \frac{\lambda_c}{n_c} \sum_{i=1}^{n_c} \varepsilon_i^c \Delta(W_i^c; \pi, \Psi_K(\pi))\right|\right]$$

$$+ \mathbb{E}\left[\sup_{\pi \in \Pi}\left|\sum_{c \in \mathcal{C}} \frac{\lambda_c}{n_c} \sum_{i=1}^{n_c} \varepsilon_i^c \left(\sum_{k=1}^K \Delta(W_i^c; \Psi_k(\pi), \Psi_{k-1}(\pi))\right)\right|\right]$$

We will obtain bounds separately for these two terms, which we refer to as the *negligible regime* term and the *effective regime* term, respectively.

**Bounding the negligible regime.** For convenience, we denote

$$B_{\lambda,n_c}(W) := \sup_{\pi_a, \pi_b \in \Pi} \sum_{c \in \mathcal{C}} \sum_{i=1}^{n_c} \frac{\lambda_c^2}{n_c^2} \Delta^2(W_i^c; \pi_a, \pi_b)$$

and $B_{\lambda,n_c} := \mathbb{E}\left[B_{\lambda,n_c}(W)\right]$.

Given any realization of independent Rademacher random variables $\epsilon = \{\varepsilon_i^c \mid c \in \mathcal{C}, i \in [n_c]\}$ and multi-source data $W$, by the Cauchy-Schwarz inequality,

$$\left| \sum_{c \in \mathcal{C}} \frac{\lambda_c}{n_c} \sum_{i=1}^{n_c} \varepsilon_i^c \Delta(W_i^c; \pi, \Psi_K(\pi)) \right|$$

$$\leq \sqrt{\sum_{c \in \mathcal{C}} \sum_{i=1}^{n_c} (\varepsilon_i^c)^2} \cdot \sqrt{\sum_{c \in \mathcal{C}} \sum_{i=1}^{n_c} \frac{\lambda_c^2}{n_c^2} \Delta^2(W_i^c; \pi, \Psi_K(\pi))}$$

$$= \sqrt{n} \cdot \sqrt{B_{\lambda, n_{\mathcal{C}}}(W)} \ell_2(\pi, \Psi_K(\pi); Z)$$

$$\leq \sqrt{n B_{\lambda, n_{\mathcal{C}}}(W)} \epsilon_k$$

$$\leq \sqrt{\frac{B_{\lambda, n_{\mathcal{C}}}(W)}{n}}.$$

Then, by Jensen's inequality,

$$\mathbb{E}\left[ \sup_{\pi \in \Pi} \left| \sum_{c \in \mathcal{C}} \frac{\lambda_c}{n_c} \sum_{i=1}^{n_c} \varepsilon_i^c \Delta(W_i^c; \pi, \Psi_K(\pi)) \right| \right] \leq \mathbb{E}\left[ \sqrt{\frac{B_{\lambda, n_{\mathcal{C}}}(W)}{n}} \right] \leq \sqrt{\frac{B_{\lambda, n_{\mathcal{C}}}}{n}}.$$

**Bounding the effective regime.**    For any $k \in [K]$, let

$$t_{k,\delta} = \sqrt{B_{\lambda, n_{\mathcal{C}}}(W)} \epsilon_k \tau_{k,\delta}$$

where $\tau_{k,\delta} > 0$ is some constant to be specified later. By Hoeffding's inequality (in Lemma 1),

$$\mathbb{P}\left( \left| \sum_{c \in \mathcal{C}} \frac{\lambda_c}{n_c} \sum_{i=1}^{n_c} \varepsilon_i^c \Delta(W_i^c; \Psi_k(\pi), \Psi_{k-1}(\pi)) \right| > t_{k,\delta} \,\Big|\, W \right)$$

$$\leq 2 \exp\left( -\frac{t_{k,\delta}^2}{2 \sum_{c \in \mathcal{C}} \sum_{i=1}^{n_c} \frac{\lambda_c^2}{n_c^2} \Delta^2(W_i^c; \Psi_k(\pi), \Psi_{k-1}(\pi))} \right)$$

$$= 2 \exp\left( -\frac{t_{k,\delta}^2}{2 B_{\lambda, n_{\mathcal{C}}}(W) \ell_2^2(\Psi_k(\pi), \Psi_{k-1}(\pi); W)} \right)$$

$$\leq 2 \exp\left( -\frac{t_{k,\delta}^2}{2 B_{\lambda, n_{\mathcal{C}}}(W) \epsilon_{k-1}^2} \right)$$

$$= 2 \exp\left( -\frac{t_{k,\delta}^2}{8 B_{\lambda, n_{\mathcal{C}}}(W) \epsilon_k^2} \right)$$

$$= 2 \exp\left( -\frac{\tau_{k,\delta}^2}{8} \right).$$

Here, we used the fact that $\epsilon_{k-1} = 2\epsilon_k$. Setting

$$\tau_{k,\delta} = \sqrt{8 \log\left( \frac{\pi^2 k^2}{3\delta} N_{\ell_{\lambda,2}}(\epsilon_k, \Pi; W) \right)}$$

and applying a union bound over the policy space, we obtain

$$
\mathbb{P}\left( \sup_{\pi \in \Pi} \left| \sum_{c \in \mathcal{C}} \frac{\lambda_c}{n_c} \sum_{i=1}^{n_c} \varepsilon_i^c \Delta(W_i^c; \Psi_k(\pi), \Psi_{k-1}(\pi)) \right| > t_{k,\delta} \,\Big|\, W \right)
$$

$$
\leq 2\,|\Pi_k| \cdot \exp\left( -\frac{\tau_{k,\delta}^2}{8} \right)
$$

$$
\leq 2 N_{\ell_2}(\epsilon_k, \Pi; W) \cdot \exp\left( -\frac{\tau_{k,\delta}^2}{8} \right)
$$

$$
= \frac{6\delta}{\pi^2 k^2}.
$$

By a further union bound over $k \in [K]$, we obtain

$$
\mathbb{P}\left( \sup_{\pi \in \Pi} \left| \sum_{c \in \mathcal{C}} \frac{\lambda_c}{n_c} \sum_{i=1}^{n_c} \varepsilon_i^c \left( \sum_{k=1}^{K} \Delta(W_i^c; \Psi_k(\pi), \Psi_{k-1}(\pi)) \right) \right| > \sum_{k=1}^{K} t_{k,\delta} \,\Big|\, W \right)
$$

$$
\leq \sum_{k=1}^{K} \mathbb{P}\left( \sup_{\pi \in \Pi} \left| \sum_{c \in \mathcal{C}} \frac{\lambda_c}{n_c} \sum_{i=1}^{n_c} \varepsilon_i^c \Delta(W_i^c; \Psi_k(\pi), \Psi_{k-1}(\pi)) \right| > t_{k,\delta} \,\Big|\, W \right)
$$

$$
\leq \sum_{k=1}^{K} \frac{6\delta}{\pi^2 k^2} \leq \delta.
$$

Therefore, given multi-source data $W$, with probability at least $1 - \delta$, we have

$$
\sup_{\pi \in \Pi} \left| \sum_{c \in \mathcal{C}} \frac{\lambda_c}{n_c} \sum_{i=1}^{n_c} \varepsilon_i^c \left( \sum_{k=1}^{K} \Delta(W_i^c; \Psi_k(\pi), \Psi_{k-1}(\pi)) \right) \right|
$$

$$
\leq \sum_{k=1}^{K} t_{k,\delta}
$$

$$
= \sqrt{B_{\lambda,n_\mathcal{C}}(W)} \sum_{k=1}^{K} \epsilon_k \sqrt{8 \log\left( \frac{\pi^2 k^2}{3\delta} N_{\ell_2}(\epsilon_k, \Pi; W) \right)}
$$

$$
= \sqrt{B_{\lambda,n_\mathcal{C}}(W)} \sum_{k=1}^{K} \epsilon_k \left( \sqrt{8 \log \frac{\pi^2}{3\delta} + 16 \log k + 8 \log N_{\ell_2}(\epsilon_k, \Pi; W)} \right)
$$

$$
\leq \sqrt{B_{\lambda,n_\mathcal{C}}(W)} \sum_{k=1}^{K} \epsilon_k \left( \sqrt{8 \log \frac{\pi^2}{3\delta}} + \sqrt{16 \log k} + \sqrt{8 \log N_{\ell_2}(\epsilon_k, \Pi; W)} \right)
$$

$$
\leq \sqrt{B_{\lambda,n_\mathcal{C}}(W)} \sum_{k=1}^{K} \epsilon_k \left( \sqrt{8 \log(4/\delta)} + \sqrt{16 \log k} + \sqrt{8 \log N_{\mathrm{H}}(\epsilon_k, \Pi)} \right)
$$

$$
\leq \sqrt{B_{\lambda,n_\mathcal{C}}(W)} \left( \sqrt{8 \log(4/\delta)} + 2 + \sqrt{8} \sum_{k=1}^{\infty} \epsilon_k \sqrt{\log N_{\mathrm{H}}(\epsilon_k, \Pi)} \right)
$$

$$
\leq \sqrt{B_{\lambda,n_\mathcal{C}}(W)} \left( \sqrt{8 \log(4/\delta)} + 2 + \sqrt{8}\kappa(\Pi) \right).
$$

Next, we turn this high-probability bound into a bound on the conditional expectation. First, let $F_R(\cdot \mid W)$ be the cumulative distribution of the random variable

$$
R := \sup_{\pi \in \Pi} \left| \sum_{c \in \mathcal{C}} \frac{\lambda_c}{n_c} \sum_{i=1}^{n_c} \varepsilon_i^c \left( \sum_{k=1}^{K} \Delta(W_i^c; \Psi_k(\pi), \Psi_{k-1}(\pi)) \right) \right|
$$

conditional on $W$. Above, we have shown that

$$
1 - F_R\left( \sqrt{B_{\lambda,n_\mathcal{C}}(W)} \left( \sqrt{8 \log(4/\delta)} + 2 + \sqrt{8}\kappa(\Pi) \right) \mid W \right) \leq \delta.
$$

For any non-negative integer $l$, let $\Delta_l = \sqrt{B_{\lambda,n_c}(W)}(\sqrt{8\log(4/\delta_l)}+2+\sqrt{8}\kappa(\Pi))$ where $\delta_l = 2^{-l}$. Since $R$ is non-negative, we can compute and upper bound the conditional expectation of $R$ given $W$ as follows:

$$\mathbb{E}\left[\sup_{\pi\in\Pi}\left|\sum_{c\in\mathcal{C}}\frac{\lambda_c}{n_c}\sum_{i=1}^{n_c}\varepsilon_i^c\left(\sum_{k=1}^{K}\Delta(W_i^c;\Psi_k(\pi),\Psi_{k-1}(\pi))\right)\right|\ \middle|\ W\right]$$

$$= \int_0^\infty (1 - F_R(r|W))\,dr$$

$$\leq \sum_{l=0}^\infty (1 - F_R(\Delta_l|W))\,\Delta_l$$

$$\leq \sum_{l=0}^\infty \delta_l\Delta_l$$

$$= \sum_{l=0}^\infty 2^{-l}\cdot\sqrt{B_{\lambda,n_c}(W)}\left(\sqrt{8(l+2)\log 2}+2+\sqrt{8}\kappa(\Pi)\right)$$

$$\leq \sqrt{B_{\lambda,n_c}(W)}\left(4\sqrt{8\log 2}+4+2\sqrt{8}\kappa(\Pi)\right)$$

$$\leq \sqrt{B_{\lambda,n_c}(W)}\left(14+6\kappa(\Pi)\right).$$

Taking the expectation with respect to $W$ and using Jensen's inequality, we obtain

$$\mathbb{E}\left[\sup_{\pi\in\Pi}\left|\sum_{c\in\mathcal{C}}\frac{\lambda_c}{n_c}\sum_{i=1}^{n_c}\varepsilon_i^c\left(\sum_{k=1}^{K}\Delta(W_i^c;\Psi_k(\pi),\Psi_{k-1}(\pi))\right)\right|\right]$$

$$\leq (14+6\kappa(\Pi))\,\mathbb{E}\left[\sqrt{B_{\lambda,n_c}(W)}\right]$$

$$\leq (14+6\kappa(\Pi))\,\sqrt{B_{\lambda,n_c}}.$$

**Refining the upper bound.** One could easily bound $B_{\lambda,n_c}$ using worst-case bounds on the AIPW element. Instead, we use Lemma 4 to get a more refined bound on $B_{\lambda,n_c}$.

To use this result, we identify the set of independent random variables $\tilde{W}_i^c = T(W_i^c) = (X_i^c, \frac{\lambda_c}{n_c}\Gamma_i^c)$ for $c \in \mathcal{C}$ and $i \in [n_c]$ and the function class $\mathcal{H} = \{\Delta(\cdot;\pi_a,\pi_b) \mid \pi_a,\pi_b \in \Pi\}$. We also identify the Lipschitz function $\varphi : u \mapsto u^2$ defined over the set $\mathcal{U} \subset \mathbb{R}$ containing all possible outputs of any $\Delta(\cdot;\pi_a,\pi_b)$ given any realization of $\tilde{W}_i^c$ for any $c \in \mathcal{C}$ as input. To further capture this domain, note that by the boundedness and overlap assumptions in Assumption 1, it is easy to verify that there exists some $U > 0$ for all $c \in \mathcal{C}$ such that $|\Gamma_i^c(a)| \leq U$ for any $a \in \mathcal{A}$ and any realization of $W_i^c = (X_i^c, \Gamma_i^c)$. This implies that

$$|\Delta(\tilde{W}_i^c;\pi_a,\pi_b)| = \frac{\lambda_c}{n_c}\left|\Gamma_i^c(\pi_a(X_i^c)) - \Gamma_i^c(\pi_b(X_i^c))\right| \leq 2U\frac{\lambda_c}{n_c},$$

for any realization of $\tilde{W}_i^c$ and any $\pi_a,\pi_b \in \Pi$. Moreover, note that

$$\frac{\lambda_c}{n_c} \leq \max_{c\in\mathcal{C}}\frac{\lambda_c}{n_c} \leq \sqrt{\sum_{c\in\mathcal{C}}\frac{\lambda_c^2}{n_c^2}} \leq \frac{1}{\sqrt{\min_{c\in\mathcal{C}}n_c}}\sqrt{\sum_{c\in\mathcal{C}}\frac{\lambda_c^2}{n_c}} = \frac{1}{\sqrt{\min_{c\in\mathcal{C}}n_c}}\sqrt{\frac{\mathfrak{s}(\lambda\|\bar{n})}{n}} =: s_{\lambda,n_c} \quad (24)$$

Therefore, $\mathcal{U} \subset [-s_{\lambda,n_c}, s_{\lambda,n_c}]$, and thus, for any $u,v \in \mathcal{U}$, we have that

$$|\varphi(u) - \varphi(v)| = |u^2 - v^2| = |u+v|\cdot|u-v| \leq 4Us_{\lambda,n_c}|u-v|.$$

Therefore, $L = 4Us_{\lambda,n_c}$ is a valid Lipschitz constant for $\varphi$. Then, through these identifications, Lemma 4 guarantees the following upper bound

$$
\begin{aligned}
B_{\lambda,n_c} &= \mathbb{E}\left[\sup_{\pi_a,\pi_b\in\Pi}\sum_{c\in\mathcal{C}}\sum_{i=1}^{n_c}\frac{\lambda_c^2}{n_c^2}\Delta^2(W_i^c;\pi_a,\pi_b)\right] \\
&= \mathbb{E}\left[\sup_{\pi_a,\pi_b\in\Pi}\sum_{c\in\mathcal{C}}\sum_{i=1}^{n_c}\varphi\circ\Delta(\tilde{W}_i^c;\pi_a,\pi_b)\right] \\
&\leq \sup_{\pi_a,\pi_b\in\Pi}\sum_{c\in\mathcal{C}}\sum_{i=1}^{n_c}\mathbb{E}\left[\varphi\circ\Delta(\tilde{W}_i^c;\pi_a,\pi_b)\right] + 16Us_{\lambda,n_c}\mathbb{E}\left[\sup_{\pi_a,\pi_b\in\Pi}\left|\sum_{c\in\mathcal{C}}\sum_{i=1}^{n_c}\varepsilon_i^c\Delta(\tilde{W}_i^c;\pi_a,\pi_b)\right|\right] \\
&= \sup_{\pi_a,\pi_b\in\Pi}\sum_{c\in\mathcal{C}}\sum_{i=1}^{n_c}\frac{\lambda_c^2}{n_c^2}\mathbb{E}\left[\Delta^2(W_i^c;\pi_a,\pi_b)\right] + 16Us_{\lambda,n_c}\mathbb{E}\left[\sup_{\pi_a,\pi_b\in\Pi}\left|\sum_{c\in\mathcal{C}}\frac{\lambda_c}{n_c}\sum_{i=1}^{n_c}\varepsilon_i^c\Delta(W_i^c;\pi_a,\pi_b)\right|\right] \\
&= \sup_{\pi_a,\pi_b\in\Pi}\sum_{c\in\mathcal{C}}\frac{\lambda_c^2}{n_c}\mathbb{E}\left[\Delta^2(W_i^c;\pi_a,\pi_b)\right] + 16U\Re_{\lambda,n_c}(\Delta\mathcal{F}_\Pi)s_{\lambda,n_c} \\
&\leq \sup_{\pi_a,\pi_b\in\Pi}\sum_{c\in\mathcal{C}}\frac{\lambda_c^2}{n_c}\mathbb{E}\left[\Delta^2(W_i^c;\pi_a,\pi_b)\right] + 32U\Re_{\lambda,n_c}(\mathcal{F}_\Pi)s_{\lambda,n_c} \\
&= \frac{V_{\lambda,n_c}}{n} + 32U\Re_{\lambda,n_c}(\mathcal{F}_\Pi)s_{\lambda,n_c}.
\end{aligned}
$$

Before proceeding, note that by the local data size scaling assumption stated in Assumption 2, $n_c = \Omega(\nu_c(n))$ for some increasing function $\nu_c$ for any $c \in \mathcal{C}$. This immediately implies that $s_{\lambda,n_c}$ is dominated as

$$
s_{\lambda,n_c} = \frac{1}{\sqrt{\min_{c\in\mathcal{C}}n_c}}\sqrt{\frac{\mathfrak{s}(\lambda\|\bar{n})}{n}} \leq o\left(\sqrt{\frac{\mathfrak{s}(\lambda\|\bar{n})}{n}}\right).
$$

**Combine results.** Thus, combining the bounds for the negligible and effective regime and including the refined bound, we have

$$
\begin{aligned}
&\Re_{\lambda,n_c}(\mathcal{F}_\Pi) \\
&\leq \sqrt{\frac{B_{\lambda,n_c}}{n}} + (14 + 6\kappa(\Pi))\sqrt{B_{\lambda,n_c}} \\
&\leq \sqrt{\frac{V_{\lambda,n_c}}{n^2} + 32U\Re_{\lambda,n_c}(\mathcal{F}_\Pi)\frac{s_{\lambda,n_c}}{n}} + (14 + 6\kappa(\Pi))\sqrt{\frac{V_{\lambda,n_c}}{n} + 32U\Re_{\lambda,n_c}(\mathcal{F}_\Pi)s_{\lambda,n_c}} \\
&\leq \sqrt{\frac{V_{\lambda,n_c}}{n^2}} + \sqrt{32U\Re_{\lambda,n_c}(\mathcal{F}_\Pi)\frac{s_{\lambda,n_c}}{n}} + (14 + 6\kappa(\Pi))\left(\sqrt{\frac{V_{\lambda,n_c}}{n}} + \sqrt{32U\Re_{\lambda,n_c}(\mathcal{F}_\Pi)s_{\lambda,n_c}}\right) \\
&\leq (14 + 6\kappa(\Pi))\sqrt{\frac{V_{\lambda,n_c}}{n}} + \sqrt{\frac{V_{\lambda,n_c}}{n^2}} + \mathcal{O}\left(\sqrt{\Re_{\lambda,n_c}(\mathcal{F}_\Pi)s_{\lambda,n_c}}\right)
\end{aligned}
\tag{25}
$$

This gives an upper bound on $\Re_{\lambda,n_c}(\mathcal{F}_\Pi)$ in terms of itself. To decouple this dependence, we express

$$
\begin{aligned}
\Re_{\lambda,n_c}(\mathcal{F}_\Pi) &\leq \mathcal{O}\left(\sqrt{\frac{V_{\lambda,n_c}}{n}}\right) + \mathcal{O}\left(\sqrt{\Re_{\lambda,n_c}(\mathcal{F}_\Pi)s_{\lambda,n_c}}\right) \\
&\leq A_1\sqrt{\frac{V_{\lambda,n_c}}{n}} + A_2\sqrt{\Re_{\lambda,n_c}(\mathcal{F}_\Pi)s_{\lambda,n_c}}
\end{aligned}
\tag{26}
$$

for some constants $A_1, A_2$, and we split this inequality into the following two exhaustive cases.

_Case 1_: $A_2\sqrt{s_{\lambda,n_c}} \leq \frac{1}{2}\sqrt{\Re_{\lambda,n_c}(\mathcal{F}_\Pi)}$

In this case, we can bound the second term in the right-hand side of inequality equation 26 to get

$$\mathfrak{R}_{\lambda,n_{\mathcal{C}}}(\mathcal{F}_\Pi) \le A_1 \sqrt{\frac{V_{\lambda,n_{\mathcal{C}}}}{n}} + \frac{\mathfrak{R}_{\lambda,n_{\mathcal{C}}}(\mathcal{F}_\Pi)}{2},$$

and so

$$\mathfrak{R}_{\lambda,n_{\mathcal{C}}}(\mathcal{F}_\Pi) \le 2A_1 \sqrt{\frac{V_{\lambda,n_{\mathcal{C}}}}{n}}. \tag{27}$$

Moreover,

$$V_{\lambda,n_{\mathcal{C}}} = \sup_{\pi_a,\pi_b \in \Pi} \sum_{c \in \mathcal{C}} \frac{\lambda_c^2}{\bar{n}_c} \mathop{\mathbb{E}}_{W^c \sim \tilde{\mathcal{D}}_c} [\Delta^2(W^c; \pi_a, \pi_b)]$$

$$\le \sup_{\pi_a,\pi_b \in \Pi} \max_{c \in \mathcal{C}} \mathop{\mathbb{E}}_{W^c \sim \mathcal{P}_c} [\Delta^2(W^c; \pi_a, \pi_b)] \cdot \sum_{c \in \mathcal{C}} \frac{\lambda_c^2}{\bar{n}_c} = \bar{V}\mathfrak{s}(\lambda \| \bar{n}),$$

where $\bar{V} = \sup_{\pi_a,\pi_b \in \Pi} \max_{c \in \mathcal{C}} \mathbb{E}_{W^c \sim \mathcal{P}_c} [\Delta(W^c; \pi_a, \pi_b)]$, which is a constant value. Note that the last equality holds by the skewness identity in established in Appendix B.2.1. Plugging this into inequality equation 27, we get

$$\mathfrak{R}_{\lambda,n_{\mathcal{C}}}(\mathcal{F}_\Pi) \le 2A_1 \sqrt{\frac{\bar{V}\mathfrak{s}(\lambda \| \bar{n})}{n}} \le \mathcal{O}\left( \sqrt{\frac{\mathfrak{s}(\lambda \| \bar{n})}{n}} \right).$$

*Case 2*: $A_2 \sqrt{s_{\lambda,n_{\mathcal{C}}}} > \frac{1}{2}\sqrt{\mathfrak{R}_{\lambda,n_{\mathcal{C}}}(\mathcal{F}_\Pi)}$

In this case, one can easily rearrange terms to get that

$$\mathfrak{R}_{\lambda,n_{\mathcal{C}}}(\mathcal{F}_\Pi) < 4A_2^2 s_{\lambda,n_{\mathcal{C}}} \le o\left( \sqrt{\frac{\mathfrak{s}(\lambda \| \bar{n})}{n}} \right).$$

Therefore, in either case, $\mathfrak{R}_{\lambda,n_{\mathcal{C}}}(\mathcal{F}_\Pi) \le \mathcal{O}\left(\sqrt{\frac{\mathfrak{s}(\lambda \| \bar{n})}{n}}\right)$. We can plug this asymptotic bound into inequality equation 25 to arrive at the desired result,

$$\mathfrak{R}_{\lambda,n_{\mathcal{C}}}(\mathcal{F}_\Pi) \le (14 + 6\kappa(\Pi)) \sqrt{\frac{V_{\lambda,n_{\mathcal{C}}}}{n}} + \sqrt{\frac{V_{\lambda,n_{\mathcal{C}}}}{n^2}} + \mathcal{O}\left( \sqrt{\mathcal{O}\left( \sqrt{\frac{\mathfrak{s}(\lambda \| \bar{n})}{n}} \right) s_{\lambda,n_{\mathcal{C}}}} \right)$$

$$\le (14 + 6\kappa(\Pi)) \sqrt{\frac{V_{\lambda,n_{\mathcal{C}}}}{n}} + \sqrt{\frac{\bar{V}\mathfrak{s}(\lambda \| \bar{n})}{n^2}} + \mathcal{O}\left( \sqrt{\mathcal{O}\left( \sqrt{\frac{\mathfrak{s}(\lambda \| \bar{n})}{n}} \right) o\left( \sqrt{\frac{\mathfrak{s}(\lambda \| \bar{n})}{n}} \right)} \right)$$

$$\le (14 + 6\kappa(\Pi)) \sqrt{\frac{V_{\lambda,n_{\mathcal{C}}}}{n}} + o\left( \sqrt{\frac{\mathfrak{s}(\lambda \| \bar{n})}{n}} \right) + o\left( \sqrt{\frac{\mathfrak{s}(\lambda \| \bar{n})}{n}} \right)$$

$$\le (14 + 6\kappa(\Pi)) \sqrt{\frac{V_{\lambda,n_{\mathcal{C}}}}{n}} + o\left( \sqrt{\frac{\mathfrak{s}(\lambda \| \bar{n})}{n}} \right).$$

$$\square$$

## C.3 BOUNDING ORACLE REGRET

**Proposition 2.** *Suppose Assumptions 1 and 2 hold. Then, with probability at least $1 - \delta$,*

$$\sup_{\pi_a,\pi_b \in \Pi} |\Delta_\lambda(\pi_a, \pi_b) - \tilde{\Delta}_\lambda(\pi_a, \pi_b)| \le \left( c_1 \kappa(\Pi) + \sqrt{c_2 \log(c_2/\delta)} \right) \sqrt{\frac{V_{\lambda,n_{\mathcal{C}}}}{n}} + o\left( \sqrt{\frac{\mathfrak{s}(\lambda \| \bar{n})}{n}} \right),$$

*where $c_1$ and $c_2$ are universal constants.*

*Proof.* First, for each client $c \in \mathcal{C}$, let $W_1^c, \dots, W_{n_c}^c$ be $n_c$ independent random variables sampled from $\tilde{\mathcal{D}}_c$, where each $W_i^c = (X_i^c, \vec{\Gamma}_i^c) \in \Omega = \mathcal{X} \times \mathbb{R}^d$. Additionally, let $W = \{W_i^c \mid c \in \mathcal{C}, i \in [n_c]\}$ represent the corresponding collection of samples across all clients.

In Lemma 6, we showed that $\mathbb{E}_{W^c \sim \tilde{\mathcal{D}}_c}[Q(W^c; \pi)] = Q_c(\pi)$. This implies that

$$\mathbb{E}_W[\tilde{Q}_\lambda(\pi)] = \sum_{c \in \mathcal{C}} \frac{\lambda_c}{n_c} \sum_{i=1}^{n_c} \mathbb{E}[Q(W_i^c; \pi)] = \sum_{c \in \mathcal{C}} \frac{\lambda_c}{n_c} \sum_{i=1}^{n_c} Q_c(\pi) = \sum_{c \in \mathcal{C}} \lambda_c Q_c(\pi) = Q_\lambda(\pi).$$

Additionally,

$$\mathbb{E}_W[\tilde{\Delta}_\lambda(\pi_a, \pi_b)] = \mathbb{E}_W[\tilde{Q}_\lambda(\pi_a)] - \mathbb{E}_W[\tilde{Q}_\lambda(\pi_b)] = Q_\lambda(\pi_a) - Q_\lambda(\pi_b) = \Delta_\lambda(\pi_a, \pi_b).$$

Therefore, we can follow a symmetrization argument to upper bound the expected oracle regret in terms of a Rademacher complexity, namely the weighted Rademacher complexity. Let $W'$ be an independent copy of $W$ and let $\epsilon = \{\varepsilon_i^c \mid c \in \mathcal{C}, i \in [n_c]\}$ be a set of independent Rademacher random variables. Then,

$$\mathbb{E}\left[\sup_{\pi_a, \pi_b \in \Pi} |\Delta_\lambda(\pi_a, \pi_b) - \tilde{\Delta}_\lambda(\pi_a, \pi_b)|\right]$$

$$= \mathbb{E}_W\left[\sup_{\pi_a, \pi_b \in \Pi} \left|\mathbb{E}_{W'}\left[\sum_{c \in \mathcal{C}} \frac{\lambda_c}{n_c} \sum_{i=1}^{n_c} \Delta(W_i^{c\prime}; \pi_a, \pi_b)\right] - \sum_{c \in \mathcal{C}} \frac{\lambda_c}{n_c} \sum_{i=1}^{n_c} \Delta(W_i^c; \pi_a, \pi_b)\right|\right]$$

$$= \mathbb{E}_W\left[\sup_{\pi_a, \pi_b \in \Pi} \left|\mathbb{E}_{W'}\left[\sum_{c \in \mathcal{C}} \frac{\lambda_c}{n_c} \sum_{i=1}^{n_c} \Delta(W_i^{c\prime}; \pi_a, \pi_b) - \sum_{c \in \mathcal{C}} \frac{\lambda_c}{n_c} \sum_{i=1}^{n_c} \Delta(W_i^c; \pi_a, \pi_b)\right]\right|\right]$$

$$\leq \mathbb{E}_W\left[\mathbb{E}_{W'}\left[\sup_{\pi_a, \pi_b \in \Pi} \left|\sum_{c \in \mathcal{C}} \frac{\lambda_c}{n_c} \sum_{i=1}^{n_c} \left(\Delta(W_i^{c\prime}; \pi_a, \pi_b) - \Delta(W_i^c; \pi_a, \pi_b)\right)\right|\right]\right]$$

$$= \mathbb{E}_{W, W', \epsilon}\left[\sup_{\pi_a, \pi_b \in \Pi} \left|\sum_{c \in \mathcal{C}} \frac{\lambda_c}{n_c} \sum_{i=1}^{n_c} \varepsilon_i^c \left(\Delta(W_i^{c\prime}; \pi_a, \pi_b) - \Delta(W_i^c; \pi_a, \pi_b)\right)\right|\right]$$

$$\leq 2\mathbb{E}_{W, \epsilon}\left[\sup_{\pi_a, \pi_b \in \Pi} \left|\sum_{c \in \mathcal{C}} \frac{\lambda_c}{n_c} \sum_{i=1}^{n_c} \varepsilon_i^c \Delta(W_i^c; \pi_a, \pi_b)\right|\right]$$

$$= 2\mathfrak{R}_{\lambda, \bar{n}}(\Delta\mathcal{F}_\Pi)$$

$$\leq 4\mathfrak{R}_{\lambda, \bar{n}}(\mathcal{F}_\Pi).$$

The first equalities and inequalities follow from standard symmetrization arguments, and the last inequality follows from Lemma 7. Next, we use this bound on the expectation of the oracle regret and Talagrand's inequality (Lemma 2), to establish a high-probability bound on the oracle regret. In particular, we identify the set of independent random variables $\tilde{W} = \{\tilde{W}_i^c = (X_i^c, \frac{\lambda_c}{n_c}\Gamma_i^c) \mid c \in \mathcal{C}, i \in [n_c]\}$ and the function class $\mathcal{H} = \{h(\cdot; \pi_a, \pi_b) \mid \pi_a, \pi_b \in \Pi\}$ where

$$h(\tilde{W}_i^c; \pi_a, \pi_b) = \mathbb{E}[\Delta(\tilde{W}_i^c; \pi_a, \pi_b)] - \Delta(\tilde{W}_i^c; \pi_a, \pi_b), \tag{28}$$

which is uniformly bounded for any $c \in \mathcal{C}$ and $i \in [n_c]$ by

$$|h(\tilde{W}_i^c; \pi_a, \pi_b)| = \frac{\lambda_c}{n_c}\left|\mathbb{E}\left[\Gamma_i^c(\pi_a(X_i^c)) - \Gamma_i^c(\pi_b(X_i^c))\right] - \left(\Gamma_i^c(\pi_a(X_i^c)) - \Gamma_i^c(\pi_b(X_i^c))\right)\right|$$

$$\leq \frac{\lambda_c}{n_c} 4U$$

$$\leq 4U s_{\lambda, n_c} =: U_{\lambda, n_c},$$

where $U > 0$ is a uniform upper bound on $|\Gamma_i^c(a)|$ for any $c \in \mathcal{C}$ and $a \in \mathcal{A}$ guaranteed by Assumption 1, and where the last inequality follows from Inequality equation 24, Additionally, we have

$$s_{\lambda, n_c} = \frac{1}{\sqrt{\min_{c \in \mathcal{C}} n_c}} \sqrt{\frac{\mathfrak{s}(\lambda \| \bar{n})}{n}} \leq o\left(\sqrt{\frac{\mathfrak{s}(\lambda \| \bar{n})}{n}}\right),$$

as discussed in the proof of Proposition 1. Lastly, to use Talagrand's inequality, we set the constant $D$ (specified in Lemma 2) to be

$$D = \sup_{\pi_a, \pi_b \in \Pi} \sum_{c \in \mathcal{C}} \sum_{i=1}^{n_c} \mathbb{E}\big[h^2(\tilde{W}_i^c; \pi_a, \pi_b)\big] + 8 U_{\lambda, n_c} \mathbb{E}\left[\sup_{\pi_a, \pi_b \in \Pi} \left|\sum_{c \in \mathcal{C}} \sum_{i=1}^{n_c} \varepsilon_i^c h(\tilde{W}_i^c; \pi_a, \pi_b)\right|\right].$$

By Lemma 4, this choice of $D$ meets the required condition to use in Talagrand's inequality. In particular, we identify $\varphi : u \mapsto u^2$ defined over the set $\mathcal{U}$ containing all possible outputs of any function in $\mathcal{H}$ given any realization of $\tilde{W}_i^c$ for any $c \in \mathcal{C}$ as input. The uniform bound established above on realizable outputs of $h$ given input $\tilde{W}_i^c$ implies that $\mathcal{U} \subset [-U_{\lambda, n_c}, U_{\lambda, n_c}]$, and therefore, the Lipschitz constant of $\varphi$ is $L = 2 U_{\lambda, n_c}$, as required.

Next, after setting $t$ to be the positive solution of

$$\frac{t^2}{CD + CU_{\lambda, n_c} t} = \log(C/\delta),$$

Talagrand's inequality guarantees

$$\mathbb{P}\left(\left|\sup_{\pi_a, \pi_b \in \Pi} \left|\Delta_\lambda(\pi_a, \pi_b) - \tilde{\Delta}_\lambda(\pi_a, \pi_b)\right| - \mathbb{E}\left[\sup_{\pi_a, \pi_b \in \Pi} \left|\Delta_\lambda(\pi_a, \pi_b) - \tilde{\Delta}_\lambda(\pi_a, \pi_b)\right|\right]\right| \geq t\right)$$

$$= \mathbb{P}\left(\left|\sup_{\pi \in \Pi} \left|\sum_{c \in \mathcal{C}} \sum_{i=1}^{n_c} h(\tilde{W}_i^c; \pi_a, \pi_b)\right| - \mathbb{E}\left[\sup_{\pi \in \Pi} \left|\sum_{c \in \mathcal{C}} \sum_{i=1}^{n_c} h(\tilde{W}_i^c; \pi_a, \pi_b)\right|\right]\right| \geq t\right)$$

$$\leq C \exp\left(-\frac{t}{CU_{\lambda, n_c}} \log\left(1 + \frac{U_{\lambda, n_c} t}{D}\right)\right)$$

$$\leq C \exp\left(-\frac{t^2}{CD + CU_{\lambda, n_c} t}\right) = \delta.$$

Here, we used the inequality $\log(1 + x) \geq \frac{x}{1+x}$ for any $x \geq 0$. Observe that, by construction,

$$t = \frac{1}{2} CU_{\lambda, n_c} \log(C/\delta) + \sqrt{\frac{1}{4} C^2 U_{\lambda, n_c}^2 \log^2(C/\delta) + CD \log(C/\delta)}$$

$$\leq CU_{\lambda, n_c} \log(C/\delta) + \sqrt{CD \log(C/\delta)}$$

and

$$
\begin{aligned}
D &= \sup_{\pi_a,\pi_b \in \Pi} \sum_{c \in \mathcal{C}} \sum_{i=1}^{n_c} \mathbb{E}\left[h^2(\tilde{W}_i^c; \pi_a, \pi_b)\right] + 8U_{\lambda,\bar{n}}\mathbb{E}\left[\sup_{\pi_a,\pi_b \in \Pi}\left|\sum_{c \in \mathcal{C}} \sum_{i=1}^{n_c} \varepsilon_i^c h(\tilde{W}_i^c; \pi_a, \pi_b)\right|\right] \\
&= \sup_{\pi_a,\pi_b \in \Pi} \sum_{c \in \mathcal{C}} \sum_{i=1}^{n_c} \frac{\lambda_c^2}{n_c^2}\mathbb{E}\left[\left(\mathbb{E}\left[\Delta(W_i^c; \pi_a, \pi_b)\right] - \Delta(W_i^c; \pi_a, \pi_b)\right)^2\right] \\
&\quad + 8U_{\lambda,n_{\mathcal{C}}}\mathbb{E}\left[\sup_{\pi_a,\pi_b \in \Pi}\left|\sum_{c \in \mathcal{C}} \sum_{i=1}^{n_c} \frac{\lambda_c}{n_c}\varepsilon_i^c\left(\mathbb{E}\left[\Delta(W_i^c; \pi_a, \pi_b)\right] - \Delta(W_i^c; \pi_a, \pi_b)\right)\right|\right] \\
&= \sup_{\pi_a,\pi_b \in \Pi} \sum_{c \in \mathcal{C}} \sum_{i=1}^{n_c} \frac{\lambda_c^2}{n_c^2}\left(\mathbb{E}\left[\Delta^2(W_i^c; \pi_a, \pi_b)\right] - \mathbb{E}\left[\Delta(W_i^c; \pi_a, \pi_b)\right]^2\right) \\
&\quad + 8U_{\lambda,n_{\mathcal{C}}}\mathbb{E}\left[\sup_{\pi_a,\pi_b \in \Pi}\left|\sum_{c \in \mathcal{C}} \sum_{i=1}^{n_c} \frac{\lambda_c}{n_c}\varepsilon_i^c\left(\mathbb{E}\left[\Delta(W_i^c; \pi_a, \pi_b)\right] - \Delta(W_i^c; \pi_a, \pi_b)\right)\right|\right] \\
&\leq \sup_{\pi_a,\pi_b \in \Pi} \sum_{c \in \mathcal{C}} \sum_{i=1}^{n_c} \frac{\lambda_c^2}{n_c^2}\mathbb{E}\left[\Delta^2(W_i^c; \pi_a, \pi_b)\right] + 16U_{\lambda,n_{\mathcal{C}}}\mathbb{E}\left[\sup_{\pi_a,\pi_b \in \Pi}\left|\sum_{c \in \mathcal{C}} \sum_{i=1}^{n_c} \frac{\lambda_c}{n_c}\varepsilon_i^c\Delta(W_i^c; \pi_a, \pi_b)\right|\right] \\
&\leq \sup_{\pi_a,\pi_b \in \Pi} \sum_{c \in \mathcal{C}} \frac{\lambda_c^2}{n_c}\mathbb{E}\left[\Delta^2(W_i^c; \pi_a, \pi_b)\right] + 16U_{\lambda,n_{\mathcal{C}}}\mathfrak{R}_{\lambda,n_{\mathcal{C}}}(\Delta\mathcal{F}_\Pi) \\
&\leq \sup_{\pi_a,\pi_b \in \Pi} \sum_{c \in \mathcal{C}} \frac{\lambda_c^2}{n_c}\mathbb{E}\left[\Delta^2(W_i^c; \pi_a, \pi_b)\right] + 32U_{\lambda,n_{\mathcal{C}}}\mathfrak{R}_{\lambda,n_{\mathcal{C}}}(\mathcal{F}_\Pi) \\
&= \frac{V_{\lambda,n_{\mathcal{C}}}}{n} + 128U\mathfrak{R}_{\lambda,n_{\mathcal{C}}}(\mathcal{F}_\Pi)s_{\lambda,n_{\mathcal{C}}}.
\end{aligned}
$$

Therefore, with this setup, Talagrand's inequality guarantees that with probability at least $1 - \delta$

$$
\begin{aligned}
&\sup_{\pi_a,\pi_b \in \Pi}\left|\Delta_\lambda(\pi_a, \pi_b) - \tilde{\Delta}_\lambda(\pi_a, \pi_b)\right| \\
&\leq \mathbb{E}\left[\sup_{\pi_a,\pi_b \in \Pi}\left|\Delta_\lambda(\pi_a, \pi_b) - \tilde{\Delta}_\lambda(\pi_a, \pi_b)\right|\right] + t \\
&= 4\mathfrak{R}_{\lambda,n_{\mathcal{C}}}(\mathcal{F}_\Pi) + \sqrt{CD\log(C/\delta)} + CU_{\lambda,n_{\mathcal{C}}}\log(C/\delta) \\
&\leq 4\mathfrak{R}_{\lambda,n_{\mathcal{C}}}(\mathcal{F}_\Pi) + \sqrt{C\left(\frac{V_{\lambda,n_{\mathcal{C}}}}{n} + 128U\mathfrak{R}_{\lambda,n_{\mathcal{C}}}(\mathcal{F}_\Pi)s_{\lambda,n_{\mathcal{C}}}\right)\log(C/\delta)} + 4CUs_{\lambda,n_{\mathcal{C}}}\log(C/\delta) \\
&\leq 4\mathfrak{R}_{\lambda,n_{\mathcal{C}}}(\mathcal{F}_\Pi) + \sqrt{C\log(C/\delta)\frac{V_{\lambda,n_{\mathcal{C}}}}{n}} + \sqrt{128UC\log(C/\delta)\mathfrak{R}_{\lambda,n_{\mathcal{C}}}(\mathcal{F}_\Pi)s_{\lambda,n_{\mathcal{C}}}} + 4UC\log(C/\delta)s_{\lambda,n_{\mathcal{C}}} \\
&\leq \left((56 + 24\kappa(\Pi))\sqrt{\frac{V_{\lambda,n_{\mathcal{C}}}}{n}} + o\left(\sqrt{\frac{\mathfrak{s}(\lambda\|\bar{n})}{n}}\right)\right) + \sqrt{C\log(C/\delta)\frac{V_{\lambda,n_{\mathcal{C}}}}{n}} \\
&\quad + \sqrt{\mathcal{O}\left(\sqrt{\frac{\mathfrak{s}(\lambda\|\bar{n})}{n}}\right)o\left(\sqrt{\frac{\mathfrak{s}(\lambda\|\bar{n})}{n}}\right)} + o\left(\sqrt{\frac{\mathfrak{s}(\lambda\|\bar{n})}{n}}\right) \\
&\leq \left(56 + 24\kappa(\Pi) + \sqrt{C\log(C/\delta)}\right)\sqrt{\frac{V_{\lambda,n_{\mathcal{C}}}}{n}} + o\left(\sqrt{\frac{\mathfrak{s}(\lambda\|\bar{n})}{n}}\right) \\
&\leq \left(c_1\kappa(\Pi) + \sqrt{c_2\log(c_2/\delta)}\right)\sqrt{\frac{V_{\lambda,n_{\mathcal{C}}}}{n}} + o\left(\sqrt{\frac{\mathfrak{s}(\lambda\|\bar{n})}{n}}\right),
\end{aligned}
$$

where $c_1 = 24$ and $c_2$ is any constant such that $56 + \sqrt{C\log(C/\delta)} \leq \sqrt{c_2\log(c_2/\delta)}$. Here, we used the bounds previously established in the proof of Proposition 1 that $\mathfrak{R}_{\lambda,n_{\mathcal{C}}}(\mathcal{F}_\Pi) \leq \mathcal{O}\left(\sqrt{\mathfrak{s}(\lambda\|\bar{n})/n}\right)$ and $s_{\lambda,n_{\mathcal{C}}} \leq o\left(\sqrt{\mathfrak{s}(\lambda\|\bar{n})/n}\right)$.

$\square$

## C.4   BOUNDING APPROXIMATE REGRET

**Proposition 3.** *Suppose Assumptions 1, 2, and 3 hold. Then,*

$$\sup_{\pi_a, \pi_b \in \Pi} |\tilde{\Delta}_\lambda(\pi_a, \pi_b) - \hat{\Delta}_\lambda(\pi_a, \pi_b)| \leq o_p \left( \sqrt{\frac{\mathfrak{s}(\lambda \| \bar{n})}{n}} \right)$$

*Proof.* Recall that $\{(X_i^c, A_i^c, Y_i^c)\}_{i=1}^{n_c}$ is the data collected by client $c \in \mathcal{C}$ as described in Section 3.3. We assume each client estimates the local nuisance parameters using a cross-fitting strategy, as discussed in Algorithm 3. Under this strategy, each client $c \in \mathcal{C}$ divides their local dataset into $K$ folds, and for each fold $k$, the client estimates $\mu_c$ and $w_c$ using the rest $K - 1$ folds. Let $k_c : [n_c] \to [K]$ denote the surjective mapping that maps a data point index to its corresponding fold containing the data point. We let $\hat{\mu}_c^{-k_c(i)}$ and $\hat{w}_c^{-k_c(i)}$ denote the estimators of $\mu_c$ and $e_c$ fitted on the $K - 1$ folds of client $c$ other than $k_c(i)$.

As discussed Section 4, recall the oracle AIPW scores

$$\Gamma_i^c(a) = \mu(X_i^c; a) + \left(Y_i^c - \mu(X_i^c; a)\right) w_c(X_i^c; a) \mathbf{1}\{A_i^c = a\}$$

and approximate AIPW scores

$$\hat{\Gamma}_i^c(a) = \hat{\mu}_c^{-k_c(i)}(X_i^c; a) + \left(Y_i^c - \hat{\mu}_c^{-k_c(i)}(X_i^c; a)\right) \hat{w}_c^{-k_c(i)}(X_i^c; a) \mathbf{1}\{A_i^c = a\}$$

for any $a \in \mathcal{A}$, where $k_c(i)$ is the fold corresponding to data point $i$ of client $c$. One can verify that the difference between the oracle and approximate AIPW scores can be expressed as

$$\hat{\Gamma}_i^c(a) - \Gamma_i^c(a) = \Gamma_i^{c\prime}(a) + \Gamma_i^{c\prime\prime}(a) + \Gamma_i^{c\prime\prime\prime}(a),$$

where

$$\Gamma_i^{c\prime}(a) = \left(\hat{\mu}_c^{-k_c(i)}(X_i^c; a) - \mu_c(X_i^c; a)\right) \left(1 - w_c(X_i^c; a)\mathbf{1}\{A_i^c = a\}\right),$$

$$\Gamma_i^{c\prime\prime}(a) = \left(Y_i^c(a) - \mu_c(X_i^c; a)\right) \left(\hat{w}_c^{-k_c(i)}(X_i^c; a) - w_c(X_i^c; a)\right) \mathbf{1}\{A_i^c = a\},$$

$$\Gamma_i^{c\prime\prime\prime}(a) = \left(\mu_c(X_i^c; a) - \hat{\mu}_c^{-k_c(i)}(X_i^c; a)\right) \left(\hat{w}_c^{-k_c(i)}(X_i^c; a) - w_c(X_i^c; a)\right) \mathbf{1}\{A_i^c = a\}.$$

This induces the following decomposition of the approximate regret:

$$\hat{\Delta}_\lambda(\pi_a, \pi_b) - \tilde{\Delta}_\lambda(\pi_a, \pi_b) = S_1(\pi_a, \pi_b) + S_2(\pi_a, \pi_b) + S_3(\pi_a, \pi_b),$$

where

$$S_1(\pi_a, \pi_b) = \sum_{c \in \mathcal{C}} \frac{\lambda_c}{n_c} \sum_{i=1}^{n_c} \Gamma_i^{c\prime}(\pi_a(X_i^c)) - \Gamma_i^{c\prime}(\pi_b(X_i^c)),$$

$$S_2(\pi_a, \pi_b) = \sum_{c \in \mathcal{C}} \frac{\lambda_c}{n_c} \sum_{i=1}^{n_c} \Gamma_i^{c\prime\prime}(\pi_a(X_i^c)) - \Gamma_i^{c\prime\prime}(\pi_b(X_i^c)),$$

$$S_3(\pi_a, \pi_b) = \sum_{c \in \mathcal{C}} \frac{\lambda_c}{n_c} \sum_{i=1}^{n_c} \Gamma_i^{c\prime\prime\prime}(\pi_a(X_i^c)) - \Gamma_i^{c\prime\prime\prime}(\pi_b(X_i^c)).$$

We further decompose $S_1$ and $S_2$ by folds as follows:

$$S_1(\pi_a, \pi_b) = \sum_{k=1}^{K} S_1^k(\pi_a, \pi_b),$$

$$S_2(\pi_a, \pi_b) = \sum_{k=1}^{K} S_2^k(\pi_a, \pi_b),$$

where

$$S_1^k(\pi_a, \pi_b) = \sum_{c \in \mathcal{C}} \frac{\lambda_c}{n_c} \sum_{\{i \mid k_c(i) = k\}} \Gamma_i^{c\prime}(\pi_a(X_i^c)) - \Gamma_i^{c\prime}(\pi_b(X_i^c)),$$

$$S_2^k(\pi_a, \pi_b) = \sum_{c \in \mathcal{C}} \frac{\lambda_c}{n_c} \sum_{\{i \mid k_c(i) = k\}} \Gamma_i^{c\prime\prime}(\pi_a(X_i^c)) - \Gamma_i^{c\prime\prime}(\pi_b(X_i^c)),$$

for each $k \in [K]$. To determine a bound on the approximate regret, we will establish high probability bounds for the worst-case absolute value over policies of each term in this decomposition. For convenience, for any policy $\pi$, we will denote $\pi(x; a) = \mathbf{1}\{\pi(x) = a\}$.

_Bounding $S_1$_: We wish to bound $\sup_{\pi_a, \pi_b \in \Pi} |S_1(\pi_a, \pi_b)|$. We first bound $\sup_{\pi_a, \pi_b \in \Pi} \left| S_1^k(\pi_a, \pi_b) \right|$ for any $k \in [K]$.

First, note that since $\hat{\mu}_c^{-k_c(i)}$ is estimated using data outside fold $k_c(i)$, when we condition on the data outside fold $k_c(i)$, $\hat{\mu}_c^{-k_c(i)}$ is fixed and each term in $S_1(\pi_a, \pi_b)$ is independent. This allows us to compute

$$\mathbb{E}\left[ \Gamma_i^{c\prime}(\pi_a(X_i^c)) - \Gamma_i^c(\pi_b(X_i^c)) \right]$$

$$= \sum_{a \in \mathcal{A}} \mathbb{E}\left[ \left( \pi_a(X_i^c; a) - \pi_b(X_i^c; a) \right) \left( \hat{\mu}_c^{-k_c(i)}(X_i^c; a) - \mu_c(X_i^c; a) \right) \left( 1 - w_c(X_i^c; a)\mathbf{1}\{A_i^c = a\} \right) \right]$$

$$= \sum_{a \in \mathcal{A}} \mathbb{E}\left[ \mathbb{E}\left[ \left( \pi_a(X_i^c; a) - \pi_b(X_i^c; a) \right) \left( \hat{\mu}_c^{-k_c(i)}(X_i^c; a) - \mu_c(X_i^c; a) \right) \left( 1 - w_c(X_i^c; a)\mathbf{1}\{A_i^c = a\} \right) \mid X_i^c \right] \right]$$

$$= \sum_{a \in \mathcal{A}} \mathbb{E}\left[ \left( \pi_a(X_i^c; a) - \pi_b(X_i^c; a) \right) \left( \hat{\mu}_c^{-k_c(i)}(X_i^c; a) - \mu_c(X_i^c; a) \right) \mathbb{E}\left[ 1 - w_c(X_i^c; a)\mathbf{1}\{A_i^c = a\} \mid X_i^c \right] \right] = 0$$

Therefore,

$$K \sup_{\pi_a, \pi_b \in \Pi} \left| S_1^k(\pi_a, \pi_b) \right|$$

$$\leq \sup_{\pi_a, \pi_b \in \Pi} \left| \sum_{c \in \mathcal{C}} \frac{\lambda_c}{n_c/K} \sum_{\{i \mid k_c(i) = k\}} \Gamma_i^{c\prime}(\pi_a(X_i^c)) - \Gamma_i^{c\prime}(\pi_b(X_i^c)) \right|$$

$$= \sup_{\pi_a, \pi_b \in \Pi} \left| \sum_{c \in \mathcal{C}} \frac{\lambda_c}{n_c/K} \sum_{\{i \mid k_c(i) = k\}} \left( \Gamma_i^{c\prime}(\pi_a(X_i^c)) - \Gamma_i^{c\prime}(\pi_b(X_i^c)) \right) - \mathbb{E}\left[ \Gamma_i^{c\prime}(\pi_a(X_i^c)) - \Gamma_i^{c\prime}(\pi_b(X_i^c)) \right] \right|.$$

Identifying $\Gamma_i^{c\prime}$ with $\Gamma_i^c$ and sample sizes $n_{\mathcal{C}}/K$ with $n_{\mathcal{C}}$, the right-hand side in the above inequality is effectively an oracle regret and so we can apply Proposition 2 to obtain that with probability at least $1 - \delta$,

$$K \sup_{\pi_a, \pi_b \in \Pi} \left| S_1^k(\pi_a, \pi_b) \right|$$

$$\leq \sup_{\pi_a, \pi_b \in \Pi} \left| \sum_{c \in \mathcal{C}} \frac{\lambda_c}{n_c/K} \sum_{\{j \mid k_i(j) = k\}} \left( \Gamma_i^{c\prime}(\pi_a(X_i^c)) - \Gamma_i^{c\prime}(\pi_b(X_i^c)) \right) - \mathbb{E}\left[ \Gamma_i^{c\prime}(\pi_a(X_i^c)) - \Gamma_i^{c\prime}(\pi_b(X_i^c)) \right] \right|$$

$$\leq C_{\Pi, \delta} \sqrt{ \frac{ \sup_{\pi_a, \pi_b \in \Pi} \sum_{c \in \mathcal{C}} \frac{\lambda_c^2}{\bar{n}_c} \mathbb{E}\left[ \left( \Gamma_i^{c\prime}(\pi_a(X_i^c)) - \Gamma_i^{c\prime}(\pi_b(X_i^c)) \right)^2 \mid \hat{\mu}_c^{-k_c(i)} \right] }{ n/K } } + o\left( \sqrt{ \frac{\mathfrak{s}(\lambda \| \bar{n})}{n/K} } \right)$$

$$\leq C_{\Pi, \delta} \sqrt{ K \sup_{\pi_a, \pi_b \in \Pi} \sum_{c \in \mathcal{C}} \frac{\lambda_c^2}{n_c} \mathbb{E}\left[ \left( \Gamma_i^{c\prime}(\pi_a(X_i^c)) - \Gamma_i^{c\prime}(\pi_b(X_i^c)) \right)^2 \mid \hat{\mu}_c^{-k_c(i)} \right] } + o\left( \sqrt{ \frac{\mathfrak{s}(\lambda \| \bar{n})}{n} } \right)$$

$$\leq C_{\Pi, \delta} (1/\eta - 1) \sqrt{ 2K \sum_{c \in \mathcal{C}} \frac{\lambda_c^2}{n_c} \mathbb{E}\left[ \| \hat{\mu}_c^{-k_c(i)}(X_i^c) - \mu_c(X_i^c) \|_2^2 \mid \hat{\mu}_c^{-k_c(i)} \right] } + o\left( \sqrt{ \frac{\mathfrak{s}(\lambda \| \bar{n})}{n} } \right),$$

where $C_{\Pi, \delta} = c_1 \kappa(\Pi) + \sqrt{c_2 \log(c_2/\delta)}$ for some universal constants $c_1$ and $c_2$, and $\eta = \min_{c \in \mathcal{C}} \eta_c$ for $\eta_c$ in the overlap assumption stated in in Assumption 1. The last inequality follows from a uniform bound on $\Gamma_i^{c\prime}(\pi_a(X_i^c)) - \Gamma_i^c(\pi_b(X_i^c))$ and the overlap assumption.

By the assumption on finite sample error bounds for the nuisance functions stated in Assumption 3, for every $c \in \mathcal{C}$

$$\mathbb{E}\left[||\hat{\mu}_c^{-k_c(i)}(X_i^c) - \mu_c(X_i^c)||^2 \mid \hat{\mu}_c^{-k_c(i)}\right] \leq \frac{g_c(\alpha_K n_c)}{(\alpha_K n_c)^{\zeta_\mu}},$$

where $\alpha_K = 1 - K^{-1}$, $g_c$ is some decreasing function, and $0 < \zeta_\mu < 1$. Then,

$$\sum_{c \in \mathcal{C}} \frac{\lambda_c^2}{n_c} \mathbb{E}\left[||\hat{\mu}_c^{-k_c(i)}(X_i^c) - \mu_c(X_i^c)||^2 \mid \hat{\mu}_c^{-k_c(i)}\right]$$

$$\leq \sum_{c \in \mathcal{C}} \frac{\lambda_c^2}{n_c} \frac{g_c(\alpha_K n_c)}{(\alpha_K n_c)^{\zeta_\mu}}$$

$$\leq \frac{\max_{c \in \mathcal{C}} g_c(\alpha_K n_c)}{\alpha_K^{\zeta_\mu} \cdot \min_{c \in \mathcal{C}} n_c^{\zeta_\mu}} \sum_{c \in \mathcal{C}} \frac{\lambda_c^2}{n_c}$$

$$\leq \frac{\max_{c \in \mathcal{C}} g_c(\alpha_K n_c)}{\alpha_K^{\zeta_\mu} \cdot \min_{c \in \mathcal{C}} n_c^{\zeta_\mu}} \frac{\mathfrak{s}(\lambda\|\bar{n})}{n}.$$

By the local data size scaling assumption in Assumption 2, for any $c \in \mathcal{C}$, we have that $n_c = \Omega(\nu_c(n))$ where $\nu_c$ is an increasing function. In other words, there exists a constant $\tau > 0$ such that $n_c \geq \tau\nu_c(n)$ for sufficiently large $n$. Then, since $g_c$ is decreasing, $g_c(\alpha_K n_c) < g_c(\tau\alpha_K \nu_c(n))$ for sufficiently large $n$. Moreover, since $\nu_c$ is increasing and $\tau\alpha_K > 0$, $\tilde{\nu}_c = \tau\alpha_K\nu_c$ is also increasing, and since $g_c$ is decreasing, the composition $\tilde{g}_c = g_c \circ \tilde{\nu}_c$ is decreasing. Therefore, $g_c(\alpha_K n_c)$ is asymptotically bounded by a decreasing function $\tilde{g}_c$ of $n$. This observation and the fact that the maximum of a set of decreasing functions is itself decreasing imply that $\max_{c \in \mathcal{C}} g_c(\alpha_K n_c)$ is asymptotically bounded by the decreasing function $\tilde{g}$ defined by $\tilde{g}(n) = \max_{c \in \mathcal{C}} \tilde{g}_c(n)$. In other words,

$$\max_{c \in \mathcal{C}} g_c(\alpha_K n_c) \leq \tilde{g}(n) \leq o(1).$$

Additionally, since $n_c = \Omega(\nu_c(n))$ and $\zeta_\mu > 0$, we also have that

$$\frac{1}{\min_{c \in \mathcal{C}} n_c^{\zeta_\mu}} \leq o(1).$$

These two observations imply

$$\sum_{c \in \mathcal{C}} \frac{\lambda_c^2}{n_c} \mathbb{E}\left[||\hat{\mu}_c^{-k_c(i)}(X_i^c) - \mu_c(X_i^c)||^2 \mid \hat{\mu}_c^{-k_c(i)}\right] \leq \frac{\max_{c \in \mathcal{C}} g_c(\alpha_K n_c)}{\alpha_K^{\zeta_\mu} \cdot \min_{c \in \mathcal{C}} n_c^{\zeta_\mu}} \frac{\mathfrak{s}(\lambda\|\bar{n})}{n} \leq o\left(\frac{\mathfrak{s}(\lambda\|\bar{n})}{n}\right).$$

Therefore,

$$\sup_{\pi_a, \pi_b \in \Pi} \left|S_1^k(\pi_a, \pi_b)\right|$$

$$\leq C_{\Pi,\delta}(1/\eta - 1) \sqrt{\frac{2}{K} \sum_{c \in \mathcal{C}} \frac{\lambda_c^2}{n_c} \mathbb{E}\left[||\hat{\mu}_c^{-k_c(i)}(X_i^c) - \mu_c(X_i^c)||_2^2 \mid \hat{\mu}_c^{-k_c(i)}\right]} + o\left(\sqrt{\frac{\mathfrak{s}(\lambda\|\bar{n})}{n}}\right)$$

$$\leq C_{\Pi,\delta}(1/\eta - 1) \sqrt{\frac{2}{K} \cdot o\left(\frac{\mathfrak{s}(\lambda\|\bar{n})}{n}\right)} + o\left(\sqrt{\frac{\mathfrak{s}(\lambda\|\bar{n})}{n}}\right) \leq o\left(\sqrt{\frac{\mathfrak{s}(\lambda\|\bar{n})}{n}}\right),$$

and

$$\sup_{\pi_a, \pi_b \in \Pi} |S_1(\pi_a, \pi_b)| \leq \sum_{k=1}^{K} \sup_{\pi_a, \pi_b \in \Pi} \left|S_1^k(\pi_a, \pi_b)\right| \leq o\left(\sqrt{\frac{\mathfrak{s}(\lambda\|\bar{n})}{n}}\right).$$

_Bounding $S_2$_: The bound for $\sup_{\pi_a, \pi_b \in \Pi} |S_2(\pi_a, \pi_b)|$ follows the same argument as that of $S_1$. We first bound $\sup_{\pi_a, \pi_b \in \Pi} \left|S_2^k(\pi_a, \pi_b)\right|$ for any $k \in [K]$.

First, note that since $\hat{w}_c^{-k_c(i)}$ is estimated using data outside fold $k_c(i)$, when we condition on the data outside fold $k_c(i)$, $\hat{w}_c^{-k_c(i)}$ is fixed and each term in $S_2(\pi_a, \pi_b)$ is independent. This allows us

to compute

$\mathbb{E}\left[\Gamma_i^{c''}(\pi_a(X_i^c)) - \Gamma_i^{c''}(\pi_b(X_i^c))\right]$

$= \mathbb{E}\left[\sum_{a \in \mathcal{A}} (\pi_a(X_i^c; a) - \pi_b(X_i^c; a))(Y_i^c(a) - \mu_c(X_i^c; a))\left(\hat{w}_c^{-k_c(i)}(X_i^c; a) - w_c(X_i^c; a)\right)\mathbf{1}\{A_i^c = a\}\right]$

$= \mathbb{E}\left[(\pi_a(X_i^c; A_i^c) - \pi_b(X_i^c; A_i^c))(Y_i^c(A_i^c) - \mu_c(X_i^c; A_i^c))\left(\hat{w}_c^{-k_c(i)}(X_i^c; a) - w_c(X_i^c; a)\right)\right]$

$= \mathbb{E}\left[\mathbb{E}\left[(\pi_a(X_i^c; A_i^c) - \pi_b(X_i^c; A_i^c))(Y_i^c(A_i^c) - \mu_c(X_i^c; A_i^c))\left(\hat{w}_c^{-k_c(i)}(X_i^c; a) - w_c(X_i^c; a)\right) \mid X_i^c, A_i^c\right]\right]$

$= \mathbb{E}\left[(\pi_a(X_i^c; A_i^c) - \pi_b(X_i^c; A_i^c))\mathbb{E}\left[Y_i^c(A_i^c) - \mu_c(X_i^c; A_i^c) \mid X_i^c, A_i^c\right]\left(\hat{w}_c^{-k_c(i)}(X_i^c; a) - w_c(X_i^c; a)\right)\right] = 0$

Therefore, we can follow the exact same argument as above, eliciting Proposition 2, to obtain that with probability at least $1 - \delta$,

$K \sup_{\pi_a, \pi_b \in \Pi} \left|S_2^k(\pi_a, \pi_b)\right|$

$\leq \sup_{\pi_a, \pi_b \in \Pi} \left|\sum_{c \in \mathcal{C}} \frac{\lambda_c}{n_c/K} \sum_{\{i \mid k_c(i) = k\}} \Gamma_i^{c''}(\pi_a(X_i^c)) - \Gamma_i^{c''}(\pi_b(X_i^c))\right|$

$= \sup_{\pi_a, \pi_b \in \Pi} \left|\sum_{c \in \mathcal{C}} \frac{\lambda_c}{n_c/K} \sum_{\{i \mid k_c(i) = k\}} \left(\Gamma_i^{c''}(\pi_a(X_i^c)) - \Gamma_i^{c''}(\pi_b(X_i^c))\right) - \mathbb{E}\left[\Gamma_i^{c''}(\pi_a(X_i^c)) - \Gamma_i^{c''}(\pi_b(X_i^c))\right]\right|$

$\leq \sup_{\pi_a, \pi_b \in \Pi} \left|\sum_{c \in \mathcal{C}} \frac{\lambda_c}{n_c/K} \sum_{\{j \mid k_i(j) = k\}} \left(\Gamma_i^{c''}(\pi_a(X_i^c)) - \Gamma_i^{c''}(\pi_b(X_i^c))\right) - \mathbb{E}\left[\Gamma_i^{c''}(\pi_a(X_i^c)) - \Gamma_i^{c''}(\pi_b(X_i^c))\right]\right|$

$\leq C_{\Pi, \delta}\sqrt{\frac{\sup_{\pi_a, \pi_b \in \Pi} \sum_{c \in \mathcal{C}} \frac{\lambda_c^2}{\bar{n}_c}\mathbb{E}\left[\left(\Gamma_i^{c''}(\pi_a(X_i^c)) - \Gamma_i^{c''}(\pi_b(X_i^c))\right)^2 \mid \hat{w}_c^{-k_c(i)}\right]}{n/K}} + o\left(\sqrt{\frac{\mathfrak{s}(\lambda \| \bar{n})}{n/K}}\right)$

$\leq C_{\Pi, \delta}\sqrt{K \sup_{\pi_a, \pi_b \in \Pi} \sum_{c \in \mathcal{C}} \frac{\lambda_c^2}{n_c}\mathbb{E}\left[\left(\Gamma_i^{c''}(\pi_a(X_i^c)) - \Gamma_i^{c''}(\pi_b(X_i^c))\right)^2 \mid \hat{w}_c^{-k_c(i)}\right]} + o\left(\sqrt{\frac{\mathfrak{s}(\lambda \| \bar{n})}{n}}\right)$

$\leq C_{\Pi, \delta}\sqrt{4BK \sum_{c \in \mathcal{C}} \frac{\lambda_c^2}{n_c}\mathbb{E}\left[\|\hat{w}_c^{-k_c(i)}(X_i^c) - w_c(X_i^c)\|_2^2 \mid \hat{w}_c^{-k_c(i)}\right]} + o\left(\sqrt{\frac{\mathfrak{s}(\lambda \| \bar{n})}{n}}\right),$

where $C_{\Pi, \delta} = c_1 \kappa(\Pi) + \sqrt{c_2 \log(c_2/\delta)}$ for some universal constants $c_1$ and $c_2$, and $B = \max_{c \in \mathcal{C}} B_c$ for the bounds $B_c$ on the outcomes defined in Assumption 1. The last inequality follows from a uniform bound on $\Gamma_i^{c''}(\pi_a(X_i^c)) - \Gamma_i^{c''}(\pi_b(X_i^c))$.

We follow the exact same argument as above to get

$$\sum_{c \in \mathcal{C}} \frac{\lambda_c^2}{n_c}\mathbb{E}\left[\|\hat{w}_c^{-k_c(i)}(X_i^c) - w_c(X_i^c)\|^2 \mid \hat{w}_c^{-k_c(i)}\right] \leq o\left(\frac{\mathfrak{s}(\lambda \| \bar{n})}{n}\right).$$

Therefore,

$\sup_{\pi_a, \pi_b \in \Pi} \left|S_2^k(\pi_a, \pi_b)\right|$

$\leq C_{\Pi, \delta}\sqrt{\frac{4B}{K} \sum_{c \in \mathcal{C}} \frac{\lambda_c^2}{n_c}\mathbb{E}\left[\|\hat{w}_c^{-k_c(i)}(X_i^c) - w_c(X_i^c)\|_2^2 \mid \hat{w}_c^{-k_c(i)}\right]} + o\left(\sqrt{\frac{\mathfrak{s}(\lambda \| \bar{n})}{n}}\right)$

$\leq C_{\Pi, \delta}\sqrt{\frac{4B}{K} \cdot o\left(\frac{\mathfrak{s}(\lambda \| \bar{n})}{n}\right)} + o\left(\sqrt{\frac{\mathfrak{s}(\lambda \| \bar{n})}{n}}\right)$

$\leq o\left(\sqrt{\frac{\mathfrak{s}(\lambda \| \bar{n})}{n}}\right),$

and

$$\sup_{\pi_a, \pi_b \in \Pi} |S_2(\pi_a, \pi_b)| \leq \sum_{k=1}^{K} \sup_{\pi_a, \pi_b \in \Pi} \left|S_2^k(\pi_a, \pi_b)\right| \leq o\left(\sqrt{\frac{\mathfrak{s}(\lambda \| \bar{n})}{n}}\right).$$

_Bounding $S_3$_: Next, we bound the contribution from $S_3$. We have that

$$\sup_{\pi_a, \pi_b \in \Pi} |S_3(\pi_a, \pi_b)|$$

$$= \sup_{\pi_a, \pi_b \in \Pi} \left|\sum_{c \in \mathcal{C}} \frac{\lambda_c}{n_c} \sum_{i=1}^{n_c} \Gamma_i^{c'''}(\pi_a(X_i^c)) - \Gamma_i^{c'''}(\pi_b(X_i^c))\right|$$

$$\leq 2 \left|\sum_{c \in \mathcal{C}} \frac{\lambda_c}{n_c} \sum_{i=1}^{n_c} \sum_{a \in \mathcal{A}} \left(\mu_c(X_i^c; a) - \hat{\mu}_c^{-k_c(i)}(X_i^c; a)\right)\left(\hat{w}_c^{-k_c(i)}(X_i^c; a) - w_c(X_i^c; a)\right)\right|$$

$$\leq 2 \sqrt{\sum_{c \in \mathcal{C}} \frac{\lambda_c}{n_c} \sum_{i=1}^{n_c} \left\|\mu_c(X_i^c) - \hat{\mu}_c^{-k_c(i)}(X_i^c)\right\|_2^2} \sqrt{\sum_{c \in \mathcal{C}} \frac{\lambda_c}{n_c} \sum_{i=1}^{n_c} \left\|\hat{w}_c^{-k_c(i)}(X_i^c) - w_c(X_i^c)\right\|_2^2}$$

$$\leq 2 \sqrt{\sum_{c \in \mathcal{C}} \lambda_c \frac{g_c(\alpha_K n_c)}{(\alpha_K n_c)^{\zeta_\mu}}} \sqrt{\sum_{c \in \mathcal{C}} \lambda_c \frac{g_c(\alpha_K n_c)}{(\alpha_K n_c)^{\zeta_w}}}$$

$$\leq \frac{2}{\alpha_K^{(\zeta_\mu + \zeta_w)/2}} \sqrt{\max_{c \in \mathcal{C}} \frac{\lambda_c}{n_c^{\zeta_\mu}} \sum_{c \in \mathcal{C}} g_c(\alpha_K n_c)} \sqrt{\max_{c \in \mathcal{C}} \frac{\lambda_c}{n_c^{\zeta_w}} \sum_{c \in \mathcal{C}} g_c(\alpha_K n_c)}$$

$$= \frac{2}{\alpha_K^{(\zeta_\mu + \zeta_w)/2}} \sum_{c \in \mathcal{C}} g_c(\alpha_K n_c) \sqrt{\max_{c \in \mathcal{C}} \frac{\lambda_c^2}{n_c^{\zeta_\mu + \zeta_w}}}$$

$$\leq \frac{2}{\alpha_K^{(\zeta_\mu + \zeta_w)/2}} \sum_{c \in \mathcal{C}} g_c(\alpha_K n_c) \sqrt{\max_{c \in \mathcal{C}} \frac{\lambda_c^2}{n_c}}$$

$$\leq \frac{2}{\alpha_K^{(\zeta_\mu + \zeta_w)/2}} \sum_{c \in \mathcal{C}} g_c(\alpha_K n_c) \sqrt{\sum_{c \in \mathcal{C}} \frac{\lambda_c^2}{n_c}}$$

$$\leq \frac{2}{\alpha_K^{(\zeta_\mu + \zeta_w)/2}} \sum_{c \in \mathcal{C}} g_c(\alpha_K n_c) \sqrt{\frac{\mathfrak{s}(\lambda \| \bar{n})}{n}}.$$

As discussed earlier, $g_c(\alpha_K n_c)$ is asymptotically bounded by a decreasing function of $n$. Since the sum of decreasing functions is decreasing, $\sum_{c \in \mathcal{C}} g_c(\alpha_K n_c)$ is asymptotically bounded by a decreasing function $\tilde{g}$ in $n$. In other words, $\sum_{c \in \mathcal{C}} g_c(\alpha_K n_c) \leq \tilde{g}(n) \leq o(1)$. Therefore,

$$\sup_{\pi_a, \pi_b \in \Pi} |S_3(\pi_a, \pi_b)| \leq \frac{2}{\alpha_K^{(\zeta_\mu + \zeta_w)/2}} \cdot o(1) \cdot \sqrt{\frac{\mathfrak{s}(\lambda \| \bar{n})}{n}} \leq o\left(\sqrt{\frac{\mathfrak{s}(\lambda \| \bar{n})}{n}}\right).$$

Putting all the above bounds together, we have

$$\sup_{\pi_a, \pi_b \in \Pi} |\tilde{\Delta}_\lambda(\pi_a, \pi_b) - \hat{\Delta}_\lambda(\pi_a, \pi_b)| \leq \sup_{\pi_a, \pi_b \in \Pi} |S_1(\pi_a, \pi_b) + S_2(\pi_a, \pi_b) + S_3(\pi_a, \pi_b)|$$

$$\leq \sup_{\pi_a, \pi_b \in \Pi} |S_1(\pi_a, \pi_b)| + \sup_{\pi_a, \pi_b \in \Pi} |S_2(\pi_a, \pi_b)| + \sup_{\pi_a, \pi_b \in \Pi} |S_3(\pi_a, \pi_b)|$$

$$\leq o\left(\sqrt{\frac{\mathfrak{s}(\lambda \| \bar{n})}{n}}\right).$$

$\square$

## C.5 PROOF OF THEOREM 1

**Theorem 1** (Global Regret Bound). *Suppose Assumption 1, 2, and 3 hold. Then, with probability at least $1 - \delta$,*

$$R_\lambda(\hat{\pi}_\lambda) \leq \left(c_1 \kappa(\Pi) + \sqrt{c_2 \log(c_2/\delta)}\right) \sqrt{V \cdot \frac{\mathfrak{s}(\lambda \| \bar{n})}{n}} + o_p\left(\sqrt{\frac{\mathfrak{s}(\lambda \| \bar{n})}{n}}\right), \tag{11}$$

*where $c_1$ and $c_2$ are universal constants and $V = \max_{c \in \mathcal{C}} \sup_{\pi \in \Pi} \mathbb{E}_{\bar{\mathcal{D}}_c}\left[\Gamma^c(\pi(X^c))^2\right]$.*

*Proof.* Let $\pi_\lambda^* = \arg\max_{\pi \in \Pi} Q_\lambda(\pi)$. Using the results of Propositions 2 and 3, with probability at least $1 - \delta$, we have

$$\begin{aligned}
R_\lambda(\hat{\pi}_\lambda) &= Q_\lambda(\pi_\lambda^*) - Q_\lambda(\hat{\pi}_\lambda) \\
&= \left(Q_\lambda(\pi_\lambda^*) - Q_\lambda(\hat{\pi}_\lambda)\right) - \left(\hat{Q}_\lambda(\pi_\lambda^*) - \hat{Q}_\lambda(\hat{\pi}_\lambda)\right) + \left(\hat{Q}_\lambda(\pi_\lambda^*) - \hat{Q}_\lambda(\hat{\pi}_\lambda)\right) \\
&= \Delta_\lambda(\pi_\lambda^*, \hat{\pi}_\lambda) - \hat{\Delta}_\lambda(\pi_\lambda^*, \hat{\pi}_\lambda) + \left(\hat{Q}_\lambda(\pi_\lambda^*) - \hat{Q}_\lambda(\hat{\pi}_\lambda)\right) \\
&\leq \Delta_\lambda(\pi_\lambda^*, \hat{\pi}_\lambda) - \hat{\Delta}_\lambda(\pi_\lambda^*, \hat{\pi}_\lambda) \\
&\leq \sup_{\pi_a, \pi_b \in \Pi} |\Delta_\lambda(\pi_a, \pi_b) - \hat{\Delta}_\lambda(\pi_a, \pi_b)| \\
&\leq \sup_{\pi_a, \pi_b \in \Pi} |\Delta_\lambda(\pi_a, \pi_b) - \tilde{\Delta}_\lambda(\pi_a, \pi_b)| + \sup_{\pi_a, \pi_b \in \Pi} |\tilde{\Delta}_\lambda(\pi_a, \pi_b) - \hat{\Delta}_\lambda(\pi_a, \pi_b)| \\
&\leq \left(\left(c_1 \kappa(\Pi) + \sqrt{c_2 \log(c_2/\delta)}\right) \sqrt{\frac{V_{\lambda, n_\mathcal{C}}}{n}} + o\left(\sqrt{\frac{\mathfrak{s}(\lambda \| \bar{n})}{n}}\right)\right) + o_p\left(\sqrt{\frac{\mathfrak{s}(\lambda \| \bar{n})}{n}}\right) \\
&\leq \left(c_1 \kappa(\Pi) + \sqrt{c_2 \log(c_2/\delta)}\right) \sqrt{\frac{V_{\lambda, n_\mathcal{C}}}{n}} + o_p\left(\sqrt{\frac{\mathfrak{s}(\lambda \| \bar{n})}{n}}\right),
\end{aligned}$$

where $c_1$ and $c_2$ are universal constants. Lastly, we decompose the weighted variance term by

$$\begin{aligned}
V_{\lambda, n_\mathcal{C}} &= \sup_{\pi_a, \pi_b \in \Pi} \sum_{c \in \mathcal{C}} \frac{\lambda_c^2}{\bar{n}_c} \mathbb{E}_{Z^c \sim \bar{\mathcal{D}}_c}\left[\left(\Gamma^c(\pi_a(X^c)) - \Gamma^c(\pi_b(X^c))\right)^2\right] \\
&\leq \max_{c \in \mathcal{C}} \sup_{\pi_a, \pi_b \in \Pi} \mathbb{E}_{Z^c \sim \bar{\mathcal{D}}_c}\left[\left(\Gamma^c(\pi_a(X^c)) - \Gamma^c(\pi_b(X^c))\right)^2\right] \cdot \sum_{c \in \mathcal{C}} \frac{\lambda_c^2}{\bar{n}_c} \\
&\leq 4 \cdot \max_{c \in \mathcal{C}} \sup_{\pi \in \Pi} \mathbb{E}_{Z^c \sim \bar{\mathcal{D}}_c}\left[\Gamma^c(\pi(X^c))\right] \cdot \sum_{c \in \mathcal{C}} \frac{\lambda_c^2}{\bar{n}_c} \\
&= 4V \cdot \mathfrak{s}(\lambda \| \bar{n}).
\end{aligned}$$

We absorb the factor of $\sqrt{4}$ into the universal constants to get the desired result. $\qquad\square$

## D BOUNDING LOCAL REGRET

### D.1 PROOF OF THEOREM 2

**Theorem 2** (Local Regret Bound). *Suppose Assumption 1 holds. Then, for any client $c \in \mathcal{C}$,*
$$R_c(\hat{\pi}_\lambda) \leq U \cdot \mathrm{TV}(\bar{\mathcal{D}}_c, \bar{\mathcal{D}}_\lambda) + R_\lambda(\hat{\pi}_\lambda), \tag{12}$$
*where $U = 3B/\eta$ with $B = \max_{c \in \mathcal{C}} B_c$ and $\eta = \min_{c \in \mathcal{C}} \eta_c$, and $\mathrm{TV}$ is the total variation distance.*

*Proof.* Let $\pi_c^* = \arg\max_{\pi \in \Pi} Q_c(\pi)$. Then,

$$
\begin{aligned}
R_c(\hat{\pi}_\lambda) &= Q_c(\pi_c^*) - Q_c(\hat{\pi}_\lambda) \\
&= Q_c(\pi_c^*) - Q_c(\hat{\pi}_\lambda) \mp Q_\lambda(\pi_c^*) \pm Q_\lambda(\hat{\pi}_\lambda) \\
&= \left(Q_c(\pi_c^*) - Q_\lambda(\pi_c^*)\right) + \left(Q_\lambda(\hat{\pi}_\lambda) - Q_c(\hat{\pi}_\lambda)\right) + \left(Q_\lambda(\pi_c^*) - Q_\lambda(\hat{\pi}_\lambda)\right) \\
&\le 2 \sup_{\pi \in \Pi} |Q_c(\pi) - Q_\lambda(\pi)| + \left(Q_\lambda(\pi_c^*) - Q_\lambda(\hat{\pi}_\lambda)\right) \\
&\le 2 \sup_{\pi \in \Pi} |Q_c(\pi) - Q_\lambda(\pi)| + \left(Q_\lambda(\pi_\lambda^*) - Q_\lambda(\hat{\pi}_\lambda)\right) \\
&= 2 \sup_{\pi \in \Pi} |Q_c(\pi) - Q_\lambda(\pi)| + R_\lambda(\hat{\pi}_\lambda).
\end{aligned}
$$

By Lemma 6, we can express the the local policy value as

$$
Q_c(\pi) = \mathop{\mathbb{E}}_{Z \sim \bar{\mathcal{D}}_c} \left[\Gamma(\pi(X))\right]
$$

where $(\Gamma(a_1), \ldots, \Gamma(a_d))$ are the constructed AIPW scores from a context-action-outcomes sample $Z = (X, A, Y(a_1), \ldots, Y(a_d)) \sim \bar{\mathcal{D}}_c$. In addition, the global policy value can be expressed as

$$
Q_\lambda(\pi) = \mathop{\mathbb{E}}_{Z \sim \bar{\mathcal{D}}_\lambda} \left[\Gamma(\pi(X))\right].
$$

where $(\Gamma(a_1), \ldots, \Gamma(a_d))$ are the constructed AIPW scores from a context-action-outcomes sample $Z = (X, A, Y(a_1), \ldots, Y(a_d))$ such that $c \sim \lambda$ and then $Z \sim \bar{\mathcal{D}}_c$. Therefore,

$$
\sup_{\pi \in \Pi} |Q_c(\pi) - Q_\lambda(\pi)| = \sup_{\pi \in \Pi} \left| \mathop{\mathbb{E}}_{Z \sim \bar{\mathcal{D}}_c} \left[\Gamma(\pi(X))\right] - \mathop{\mathbb{E}}_{Z \sim \bar{\mathcal{D}}_\lambda} \left[\Gamma(\pi(X))\right] \right| \tag{29}
$$

By the boundedness and overlap assumption in Assumption 1, one can easily verify the uniform bound

$$
|\Gamma^c(a)| \le B_c + 2B_c/\eta_c \le 3B_c/\eta_c \le 3B/\eta =: U
$$

for any constructed AIPW score $\Gamma^c(a)$ for any $a \in \mathcal{A}$ and any client $c \in \mathcal{C}$. Therefore, Equation equation 29 is equivalent to the integral probability metric distance (Sriperumbudur et al., 2009) between $\bar{\mathcal{D}}_c$ and $\bar{\mathcal{D}}_\lambda$ under uniformly bounded test functions

$$
\{Q(T(\cdot); \pi) \mid \pi \in \Pi\} \subset \mathcal{F}_\infty^U := \{f \mid \|f\|_\infty \le U\},
$$

where $T(X, A, Y(a_1), \ldots, Y(a_d)) = (X, \Gamma(a_1), \ldots, \Gamma(a_d))$. Thus,

$$
\begin{aligned}
\sup_{\pi \in \Pi} |Q_c(\pi) - Q_\lambda(\pi)| &= \sup_{\pi \in \Pi} \left| \mathop{\mathbb{E}}_{Z \sim \bar{\mathcal{D}}_c} \left[Q(T(Z); \pi)\right] - \mathop{\mathbb{E}}_{Z \sim \bar{\mathcal{D}}_\lambda} \left[Q(T(Z); \pi)\right] \right| \\
&\le \sup_{f \in \mathcal{F}_\infty^U} \left| \mathop{\mathbb{E}}_{Z \sim \bar{\mathcal{D}}_c} \left[f(Z)\right] - \mathop{\mathbb{E}}_{Z \sim \bar{\mathcal{D}}_\lambda} \left[f(Z)\right] \right| \\
&= U \cdot \mathrm{TV}(\bar{\mathcal{D}}_c, \bar{\mathcal{D}}_\lambda).
\end{aligned}
$$

The last equality holds by the definition of the total variation distance as an integral probability metric with uniformly bounded test functions. $\square$

## D.2 DISTRIBUTION SHIFT BOUND

First, we state some important properties of the KL divergence.

**Lemma 8.** *The KL divergence has the following properties.*

- *Tensorization Property: Let $\mathcal{P} = \prod_{i=1}^m \mathcal{P}_i$ and $\mathcal{Q} = \prod_{i=1}^m \mathcal{Q}_i$ be two product distributions. Then,*

$$
\mathrm{KL}(\mathcal{P}||\mathcal{Q}) = \sum_{i=1}^m \mathrm{KL}(\mathcal{P}_i||\mathcal{Q}_i).
$$

- *Chain Rule: Let $\mathcal{P}_{XY} = \mathcal{P}_X \mathcal{P}_{Y|X}$ and $\mathcal{Q}_{XY} = \mathcal{Q}_X \mathcal{Q}_{Y|X}$ be two distributions for a pair of random variables $X, Y$. Then,*

$$
\mathrm{KL}(\mathcal{P}_{XY}||\mathcal{Q}_{XY}) = \mathrm{KL}(\mathcal{P}_X||\mathcal{Q}_X) + \mathrm{KL}(\mathcal{P}_{Y|X}||\mathcal{Q}_{Y|X} \mid \mathcal{P}_X)
$$

  *where*

$$
\mathrm{KL}(\mathcal{P}_{Y|X}||\mathcal{Q}_{Y|X} \mid \mathcal{P}_X) = \mathop{\mathbb{E}}_{X \sim \mathcal{P}_X} \left[\mathrm{KL}(\mathcal{P}_{Y|X}||\mathcal{Q}_{Y|X})\right].
$$

Then, we can use these properties to additively separate the sources of distribution shift in our local regret bound.

**Theorem 3** (Local Distribution Shift Bound). *For any given client $c \in \mathcal{C}$, suppose $(X^c, \vec{Y}^c) \sim \mathcal{D}_c$. We let $p_{X^c}$ denote the marginal distribution of $X^c$ and let $p_{\vec{Y}^c|X^c}$ denote the conditional distribution of $\vec{Y}^c$ given $X^c$. Then, the irreducible distribution shift term in the local regret bound can be further bounded as*

$$\text{TV}(\bar{\mathcal{D}}_c, \bar{\mathcal{D}}_\lambda) \leq \underset{k \sim \lambda}{\mathbb{E}} \left[ \sqrt{\text{KL}(p_{X^c}||p_{X^k})} + \sqrt{\text{KL}(e_c||e_k)} + \sqrt{\text{KL}(p_{\vec{Y}^c|X^c}||p_{\vec{Y}^k|X^k})} \right], \quad (13)$$

*where $\text{TV}$ is the total variation distance and $\text{KL}$ is the Kullback-Leibler divergence.*[5]

*Proof.* For any $c \in \mathcal{C}$, the joint probability density function of $\bar{\mathcal{D}}_c$ factorizes as

$$p_{X^c, A^c, \vec{Y}^c}(x, a, y) = p_{X^c}(x)e_c(a|x)p_{\vec{Y}^c|X^c, A^c}(y|x, a) = p_{X^c}(x)e_c(a|x)p_{\vec{Y}^c|X^c}(y|x)$$

for any $(x, a, y) \in \mathcal{X} \times \mathcal{A} \times \mathcal{Y}^d$, where the last equality holds by the unconfoundedness property stated in Assumption 1. Next, let $\Sigma$ be the $\sigma$-field over $\mathcal{X} \times \mathcal{A} \times \mathcal{Y}^d$ on which the $\bar{\mathcal{D}}_c$ are defined. We have that

$$\begin{aligned}
\text{TV}(\bar{\mathcal{D}}_c, \bar{\mathcal{D}}_\lambda) &= \sup_{A \subset \Sigma} \left| \bar{\mathcal{D}}_c(A) - \bar{\mathcal{D}}_\lambda(A) \right| \\
&= \sup_{A \in \Sigma} \left| \bar{\mathcal{D}}_c(A) - \sum_{c \in \mathcal{C}} \lambda_c \bar{\mathcal{D}}_k(A) \right| \\
&= \sup_{A \in \Sigma} \left| \sum_{c \in \mathcal{C}} \lambda_c \left( \bar{\mathcal{D}}_c(A) - \bar{\mathcal{D}}_k(A) \right) \right| \\
&\leq \sup_{A \in \Sigma} \sum_{c \in \mathcal{C}} \lambda_c \left| \bar{\mathcal{D}}_c(A) - \bar{\mathcal{D}}_k(A) \right| \\
&\leq \sum_{c \in \mathcal{C}} \lambda_c \sup_{A \in \Sigma} \left| \bar{\mathcal{D}}_c(A) - \bar{\mathcal{D}}_k(A) \right| \\
&= \underset{k \sim \lambda}{\mathbb{E}} \left[ \text{TV}(\bar{\mathcal{D}}_c, \bar{\mathcal{D}}_k) \right] \\
&\leq \underset{k \sim \lambda}{\mathbb{E}} \left[ \sqrt{\text{KL}(\bar{\mathcal{D}}_c||\bar{\mathcal{D}}_k)} \right],
\end{aligned}$$

where the last inequality holds by Pinsker's inequality. Moreover,

$$\begin{aligned}
\text{KL}(\bar{\mathcal{D}}_c||\bar{\mathcal{D}}_k) &= \text{KL}(p_{X^c, A^c, \vec{Y}^c}||p_{X_k, A_k, \vec{Y}_k}) \\
&= \text{KL}(p_{X^c}||p_{X^k}) + \text{KL}(e_c p_{\vec{Y}^c|X^c}||e_k p_{\vec{Y}^k|X^k} \mid p_{X^c}) \\
&= \text{KL}\left(p_{X^c}||p_{X^k}\right) + \text{KL}\left(e_c||e_k \mid p_{X^c}\right) + \text{KL}\left(p_{\vec{Y}^c|X^c}||p_{\vec{Y}^k|X^k} \mid p_{X^c}\right) \\
&= \text{KL}\left(p_{X^c}||p_{X^k}\right) + \text{KL}\left(e_c||e_k\right) + \text{KL}\left(p_{\vec{Y}^c|X^c}||p_{\vec{Y}^k|X^k}\right),
\end{aligned}$$

where the first equality holds by the chain rule of the KL divergence and the second equality holds by the tensorization property of KL divergence. In the last inequality, for the sake of brevity, we just get rid of the explicit marker representing conditional KL divergence. It is understood that when the distributions are conditional distributions, their KL divergence is a conditional KL divergence. $\square$

### D.3 ALTERNATIVE LOCAL REGRET BOUND

We provide an alternative local regret bound that is applicable in scenarios where the AIPW score variance is significantly less than the AIPW score range.

**Theorem 4.** *Suppose Assumption 1 holds. Then, for any $c \in \mathcal{C}$,*

$$R_c(\hat{\pi}_\lambda) \leq \sqrt{4V \cdot \chi^2(\bar{\mathcal{D}}_c||\bar{\mathcal{D}}_\lambda)} + R_\lambda(\hat{\pi}_\lambda),$$

*where $V = \max_{c \in \mathcal{C}} \sup_{\pi \in \Pi} \mathbb{E}_{\bar{\mathcal{D}}_c}[\Gamma^c(\pi(X^c))]$.*

---

[5]Note that the last two terms in the expectation of this inequality are conditional KL divergences on $p_{X^c}$. See Appendix D.2 for more details.

*Proof.* As shown in Theorem 2,

$$R_c(\hat{\pi}_\lambda) \le 2 \sup_{\pi \in \Pi} |Q_c(\pi) - Q_\lambda(\pi)| + R_\lambda(\hat{\pi}_\lambda).$$

Thus, we seek a new bound on the first term due to distribution shift.

Let $p_c(z)$ and $p_\lambda(z)$ for any $z \in \mathcal{X} \times \mathcal{A} \times \mathcal{Y}^d$ be the joint probability density functions of $\bar{\mathcal{D}}_c$ and $\bar{\mathcal{D}}_\lambda$, respectively. Additionally, for any $c \in \mathcal{C}$ and any $Z \sim \bar{\mathcal{D}}_c$, let $f(Z; \pi) = \Gamma^c(\pi(X^c))$. Then, we can do the following calculations to get

$$
\begin{aligned}
\sup_{\pi \in \Pi} |Q_c(\pi) - Q_\lambda(\pi)| &= \sup_{\pi \in \Pi} \left| \mathbb{E}_{Z^c \sim \bar{\mathcal{D}}_c} [\Gamma^c(\pi(X^c))] - \mathbb{E}_{c \sim \lambda} \mathbb{E}_{Z^c \sim \bar{\mathcal{D}}_c} [\Gamma^c(\pi(X^c))] \right| \\
&= \sup_{\pi \in \Pi} \left| \mathbb{E}_{Z \sim \bar{\mathcal{D}}_c} [f(Z; \pi)] - \mathbb{E}_{Z \sim \bar{\mathcal{D}}_\lambda} [f(Z; \pi)] \right| \\
&= \sup_{\pi \in \Pi} \left| \int f(z; \pi) p_c(z) dz - \int f(z; \pi) p_\lambda(z) dz \right| \\
&= \sup_{\pi \in \Pi} \left| \int f(z; \pi) \sqrt{p_\lambda(z)} \left( \frac{p_c(z) - p_\lambda(z)}{\sqrt{p_\lambda(z)}} \right) dz \right| \\
&\le \sup_{\pi \in \Pi} \sqrt{\int f(z; \pi)^2 p_\lambda(z) dz \cdot \int \frac{(p_c(z) - p_\lambda(z))^2}{p_\lambda(z)} dz} \\
&= \sup_{\pi \in \Pi} \sqrt{\mathbb{E}_{Z \sim \bar{\mathcal{D}}_\lambda} [f(Z; \pi)^2] \cdot \chi^2(\bar{\mathcal{D}}_c || \bar{\mathcal{D}}_\lambda)} \\
&\le \sup_{\pi \in \Pi} \sqrt{\max_{c \in \mathcal{C}} \mathbb{E}_{Z \sim \bar{\mathcal{D}}_c} [\Gamma^c(\pi(X^c))^2] \cdot \chi^2(\bar{\mathcal{D}}_c || \bar{\mathcal{D}}_\lambda)} \\
&= \sqrt{V \cdot \chi^2(\bar{\mathcal{D}}_c || \bar{\mathcal{D}}_\lambda)}.
\end{aligned}
$$

Thus, we get the desired result. $\qquad \square$

Compare the distribution shift term in this alternate result to the distribution shift term $U \cdot \mathrm{TV}(\bar{\mathcal{D}}_c, \bar{\mathcal{D}}_\lambda)$ in the local regret bound we established in Theorem 2. The alternate bound is useful in that it does not rely on bounded AIPW scores, and instead is scaled by the maximum variance of the AIPW scores, which may be smaller than the range and it also appears in our global regret bound. Therefore, it is a more natural bound in this sense. However, the chi-squared divergence does not have a chain rule that would allow us to additively separate the sources of distribution shift in this bound, as we did in Proposition 3. The reason for this limitation is that the chi-squared divergence cannot be bounded by the KL divergence to leverage its chain rule as we did for the TV distance. In this sense, this alternate bound is not useful for elucidating the contributions of distribution shift in the local regret bound.

## E  VALUE OF INFORMATION

The local regret bound result in Theorem 2 is useful to capture the value of information provided by the central server. Suppose a given client $c \in \mathcal{C}$ has agency to decide whether to participate in the federated system including all other clients. If we consider the client as a local regret-minimizing agent, we can use the dominant terms in the appropriate local regret bounds to model the expected utility of the client. In particular, using prior results of standard offline policy learning (Zhou et al., 2023) and our findings in Theorems 1 and 2, the value of information provided by the central server can be modeled as the comparison of the client's utility (as captured by the negative local regret) with and without participation

$$\mathcal{V}_c(\lambda) = C_0 \kappa(\Pi) \sqrt{V_c/n_c} - C_1 \kappa(\Pi) \sqrt{V\mathfrak{s}(\lambda \| \bar{n})/n} - U \cdot \mathrm{TV}(\bar{\mathcal{D}}_c, \bar{\mathcal{D}}_\lambda),$$

where $C_0, C_1$ are universal constants and $V_c = \sup_{\pi \in \Pi} \mathbb{E}[\Gamma^c(\pi(X^c))^2]$ is the local AIPW variance.

Then, we say it is more valuable for client $c$ to participate in federation if $\mathcal{V}_c(\lambda) > 0$. One can easily show that this condition is satisfied if and only if

$$\text{TV}(\bar{\mathcal{D}}_c, \bar{\mathcal{D}}_\lambda) < \alpha r_c/U \quad \wedge \quad \mathfrak{s}(\lambda\|\bar{n}) < \beta^2 r_c^2/r^2$$

for some $\alpha + \beta \leq 1$, where $r_c = C_0\kappa(\Pi)\sqrt{V_c/n_c}$ is the local regret bound of the locally trained model and $r = C_1\kappa(\Pi)\sqrt{V/n}$ is the global regret bound of the globally trained model under no skewness. The $\alpha$ and $\beta$ factors indicate a trade-off between distribution shift and skewness of the two conditions. If there is low distribution shift, the skewness can be large.

Thus, we see how the design choice on the client distribution must balance a scaled trade-off to achieve relative low skewness and relative low expected distribution shift. Indeed, in the experiments in Section 7 we see how a skewed client distribution can help improve the local regret guarantees of a heterogeneous client. We observe from the first condition that the client distribution shift must must be smaller than the local regret of the locally trained model relative to a scaled range of the data. Intuitively, this states that the client will not benefit from federation if the regret they suffer due to their distribution shift from the global mixture distribution is greater than the relative local regret from just training locally. The second condition states that the skewness must be less than the local regret relative to the global regret. Large relative variance or small relative sample size are the primary factors that can lead small relative regret and therefore tight limitations on skewness budget. Overall, all of these conditions can be satisfied under sufficiently low expected distribution shift from the global distribution, low client distribution skewness, comparable AIPW variance across clients, and large global sample size relative to the local sample size.

However, it should be noted that this analysis simplifies the setting by considering only a single client that unilaterally decides to participate in the federation, without considering the choices of other clients. A more comprehensive analysis would assess the value of information provided by the central server in an equilibrium of clients with agency to participate. Game-theoretic aspects are crucial in this context, necessitating an understanding of client behavior and incentives in federated settings. Recent research has started to delve into game-theoretic considerations in federated supervised learning. Donahue & Kleinberg (2021) provided valuable insights into the behavior of self-interested, error-minimizing clients forming federated coalitions to learn supervised models. Moreover, designing incentive mechanisms in federated learning has been identified as a significant research area (Zhan et al., 2021b). This work aims to understand the optimal ways to incentivize clients to share their data. Applying these concepts to our setting would offer valuable insights on the incentives and behavior that motivate clients to participate in federated policy learning systems.

# F    ADDITIONAL ALGORITHM DETAILS

## F.1    NUISANCE PARAMETER ESTIMATION

Our results rely on efficient estimation of $Q_\lambda(\pi)$ for any policy $\pi$, which in turn relies on efficient estimation of $Q_c(\pi)$. We leverage ideas of double machine learning (Chernozhukov et al., 2018) to guarantee efficient policy value estimation given only high-level conditions on the predictive accuracy of machine learning methods on estimating the nuisance parameters of doubly robust policy value estimators. In this work, we use machine learning and cross-fitting strategies to estimate the nuisance parameters locally. The nuisance parameter estimates must satisfy the conditions of Assumption 3. Under these conditions, extensions of the results of (Chernozhukov et al., 2018; Athey & Wager, 2021) would imply that the doubly robust local policy value estimates $\hat{Q}_c(\pi)$ for any policy $\pi$ are asymptotically efficient for estimating $Q_c(\pi)$.

The conditions and estimators that guarantee these error assumptions have been extensively studied in the estimation literature. These include parametric or smoothness assumptions for non-parametric estimation. The conditional response function $\mu_c(x; a) = \mathbb{E}_{\bar{\mathcal{D}}_c}[Y^c(a)|X^c = x]$ can be estimated by regressing observed rewards on observed contexts. The inverse conditional propensity function $w_c(x; a) = 1/\mathbb{P}_{\bar{\mathcal{D}}_c}(A^c = a|X^c = x)$ can be estimated by estimating the conditional propensity function $e_c(x; a) = \mathbb{P}_{\bar{\mathcal{D}}_c}(A^c = a|X^c = x)$ and then taking the inverse. Under sufficient regularity and overlap assumptions, this gives accurate estimates. We can take any flexible approach to estimate these nuisance parameters. We could use standard parametric estimation methods like logistic regression and linear regression, or we could use non-parametric methods like classification and

regression forests to make more conservative assumptions on the true models. Lastly, we note that if it is known that some clients have the same data-generating distribution, it should be possible to learn the nuisance parameters across similar clients.

In our experiments, we decided to estimate the $\mu_c$ with linear regression and $e_c$ with logistic regression. We used the sklearn Python package to fit the nuisance parameters. The true expected rewards are non-linear but the propensities are simple uniform probabilities. So our experiments emulate the scenario where accurate estimation of $\mu_c$ is not perfectly possible but accurate estimation of $e_c$ is easy, thus leveraging the properties of double machine learning for policy value estimation.

## F.2 CROSS-FITTED AIPW ESTIMATION

Once the nuisance parameters are estimated, they can be used for estimating AIPW scores. Refer to Algorithm 3 for the pseudocode on how we conduct the cross-fitting strategy for AIPW score estimation. Under this strategy, each client $c \in \mathcal{C}$ divides their local dataset into $K$ folds, and for each fold $k$, the client estimates $\mu_c$ and $w_c$ using the rest $K - 1$ folds. During AIPW estimation for a single data point, the nuisance parameter estimate that is used in the AIPW estimate is the one that was not trained on the fold that contained that data point. This cross-fitting estimation strategy is described in additional detail in (Zhou et al., 2023).

---

**Algorithm 3** Cross-fitted AIPW: Client-Side

---

**Require:** local data $\{(X_i^c, A_i^c, Y_i^c)\}_{i=1}^{n_c}$, number of folds $K$
 1: Partition local data into $K$ folds
 2: Define surjective mapping $k_c : [n_c] \rightarrow [K]$ of point index to corresponding fold index
 3: **for** $k = 1, \ldots, K$ **do**
 4:      Fit estimators $\hat{\mu}_c^{-k}$ and $\hat{w}_c^{-k}$ using rest of data not in fold $k$
 5: **end for**
 6: **for** $i = 1, \ldots, n_c$ **do**
 7:      **for** $a \in \mathcal{A}$ **do**
 8:          $\hat{\Gamma}_i^c(a) \leftarrow \hat{\mu}_c^{-k_c(i)}(X_i^c; a) + \left(Y_i^c - \hat{\mu}_c^{-k_c(i)}(X_i^c; a)\right) \cdot \hat{w}_c^{-k_c(i)}(X_i^c; a) \cdot \mathbf{1}\{A_i^c = a\}$
 9:      **end for**
10: **end for**

---

## F.3 IMPLEMENTATION DETAILS

The local optimization problems we face in our formulation in Section 6 are equivalent to cost-sensitive multi-class classification (CSMC). There are many off-the-shelf methods available for such problem. We rely on implementations that can do fast online learning for parametric models in order to be able to do quick iterated updates on the global models at each local client and send these models for global aggregation. So we make use of the cost-sensitive one-against-all (CSOAA) implementation for cost-sensitive multi-class classification in the Vowpal Wabbit library (Langford et al., 2023). This implementation performs separate online regressions of costs on contexts for each action using stochastic gradient descent updates. At inference time, to make an action prediction, the action whose regressor gives the lowest predicted cost is chosen.

The idea behind this method is that if the classifiers admit regression functions that predict the costs, i.e., $\pi_\theta(x) = \arg\max_{a \in \mathcal{A}} f_\theta(x; a)$ for some $f_\theta \in \mathcal{F}_\Theta$ such that $f^*(x; a) \in \mathcal{F}_\Theta$ where $f^*(x; a) = \mathbb{E}[\Gamma^c(a)|X^c = x]$, then efficient regression oracles will return an (near) optimal model (Agarwal et al., 2017). If realizability does not hold one may need to use more computationally expensive CSMC optimization techniques (Beygelzimer et al., 2009). For example, we could use the weighted all pairs (WAP) algorithm (Beygelzimer et al., 2008) that does $\binom{d}{2}$ pairwise binary classifications and predicts the action that receives majority predictions. Unlike the CSOAA implementation, the WAP method is always consistent in that an optimal model for the reduced problem leads to an optimal cost-sensitive prediction classifier. In our experiments, the rewards are non-linear so realizability does not exactly hold. Yet, we still observe good performance with the CSOAA regression-based algorithm.

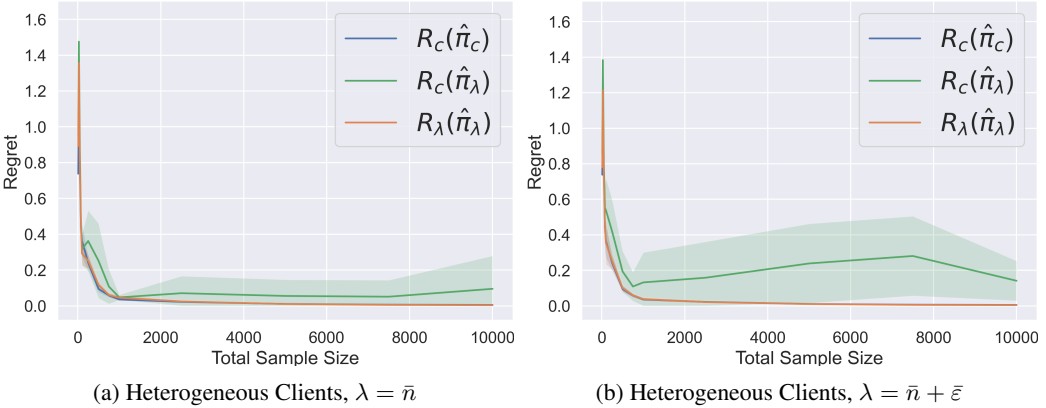

(a) Heterogeneous Clients, $\lambda = \bar{n}$        (b) Heterogeneous Clients, $\lambda = \bar{n} + \bar{\varepsilon}$

Figure 3: Empirical regret curves for simulation experiments. Local regrets are for client 2.

## G   ADDITIONAL EXPERIMENTAL RESULTS

We follow up on the simulations with heterogeneous clients in Section 7. Here, we observe the regret performance for one of the other clients that have less distribution shift from the average. Figure 3a plots the local regret for client 2 of the globally trained policy (green) and the global regret of the globally trained policy (orange), all using the empirical mixture $\lambda = \bar{n}$. For comparison, we also plot the local regret for client 2 of the locally trained policy (blue). The bands show the one standard deviation from the regrets over five different runs. As expected, we see that the other clients have less distribution shift so the local regret of the global policy nearly matches the global regret, similar to what was observed in the homogeneous experiments but with some level of degradation. Indeed, the local distributions nearly match the global distribution, by construction. In Figure 3b we plot the same type of regret curves, but instead with the global policy trained with the skewed mixture. We see that their performance degrades. This is in contrast to what we observe for client 1 where the skewed mixture improved performance. This is because, we are increasing distribution shift as measured by $\mathrm{TV}(\bar{\mathcal{D}}_c, \bar{\mathcal{D}}_\lambda)$. This is another indicator that our theoretical regret guarantees may be tight.

## H   ADDITIONAL DISCUSSION

### H.1   POLICY VALUE ESTIMATION

One might inquire on the need forpolicy value estimation using propensity-weighted strategies, especially under realizability assumptions, when we can just estimate the conditional response function over the class of regressors. One of the issues is the fact that the data was collected under the historical policy which may not necessarily be the target optimal policy so estimating the reward function may lead to policies that are optimal for locations in the decision space where the reward function was able to have been well estimated given the historical data, but the guarantees on data sampled on the optimal target policy may not be as robust. However, there is another another practical reason. (Kitagawa & Tetenov, 2018; Athey & Wager, 2021) provide a good discussion on why separating the assumptions for the nuisance parameters and the class of policies is helpful, and in some cases necessary, component of a comprehensive analysis of policy learning. For one, since the nuisance components are inherent quantities of the distributions are not in control by the central learner, it is a prudent choice to make the least functional form assumptions on these parameters. In contrast, the class of policies $\Pi$ is specified by the central learner and can be used to impose restrictions on the type of policies. For example, for privacy reasons, the central server can impose restrictions on what covariates can be used to learn a policy, such as personally identifiable information. This used in conjunction with federated learning strategies can leverage the use of high-capacity flexible models to learn nuisance parameters locally under no restrictions, but then the central server can impose restrictions on the type of models used for policy evaluation and learning.

## H.2 Limitations & Future Work

There are various limitations to our present work that require further consideration. We discuss some of these limitations throughout the main paper. We provide a more exhaustive list here.

- We make certain assumptions on the data-generating process which may not always be satisfied. Throughout the paper, we discussed how some of these assumptions may be relaxed with additional investigations, such as relaxing the boundedness and uniform overlap assumption in the data-generating distributions.

- In this work, nuisance parameters are estimated locally. If it is known that some clients have the same data-generating distribution, it should be possible to learn the nuisance parameters across similar clients.

- Although our framework offers a great leap towards privacy-preserving policy learning, we did not consider additional differential privacy considerations in this work. It would be worthwhile to explore the effects of differential privacy on our regret analysis and empirical results.

- Our optimization procedure depended on access to efficient online cost-sensitive classification methods. There are many fast implementations that are widely available, but these are restricted to particular parametric classes. However, in many policy learning scenarios, especially in public policy where decisions must be audited, simple tree-based policy classes are preferable. Research on developing efficient federated tree-based policy learning algorithms would be highly valuable for this problem setting. In general, further research needs to be conducted on developing federated methods for learning on general policy classes beyond simple parametric policies, including tree-based policies, finite policies, and neural policies.

- We assumed the mixture distribution $\lambda$ is known. We could extend our work to a more agnostic setting where this mixture distribution is optimized. We could take a more principled approach such as the minimax framework proposed in (Mohri et al., 2019). This would also have implications on robustness and fairness of the global policy.

- The local regret bounds for each client were found to depend on an irreducible term due to distribution shift. Can this irreducible regret be quantified in a federated manner to allow servers to determine if any given client benefits from federation?

- In Section E, we discussed the value of information provided by the central server to an individual client in the scenario where all clients are assumed to be participating in federation. A more complete analysis would consider the value of information in an equilibrium where all clients have agency to participate, rather than just one client.

- Lastly, we leave open the question of whether the bounds we establish are regret optimal. Mohri et al. (2019) discuss how similar skewness-based bounds for distributed supervised learning are optimal. Moreover, in the homogeneous setting, our results immediately reduce to the regret optimal results obtained in (Athey & Wager, 2021; Zhou et al., 2023). Thus, there is good indication that our bounds are regret optimal. We leave establishing lower bounds for future work.

