# OpenReview forum: "Federated Offline Policy Learning with Heterogeneous Observational Data"
_ICLR.cc/2024/Conference — Submitted to ICLR 2024_

### Official Review · Reviewer_38dU · 2023-10-26

**Soundness:** 3 good
**Presentation:** 4 excellent
**Contribution:** 2 fair
**Rating:** 6
**Confidence:** 4

**Summary:**

This paper studies the problem of learning personalized policies on observational data collected from heterogeneous multiple sources. To ensure privacy and other data safety requirements, this paper studies the problem under a specific federated setting with a central server that collects no raw data from individual data sources.

First, based on a federated averaging algorithm, this paper proposes a policy learning algorithm that abides by the federation requirement.

Then a regret analysis is provided for the algorithm, which considers two notions called global regret and local regret. For both notions, finite sample regret upper bounds depending on quantified heterogeneity are presented.

Finally, experimental results verify the dependence of the regret on the client heterogeneity.

**Strengths:**

**Significance**: this paper studies offline learning under heterogeneous data sources, an important problem setting in machine learning

**Quality**: the quality of the paper is good. Definitions are introduced without ambiguity; theoretical results looks solid to me; experimental details are given.

**Originality**: this paper considers the federation under the problem setting, which I deem as original

**Clarity**: this paper is very well written and easy for the readers to follow.

**Weaknesses:**

I did not detect any major technical flaw or major weakness in this paper.

Still, I have a few questions I hope the author can address. Please see the Questions session.

Furthermore, I think this paper, as a theoretical work, would significantly benefit from adding a sketch of proof for its main results.

**Questions:**

I think the problem setting is original. However, it is unclear to me what the technical novelty of this paper is.

Specifically, in the analysis/proof of the theorems, does there exist any technical challenge and how are they resolved?

Any novel trick adopted?

It would be great if the author could elaborate on this.

---

> ### Author Response · Authors · 2023-11-17
>
> We express our sincere gratitude to the reviewer for dedicating their time to our submission and for their positive feedback. We are committed to addressing their questions thoughtfully.
>
> *Answering Questions:*
>
> We appreciate the reviewer's suggestion regarding the inclusion of a proof sketch in the main paper, given our emphasis on the theoretical contributions. While we did provide a brief general proof sketch in Appendix C.1.4 due to space constraints, we acknowledge the potential value of having such a sketch in the main paper could better highlight our theoretical contributions. Although the devil is in the details, we attempt to provide an overview of some of the high-level technical challenges and novelty.
>
> In the standard approach for proving finite-sample regret bounds in offline policy learning, the focus is on establishing uniform concentration around a proper notion of empirical complexity, which is then further bounded by class-dependent vanishing rates (as seen in Zhou et al. 2023). Typically, offline policy learning in the standard setting involves bounding the Rademacher complexity of an appropriate policy value-based function class. However, this approach is not applicable to our scenario where the data may not come from the same source distribution. In our proof, we draw inspiration from the work on empirical risk bounds in multiple-source supervised learning settings, particularly Mohri et al. (2019), to identify the suitable notion of complexity—namely, the weighted Rademacher complexity of a policy value function class. Bounding the weighted Rademacher complexity required a nuanced understanding of how source heterogeneity, captured by skewness, influences the analysis. This required introducing the local data size scaling assumption stated in Assumption 2 to establish a sharper upper bound on the constant in Talagrand’s inequality to scale as $O(\sqrt{\text{skewness}/n})$. This is similar to how the primary theoretical contribution of the work of Zhou et al. (2023) in terms of regret bounds was to refine a similar constant in the analysis for the standard multi-action offline policy learning. Moreover, we note that we were able to establish rates with respect to the total sample size, rather than some other softer quantity of the sample sizes, such as the average or the minimum.
>
> While Mohri et al. (2019) provided a starting framework for a multiple-source analysis in supervised learning, the proof techniques for establishing class-dependent uniform concentration results in offline policy learning are typically more involved than those for empirical risk bounds in supervised learning. Our bounds necessitate more complex Dudley-type chaining arguments with applications of Talagrand’s inequalities, as evident in the proof of Proposition 2 in Appendix C. Moreover, our bounds required additional assumptions and understanding of doubly robust machine learning proof techniques to asymptotically bound the remaining approximation error, as seen in the proof of Proposition 4 in Appendix C.
>
> Furthermore, we established bounds for the notion of local regret, unique to our problem setting. This insight arises from recognizing the mismatch between global server-level performance and local client-level performance. We derive a local regret bound dependent on measures of distribution shift between clients, providing valuable insights into the value of information in heterogeneous client participation and how exactly heterogeneity affects policy performance for any given client. This exact quantification is highlighted in our Theorem 3 that decomposes the sources of heterogeneity at the population, environment, and treatment level. We also point to Theorem 4 in Appendix D.3 for an alternative local regret bound that does not require bounded inverse propensity weighted scores.
>
> *References:*
>
> Zhou, Zhengyuan, Susan Athey, and Stefan Wager. "Offline multi-action policy learning: Generalization and optimization." Operations Research 71.1 (2023): 148-183.
>
> Mohri, Mehryar, Gary Sivek, and Ananda Theertha Suresh. "Agnostic federated learning." International Conference on Machine Learning. PMLR, 2019.

---

> > ### Comment · Reviewer_38dU · 2023-11-21
> > **Response to rebuttal**
> >
> > Dear Authors,
> >
> > Thank you very much for responding to my questions!
> >
> > I really appreciate the explanation on the technical novelty of the paper. I would suggest adding the explanation in the revision of the paper, with detailed explanation and probably with mathematical details.
> >
> > Best,
> > Anonymous Reviewer

---

> > > ### Author Response · Authors · 2023-11-21
> > >
> > > We thank the reviewer for their suggestion. We added an explanation to Appendix C.1.4 (for now due to space constraints). If the reviewer requires any additional clarification to facilitate their evaluation of our work, please feel entirely free to pose any further questions.

---

### Official Review · Reviewer_D9wE · 2023-10-29

**Soundness:** 3 good
**Presentation:** 3 good
**Contribution:** 2 fair
**Rating:** 5
**Confidence:** 3

**Summary:**

This paper considers the problem of learning personalized policies from heterogeneous data sources in the federated setting.
They proposed a federated policy learning algorithm that averages locally trained policies with doubly robust policy evaluation. And, they provided finite-sample analysis on global and local regret bounds in terms of a mixture of distribution of clients and a relative distribution shift to all other clients, respectively.

**Strengths:**

1. This work provides finite-time regret upper bounds of the proposed policy learning algorithm in global and local perspectives, which characterize the effect of client skewness and client heterogeneity on global policy learning and individual policy learning of clients, respectively.
2. This work empirically demonstrated the effect of client heterogeneity on federated policy learning and suggested a skewed mixture for global policy training to overcome the performance degradation due to distribution shift.

**Weaknesses:**

1. It seems that the proposed algorithm is an application of FedAvg to CSMC. I wonder if there are some special challenges when extending offline policy learning algorithms to the federated setting with FedAvg, which have not been addressed in other federated learning or federated RL literature.
2. The regret analysis holds only when the algorithm converges to the optimal policy, which is not always guaranteed. In the paper, the authors claimed that it is still possible to achieve optimal policies via some additive term and appropriate choice of policy class, but it is vague and not convincing enough. It would be nice if you could provide further clarifications on the solution that enables the algorithm to achieve the optimal policy for general policy classes beyond linear policy classes.
3. It seems that Assumption 1-(c) requires a full exploration of all actions, which is quite strong given that there are some offline RL works suggesting that full coverage of state-action space is not necessary.
4. In the experiments, the algorithm was demonstrated in very limited settings. I wonder if this could be applied to more general and realistic settings. Also, it would be nice if you could compare the performance with other baseline algorithms.

**Questions:**

See the weaknesses above.

---

> ### Author Response · Authors · 2023-11-17
>
> We express our gratitude to the reviewer for dedicating their time to our submission. We are committed to addressing their concerns and questions thoroughly.
>
> *Addressing Weaknesses:*
>
> 1)
> Indeed, the algorithmic aspect of our work involves adapting federated averaging to offline policy learning with parametric function classes. The purpose of this portion was to present a feasible approach to offline policy learning in a federated setting that supplements our theoretical results. However, we do believe that there are additional questions at the intersection of federated learning and offline policy learning that merit further exploration.  For instance, the doubly robust offline policy learning approach employs a two-stage process of estimating nuisance functions before defining the optimization objective, and we simply employed a local nuisance function estimation procedure. Intriguing questions arise on aggregate nuisance parameter estimation among similar clients and how it could enhance performance. Furthermore, our FedAvg-based approach currently only applies to parametric policy classes. It would be valuable to develop a federated offline policy learning optimization procedure for non-parametric policy classes, such as decision trees, commonly used in real-world applications for their interpretability in decision-making.
>
>
> 2)
> The reviewer astutely pointed out that the optimal policy in the optimization procedure is not always guaranteed, with exceptions for specific cases like linear policy classes as examined in our experiments. However, our analysis can be readily extended to encompass scenarios where the optimal policy is not necessarily attainable, introducing the possibility of approximation errors. In the concluding paragraph of Section 6, we explain how this approximation error simply becomes an additional additive term in the regret bounds, that is, $\text{Regret bound}+\epsilon$, where $\epsilon$ represents the policy value optimality gap—indicating how far the policy value of the learned policy deviates from that of the optimal policy. While this approximation error may not necessarily diminish with increased data, with a judicious choice of policy class and the corresponding optimization procedure, the optimality gap can be rendered insignificant or of a similar order of magnitude as other terms in the regret bounds. This holds particularly true in moderately to highly heterogeneous environments, as the local regret bound contains an additive distribution shift term that is also inherently irreducible and captures the extent of heterogeneity.
>
>
> 3)
> We agree with the reviewer's observation that the overlap assumption (Assumption 1c) is strong. We note that this assumption is standard and commonly adopted in the offline policy learning literature to facilitate the identification of causal effects in observational studies. However, we agree with the reviewer's point that it may not be necessary as recent work has shown that, under a pessimism principle, overlap only under the optimal policy is sufficient. This insight is discussed in Section 3.4, following Assumption 1. The decision to adopt the more stringent standard assumption was made to simplify our analysis and maintain a focus on our contributions on the effects of data heterogeneity on policy learning.
>
>     Nevertheless, we acknowledge the intriguing questions that arise when considering what the pessimism principle would grant in our setting. Specifically, would it be necessary to have coverage under the locally optimal policy for each data source, or is coverage under the globally optimal policy sufficient? Additionally, how do the mixture weights impact the satisfaction of this assumption? We believe our work provides an initial study that opens up numerous follow-up questions in this direction.
>
>
> 4)
> Given that our algorithm builds on FedAvg, extending its application to more general settings, such as more general policy classes or cross-device scenarios with thousands of devices, is relatively straightforward. As for baseline comparisons in our empirical evaluation, we compared our approach with the baseline of standard offline policy learning where all the data is assumed to be consolidated into a single source. As our work is pioneering in studying offline policy learning with bandit feedback data in a federated setting, identifying direct comparison baselines relevant to our results proved challenging. Moreover, the objective of our experiments was to contrast the policies learned in a federated setting with those in a non-federated setting. While we stress that our primary contribution is theoretical rather than empirical, we acknowledge the potential for further development in the empirical evaluations. Your feedback is valuable, and we appreciate your understanding of the balance we aimed to strike between the theoretical and empirical aspects of our work.

---

### Official Review · Reviewer_eGKx · 2023-10-30

**Soundness:** 3 good
**Presentation:** 3 good
**Contribution:** 2 fair
**Rating:** 5
**Confidence:** 3

**Summary:**

This paper studies the problem of learning personalized decision policies from observational bandit feedback data across multiple heterogeneous data sources. In the federated setting, a central server aims to train a policy on data distributed across the data sources without directly accessing the raw data. The paper proposed a policy learning algorithm amenable to federation, based on the
federated averaging algorithm with local model updates provided by online cost-sensitive classification oracles.  Finite-sample upper bounds are provided for a notion of global regret, and local regrets for each agent. Empirical local and global regret bounds are compared across different experimental settings.

**Strengths:**

- Overall the paper is written with a good clarity. The authors studied a practically significant problem of learning personalized decision policies from multiple heterogeneous data sources, and demonstrated that the proposed algorithm can be extended to the federated setting.
- The assumptions and the regret upper bound analysis were detailedly described. In particular the theoretical analysis on local regret was a good compliment to the analysis on global regret, and shows their discrepancy due to client heterogeneity.
- The local and global regret bounds were compared empirically via simulated data.

**Weaknesses:**

- The upper bounds were based on a few detailed data assumptions, such as local ignorability, unconfoundedness, and overlap. Although the paper mentioned that some of the assumptions can potentially be relaxed, there is a lack of details on the discussion, and which assumption may not be relaxed fundamentally.
- In the theoretical analysis part (section 4-5), the main innovation part for the algorithm / estimator design and regret analysis in comparison to prior works was not clearly highlighted.
- The empirical evaluation did not include any comparisons with other baselines, or any real dataset, and the heterogeneous setting was fairly simply constructed.

**Questions:**

- Can the nuisance parameters be learnt jointly instead of being required to be separately known or estimated in the policy value estimates?

---

> ### Author Response · Authors · 2023-11-17
>
> We express our gratitude to the reviewer for dedicating their time to our submission. We are committed to addressing their concerns and questions thoroughly.
>
> *Addressing Weaknesses:*
>
> 1)
> Due to space limitations, our discussion regarding the necessity of certain assumptions may appear concise. Another contributing factor to this condensed discussion is that these assumptions are relatively standard within the offline policy learning literature for ensuring consistent policy value estimation. The cited works in this section serve as references indicating how some of these assumptions have been relaxed in recent literature. We acknowledge the need for more detailed exploration of these assumptions in future revisions.
>
> 2)
> Our primary contribution lies in the adaptation of offline policy learning to novel problem settings, specifically in the context of learning from multiple heterogeneous datasets and federated learning. The theoretical results presented in our work almost directly correspond with established findings in offline policy learning within the standard setting involving a single data source. We made efforts to underscore this parallel throughout the theoretical results section and the paper as a whole. Acknowledging the constraints posed by limited space, we recognize that our discussion might not have fully elaborated on this aspect. In response to your feedback, we have incorporated additional citations and context to enhance clarity. However, the space constraints still limit our ability to delve extensively into this matter. We appreciate the reviewer's input and we are open to addressing any specific concerns regarding comparisons with prior work here. We also kindly encourage you to refer to our responses to other reviewers' comments, where we delve more deeply into the question of the novelty of our work.
>
> 3)
> In our empirical evaluation, we compared our approach with the baseline of standard offline policy learning. This comparison involved assessing our method against the scenario where all the data is assumed to be consolidated in a single source, allowing for the application of standard offline policy learning methods. Given that our work is pioneering in the study of offline policy learning with bandit feedback data in a heterogeneous multiple-source and federated setting, we faced a challenge in identifying direct comparison baselines that we deemed relevant to our results. The objective of our experiments was to contrast the policies learned in a federated setting with those in a non-federated setting.
>
>     It's crucial to emphasize that our primary contribution lies in the theoretical domain rather than the empirical one. While our focus is on advancing theoretical understanding, we acknowledge that there is potential for further development in the empirical evaluations. We appreciate your understanding of the balance we aimed to strike between the theoretical and empirical aspects of our work.
>
> *Answering Questions:*
>
> 1. Indeed, if the central server can group similar clients, it can learn the nuisance parameters more effectively. The doubly robust approach relies on a two-stage approach of estimating nuisance functions prior to forming the optimization objective. We relied on a simple local nuisance function estimation procedure. However, there are interesting questions regarding the aggregate nuisance parameter estimation among similar clients and how that may improve performance.

---

### Official Review · Reviewer_oHDw · 2023-11-02

**Soundness:** 3 good
**Presentation:** 4 excellent
**Contribution:** 2 fair
**Rating:** 6
**Confidence:** 3

**Summary:**

This paper proposes an federated optimization procedure for off-policy learning over heterogeneous data sources, where the observational data is collected by multiple clients using different behaviour policies and stored only locally.
Specifically, the central server needs to maximize the doubly robust policy value estimator (Zhou et al. 2022b) over the (parametric) policy space, without transferring the raw observational data. In the proposed procedure, this is achieved by letting each client locally compute its model update by calling an online const-sensitive multi-class classification (CSMC) oracle and then the server performs a global update via a weighted average over the local model updates.

To the best of my knowledge, this is the first work that studies off-policy learning under federated setting.

**Strengths:**

The problem of off-policy learning in federated setting is well-motivated, and the authors have provided a solution with regret guarantee.

**Weaknesses:**

1. My main concern is the technical novelty, and I'd appreciate if the authors can provide more clarification on the contribution compared with the existing work mentioned below.

Based on my understanding, the main difference of this paper, compared with existing off-policy learning method using doubly robust estimator, e.g., Zhou et al. (2022b), lie in the optimization oracle. i.e., this paper needs to solve the CSMC problem over multiple heterogeneous clients to update the policy, instead of in a centralized setting. However, I am not sure if this has led to any technical challenge in obtaining the global regret bound in Theorem 1.

2. With CSC, we typicaly have a non-concave non-convex objective function to optimize, which makes finding the policy that maximizes Eq 8 difficult. Therefore, I expected the regret analysis to cover the situation where the FedAVG-CSMC procedure can only provide policy with certain approximation error. Moreover, even if the objective function is easy to optimize, it still seems to be unrealistic to assume we can obtain the exact maximizer of Eq 8 under federated setting, as this requires infinite number of iteration/communication rounds.
I'd appreciate it if the authors can provide more insights on how the current analysis can be extended to allow for approximation error of Eq 8.

**Questions:**

In Section 3.2, notations like $X^{C}, Y^{C}$ are not formaly defined.

The author mentioned that Assumption 3 can be easilly satisfied under regularity assumption. Can the authors provide a more formal description of the regularity assumption?

---

> ### Author Response · Authors · 2023-11-17
>
> We express our gratitude to the reviewer for dedicating their time to our submission. We are committed to addressing their concerns and questions thoroughly.
>
> *Addressing Weaknesses:*
>
> 1)
> Our work differs from prior work in two main ways. First, we explore offline policy learning (Zhou et al. 2023) under a novel setting of learning from multiple heterogeneous data sources. Prior studies, such as those by He et al. (2019) and Kallus et al. (2021), have considered a related, yet more restricted, problem of offline policy learning from multiple logging policies, where multiple datasets are also leveraged, but the underlying environment is assumed to be the same and only the action sample strategy differs across datasets. The more general problem setting we consider in this work has not been investigated, possibly due to the challenge of understanding the tradeoffs of a policy learned from diverse provenance. In our study, we are able to exactly quantify the impact of such heterogeneity on downstream performance for any target client by characterizing finite-sample upper bounds on suitable notions of regret. Note that this main result is independent of whatever optimization oracle is used to solve the policy learning problem we consider.
>
>     Second, we considered this policy learning problem in a setting where such heterogeneities may naturally arise in practice: in a federated setting. In federated settings, there tends to be less coordination on data collection procedures among clients (e.g., institutions, devices) compared to centralized learning settings. Consider a consortium of hospitals that conduct randomized control trials on a new treatment as an example; since these hospitals may not fully communicate their idiosyncratic treatment procedures, each hospital might conduct its unique trial on distinct populations, treatment formulations, and treatment assignment mechanisms. How should the central healthcare policymaker account for this heterogeneity across clients when training a treatment policy in a federated manner? With increasing privacy regulations, these coordination challenges may become more common, requiring federated policymakers to design policy learning procedures adaptable to such inherent heterogeneities for the intended purpose.
>
>     Our work demonstrates a practical way of adapting the federated learning framework to policy learning while accounting for underlying heterogeneities. As a first approach to this novel problem setting, we relied on the most well-known federated learning procedure for parametric models: federated averaging. We are not claiming that we are introducing a novel algorithm to the federated learning literature. We are simply adapting a well-known federated learning procedure to solve our optimization problem in a setting of practical interest for the above reasons. We also emphasize that the optimization problem is somewhat orthogonal to the technical analysis we conducted to arrive at our regret bounds. The regret bounds hold for the solution to the optimization problem and our optimization procedure demonstrates the feasibility and simplicity of solving this problem in the federated setting. However, that being said, we believe that the adaptation of FedAvg to offline policy learning is not entirely trivial and there are interesting questions at the intersection of federated learning and offline policy learning that warrant further exploration. For instance, the doubly robust offline policy learning approach employs a two-stage process of estimating nuisance functions before defining the optimization objective, and we simply employed a local nuisance function estimation procedure. Intriguing questions arise on aggregate nuisance parameter estimation among similar clients and how it could enhance performance. Furthermore, our FedAvg-based approach currently only applies to parametric policy classes. It would be valuable to develop a federated offline policy learning optimization procedure for non-parametric policy classes, such as decision trees, commonly used in real-world applications for their interpretability in decision-making.

---

> > ### Author Response · Authors · 2023-11-17
> > **Continuation**
> >
> > 2)
> > The reviewer astutely pointed out that the maximization objective is not necessarily concave, with exceptions for specific cases like linear policy classes as examined in our experiments. Therefore, determining the optimal policy in Eq. 8 may not be so straightforward. However, our analysis can be readily extended to encompass scenarios where the optimal policy is not necessarily attainable, introducing the possibility of approximation errors. In the concluding paragraph of Section 6, we explain how this approximation error simply becomes an additional additive term in the regret bounds, that is, $\text{Regret bound}+\epsilon$, where $\epsilon$ represents the policy value optimality gap—indicating how far the policy value of the learned policy deviates from that of the optimal policy. While this approximation error may not necessarily diminish with increased data, with a judicious choice of policy class and the corresponding optimization procedure, the optimality gap can be rendered insignificant or of a similar order of magnitude as other terms in the regret bounds. This holds particularly true in moderately to highly heterogeneous environments, as the local regret bound contains an additive distribution shift term that is also inherently irreducible and captures the extent of heterogeneity.
> >
> >
> > *Answering questions:*
> >
> > 1. In definition 1, we defined $(X^c, Y^c(a_1), \dots, Y^c(a_d))$ to be a sample from the local data-generating distribution $\mathcal{D}_c$ over the decision space $\mathcal{X}\times\mathcal{Y}^d$. However, for further clarity, in the updated draft we introduce this notation earlier in Section 3.1.
> > 2. Assumption 3 is a requirement on the estimation error rates of the two separately estimated nuisance functions. For simplicity, note that this assumption is sufficiently satisfied under any conditions that warrant $o(1/\sqrt{n})$ error rates for each nuisance function. Therefore, the vaguely mentioned regularity conditions essentially refer to whatever sufficient conditions enable efficient estimation of these two functions at the desired rates. For example, if the nuisance functions are parametric, then the least squares estimators under conditions like boundedness achieve O(1/n) rates. For non-parametric function classes conditions like belonging to a reproducing kernel Hilbert space with smooth kernels or having Lipschitz smoothness (along with their appropriate estimation procedures) suffice. The efficient estimation of these nuisance parameters is, although necessary, somewhat orthogonal to the policy learning. This is, in fact, the benefit of double machine learning.
> >
> >
> > *References:*
> >
> > Zhou, Zhengyuan, Susan Athey, and Stefan Wager. "Offline multi-action policy learning: Generalization and optimization." Operations Research 71.1 (2023): 148-183.
> >
> > He, Li, et al. "Off-policy learning for multiple loggers." Proceedings of the 25th ACM SIGKDD International Conference on Knowledge Discovery & Data Mining. 2019.
> >
> > Kallus, Nathan, Yuta Saito, and Masatoshi Uehara. "Optimal off-policy evaluation from multiple logging policies." International Conference on Machine Learning. PMLR, 2021.

---

### Meta-Review · Area_Chair_LeuY · 2023-12-05

**Metareview:**

This paper studies federated off-policy learning from bandit feedback. It is well written. The proposed approach is also analyzed and empirically evaluated. The discussion with reviewers revealed several recent works that chip away a lot of novelty of this paper. A proper comparison, both in theory and experiments, is needed. One work is

* [Multi-Task Off-Policy Learning from Bandit Feedback](https://proceedings.mlr.press/v202/hong23a.html)

This work is closely related because you can think of clients in federated learning as tasks in multi-task learning. There are differences between the reviewed and above papers. For instance, Hong et al. (2023) study a model-based approach based on a hierarchical Bayesian model while the reviewed paper is model free. However, there are also model-free techniques that take something like a hierarchy into account. One example is

* [Off-Policy Evaluation for Large Action Spaces via Conjunct Effect Modeling](https://proceedings.mlr.press/v202/saito23b.html)

The idea of relating actions through embeddings started in

* [Off-Policy Evaluation for Large Action Spaces via Embeddings](https://proceedings.mlr.press/v162/saito22a.html)

After the discussion with the reviewers, the novelty of this paper can be summarized as follows:

* The idea of improving statistical efficiency in off-policy learning, because the actions / tasks / clients are related, is out there. This is true for both model-based and model-free techniques.

* The main novelty of this paper is that it applies these ideas in federated learning. Federated learning papers often focus on communication and its cost. This is not the focus here.

* The fact that this paper missed a few recent papers may not be a big deal. However, it definitely indicates that this paper is much less novel than it seemed initially.

This was discussed with all reviewers and we agreed not to accept the paper.

**Justification For Why Not Higher Score:**

There are several recent papers that chip away a lot of novelty of this paper. A proper comparison, both in theory and experiments, is needed.

**Justification For Why Not Lower Score:**

N/A

---

### Decision · Program_Chairs · 2024-01-16

Reject